# Crumbs is an essential regulator of cytoskeletal dynamics and cell-cell adhesion during dorsal closure in *Drosophila*

David Flores-Benitez, Elisabeth Knust*

Max-Planck-Institute of Molecular Cell Biology and Genetics, Dresden, Germany

**Abstract** The evolutionarily conserved Crumbs protein is required for epithelial polarity and morphogenesis. Here we identify a novel role of Crumbs as a negative regulator of actomyosin dynamics during dorsal closure in the *Drosophila* embryo. Embryos carrying a mutation in the FERM (protein 4.1/ezrin/radixin/moesin) domain-binding motif of Crumbs die due to an overactive actomyosin network associated with disrupted adherens junctions. This phenotype is restricted to the amnioserosa and does not affect other embryonic epithelia. This function of Crumbs requires *D*Moesin, the Rho1-GTPase, class-I p21-activated kinases and the Arp2/3 complex. Data presented here point to a critical role of Crumbs in regulating actomyosin dynamics, cell junctions and morphogenesis.

*For correspondence: knust@mpi-cbg.de

**Competing interests:** The authors declare that no competing interests exist.

## Introduction

Dorsal closure (DC) in the *Drosophila* embryo is an established model for epithelial morphogenesis. The power of *Drosophila* genetics and cell biological tools have contributed to understand how signalling pathways, cell polarity and cell adhesion regulate the coordinated movements of two epithelial sheets, the epidermis and the amnioserosa (AS), a transient extraembryonic tissue [reviewed in (*Ríos-Barrera and Riesgo-Escovar, 2013*)]. More recently, elaborate biophysical techniques combined with high resolution imaging have elucidated how contractile forces are coordinated between cells in order to drive coherent changes in tissue morphology (*Sokolow et al., 2012*; *Jayasinghe et al., 2013*; *Fischer et al., 2014*; *Wells et al., 2014*; *Eltsov et al., 2015*; *Saias et al., 2015*). DC is a complex morphogenetic process taking about 2 hr, during which the epidermis expands dorsally to encompass the embryo. The process can be subdivided into three phases: i) elongation of the dorsal-most epidermal cells (DME) along the dorso-ventral axis; ii) contraction of AS cells and migration of the lateral epidermal cells towards the dorsal midline; iii) "zippering", i.e. adhesion of the epidermal cells from both sides on the dorsal midline [reviewed in (*Gorfinkiel et al., 2011*)]. Several forces contribute to these processes. First, pulsed contraction of AS cells produces a pulling force. These pulsed contractions are correlated with dynamic apical actomyosin foci, which transiently form in the apical medial cytocortex (*Kiehart et al., 2000*; *Hutson et al., 2003*; *Solon et al., 2009*; *Gorfinkiel et al., 2009*; *Blanchard et al., 2010*; *Heisenberg and Bellaiche, 2013*). Cells delaminating from the AS contribute additional pulling forces (*Muliyil et al., 2011*; *Sokolow et al., 2012*; *Toyama et al., 2008*). Second, a supracellular actomyosin cable, formed in the DME cells, surrounds the opening and provides contractile forces (*Hutson et al., 2003*; *Rodriguez-Diaz et al., 2008*). Finally, "zippering" of the two lateral epithelial sheets occurs, mediated by dynamic filopodia and lamellipodia (*Eltsov et al., 2015*; *Jacinto et al., 2000*).

**eLife digest** A layer of epithelial cells covers the body surface of animals. Epithelial cells have a property known as polarity; this means that they have two different poles, one of which is in contact with the environment. Midway through embryonic development, the *Drosophila* embryo is covered by two kinds of epithelial sheets; the epidermis on the front, the belly and the sides of the embryo, and the amnioserosa on the back. In the second half of embryonic development, the amnioserosa is brought into the embryo in a process called dorsal closure, while the epidermis expands around the back of the embryo to encompass it.

One of the major activities driving dorsal closure is the contraction of amnioserosa cells. This contraction depends on the highly dynamic activity of the protein network that helps give cells their shape, known as the actomyosin cytoskeleton. One major question in the field is how changes in the actomyosin cytoskeleton are controlled as tissues take shape (a process known as "morphogenesis") and how the integrity of epithelial tissues is maintained during these processes.

A key regulator of epidermal and amnioserosa polarity is an evolutionarily conserved protein called Crumbs. The epithelial tissues of mutant embryos that do not produce Crumbs lose polarity and integrity, and the embryos fail to develop properly.

Flores-Benitez and Knust have now studied the role of Crumbs in the morphogenesis of the amnioserosa during dorsal closure. This revealed that fly embryos that produce a mutant Crumbs protein that cannot interact with a protein called Moesin (which links the cell membrane and the actomyosin cytoskeleton) are unable to complete dorsal closure. Detailed analyses showed that this failure of dorsal closure is due to the over-activity of the actomyosin cytoskeleton in the amnioserosa. This results in increased and uncoordinated contractions of the cells, and is accompanied by defects in cell-cell adhesion that ultimately cause the amnioserosa to lose integrity. Flores-Benitez and Knust's genetic analyses further showed that several different signalling systems participate in this process.

Flores-Benitez and Knust's results reveal an unexpected role of Crumbs in coordinating polarity, actomyosin activity and cell-cell adhesion. Further work is now needed to understand the molecular mechanisms and interactions that enable Crumbs to coordinate these processes; in particular, to unravel how Crumbs influences the periodic contractions that drive changes in cell shape. It will also be important to investigate whether Crumbs is involved in similar mechanisms that operate in other developmental events in which actomyosin oscillations have been linked to tissue morphogenesis.

A plethora of proteins contribute to coordinate this highly dynamic morphogenetic process. Beside transcription factors, these include adhesion molecules and signalling pathways, a variety of cytoskeletal proteins and their regulators. Non-muscle myosin-II heavy chain (MHC) and the non-muscle myosin regulatory light chain (MRLC), encoded by *zipper (zip)* and *spaghetti-squash (sqh)*, respectively, are, together with the essential light chain, part of a force-producing molecular motor during DC [reviewed in (*Vicente-Manzanares et al., 2009*; *Liu and Cheney, 2012*)]. The small G-proteins of the Rho family, namely Rho1, Rac1, Rac2, Mtl, and Cdc42, regulate actomyosin activity and cell-cell adhesion (*Abreu-Blanco et al., 2014*; *Magie et al., 1999*; 2002). These GTPases stimulate myosin contraction through Rho-kinase (Rok) (*Mizuno et al., 1999*; *Harden et al., 1999*) or p21-activated kinase (*D*Pak) (*Harden et al., 1996*; *Conder et al., 2004*; *Hofmann et al., 2004*). They also modulate the Arp2/3 complex, which consists of seven subunits conserved in almost all eukaryotes (*Rotty et al., 2013*; *Veltman and Insall, 2010*). The Arp2/3 complex promotes the formation of densely branched, rapidly treadmilling actin filament arrays that, together with the Wiskott-Aldrich syndrome protein (WASP) and the WASP-family verprolin-homologous protein (WAVE), coordinate membrane-cytoskeleton dynamics (*Lecuit et al., 2011*; *Kurisu and Takenawa, 2009*; *Blanchoin et al., 2014*). The Arp2/3 complex also regulates endocytosis of *D*E-cadherin (*Georgiou et al., 2008*; *Leibfried et al., 2008*) and thus contributes to the regulation of the *zonula adherens* (ZA), an adhesion belt encircling the apex of epithelial cells (*Tepass et al., 1996*; *McEwen et al., 2000*; *Sarpal et al., 2012*). Moreover, the *Drosophila* WAVE homolog SCAR, the main activator of Arp2/3 in fly embryos (*Zallen et al., 2002*), is a downstream effector of Rac, Cdc42

and *D*Pak (*Lecuit et al., 2011*; *Kurisu and Takenawa, 2009*). *D*Pak, in turn, can also activate the Arp2/3 complex independently of SCAR (*Lecuit et al., 2011*; *Kurisu and Takenawa, 2009*; *Zallen et al., 2002*). Thus, the regulation of cell-cell adhesion and cytoskeleton activity is closely linked to each other.

During epithelial morphogenesis, mechanisms controlling cell polarity have to be set in place to ensure tissue integrity. One of the key regulators of epithelial cell polarity in the *Drosophila* embryo is the Crumbs protein complex. Its core components are the type I transmembrane protein Crumbs (Crb) and the scaffolding proteins Stardust (Sdt), *D*Lin-7 and *D*PATJ, which are conserved from flies to mammals [reviewed in (*Bulgakova and Knust, 2009*; *Tepass, 2012*)]. *Drosophila* embryos mutant for *crb* or *sdt* are unable to maintain apico-basal polarity in most of their epithelia (*Tepass and Knust, 1990*; *1993*; *Bachmann et al., 2001*; *Hong et al., 2001*). This leads to a complete break-down of tissue integrity due to failure in positioning and maintaining the ZA, followed by apoptosis in many tissues, e.g. the epidermis and the AS (*Grawe et al., 1996*; *Tepass and Knust, 1990*; *1993*; *Tepass, 1996*). Comparable defects in epithelial integrity are observed in mice lacking Crb2 or Crb3 (*Whiteman et al., 2014*; *Xiao et al., 2011*; *Szymaniak et al., 2015*). Conversely, over-expression of *Drosophila* Crb can lead to an expansion of the apical membrane domain, both in embryonic epithelial cells (*Wodarz et al., 1995*) and in photoreceptor cells (*Muschalik and Knust, 2011*; *Pellikka et al., 2002*; *Richard et al., 2009*). These results define Crb as an important apical determinant of epithelial cells. Besides a role in epithelial cell polarity, *Drosophila crb* controls tissue size in imaginal discs by acting upstream of the Hippo pathway [reviewed in (*Boggiano and Fehon, 2012*; *Genevet and Tapon, 2011*)], regulates morphogenesis of photoreceptor cells and prevents light-dependent retinal degeneration [reviewed in (*Bazellières et al., 2009*; *Bulgakova and Knust, 2009*)].

Crb contains in its extracellular domain an array of epidermal growth factor-like repeats, inter-spersed by four laminin A globular domain-like repeats. Its small cytoplasmic portion of only 37 amino acids contains two highly conserved motifs, a C-terminal PDZ (**P**ostsynaptic density/**D**iscs large/**Z**O-1) domain-binding motif (PBM), -ERLI, which can bind the PDZ-domain of Sdt and *D*Par-6 (*Li et al., 2014*; *Roh et al., 2002*; *Bulgakova et al., 2008*; *Bachmann et al., 2001*; *Hong et al., 2001*; *Kempkens et al., 2006*; *Ivanova et al., 2015*), and a FERM (protein **4**.1/**e**zrin/**r**adixin/**m**oesin) domain-binding motif (FBM) (*Klebes and Knust, 2000*), which can directly interact with the FERM-domain of Yurt (Yrt), Expanded (Ex) and Moesin (Moe) (*Laprise et al., 2006*; *Ling et al., 2010*; *Wei et al., 2015*). Our previous structure-function analysis of Crb using a fosmid-based approach revealed that the PBM is essential for the maintenance of cell polarity in embryonic epithelia (*Klose et al., 2013*). In contrast, the FBM is non-essential for normal development of most embryonic epithelia. At later stages of development, however, embryos with a mutation in the FBM fail to undergo DC (*Klose et al., 2013*). This phenotype now provides access to unravel additional functions of this highly conserved polarity regulator. Using live imaging and genetic analysis we elucidate a novel function of Crb as a key negative regulator of actomyosin dynamics during DC. Our results also further our understanding on the mechanisms that couple the regulation of the cytoskeleton and cell-cell adhesion with the control of embryonic morphogenesis.

## Results

### The FBM of Crb is essential for dorsal closure

We previously showed (*Klose et al., 2013*) that a fosmid covering the entire *crb* locus, named *foscrb*, completely rescues the lethality caused by the lack of endogenous *crb*. We also showed that a variant, in which the conserved tyrosine$_{10}$ in the FERM-domain binding motif (FBM) is replaced by an alanine (*foscrb*$_{Y10A}$ variant) does not rescue embryonic lethality. Interestingly, the fosCrb$_{Y10A}$ variant properly localises at the apical domain in most embryonic epithelia, which undergo normal morphogenesis (i.e. germ band elongation, salivary gland invagination). But later in development, germ band (GB) retraction, dorsal closure (DC) and head involution fail to occur properly (*Klose et al., 2013*). This indicated that the FBM of Crb fulfils a tissue- and stage-specific morphogenetic function in the embryo. Moreover, these defects appear to be independent of a putative Tyr phosphorylation, because another variant, in which the Y10 is replaced by a phenylalanine (*foscrb*$_{Y10F}$), completely rescues the embryonic lethality of *crb* mutants (*Klose et al., 2013*). To get a better

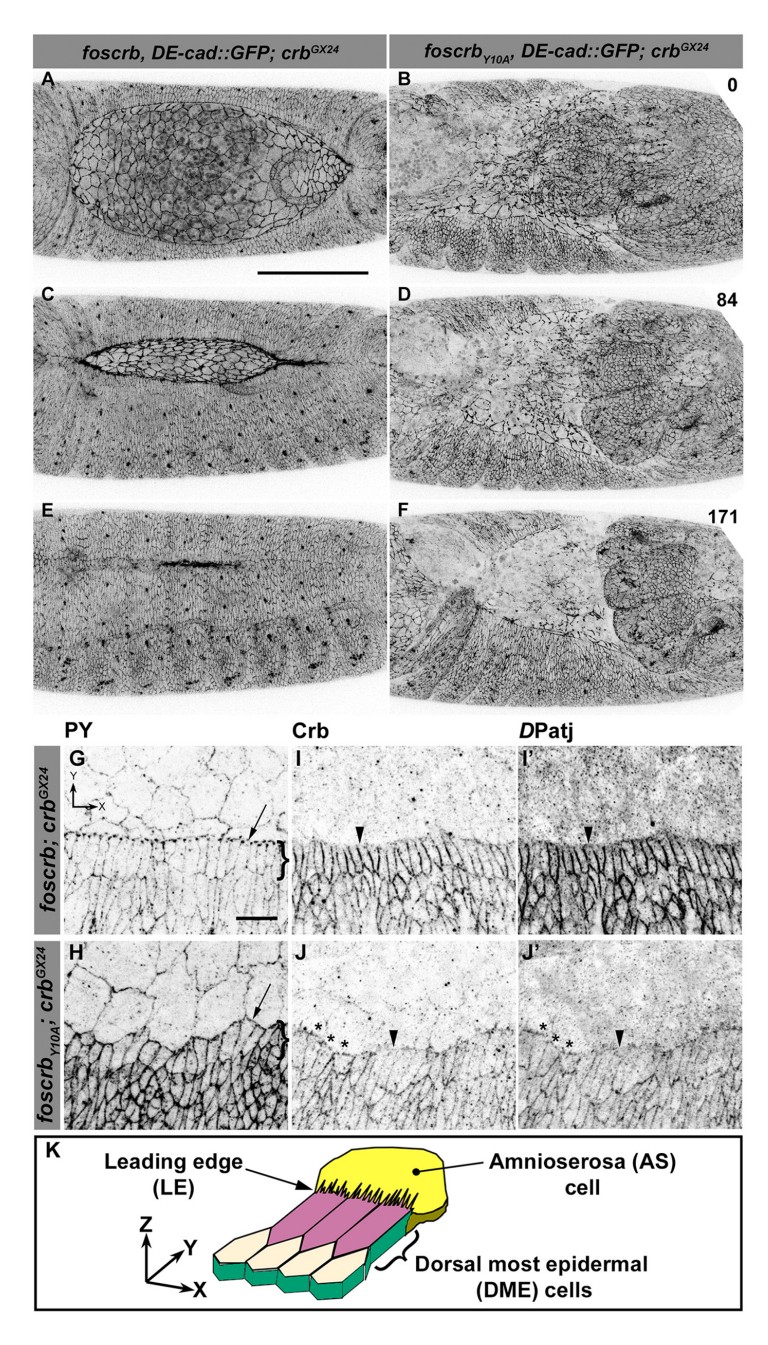

**Figure 1.** The FERM-binding domain motif (FBM) of Crb is essential for dorsal closure (DC). (A-F) Stills from dorsal views of live imaging of embryos expressing $D$E-cad::GFP. In all images the anterior part is towards the left. A, C and E, w;foscrb,DE-cad::GFP;crb$^{GX24}$ (**Video 1**). B, D and F, w;foscrb$_{Y10A}$,DE-cad::GFP;crb$^{GX24}$ (**Video 2**). All embryos were collected at the same time (1 hr collection), incubated at 28°C for 7 hr and imaged together. Numbers in (B,D and F) indicate the time in minutes for the corresponding row. While DC is completed in foscrb embryos (E), in foscrb$_{Y10A}$ embryos, the amnioserosa (AS) is disorganised and progressively lost (F). Scale bar: 100 μm. (G-J') Localisation of phosphotyrosine (PY), Crb and $D$Patj in the dorsal epidermis at the beginning of DC. In all images the AS is at the top (see reference axis in **G** and in the scheme K). (G, I,I') w;foscrb;crb$^{GX24}$. (H, J,J') w; foscrb$_{Y10A}$;crb$^{GX24}$. (K) Schematic representation of the dorsal epidermis at the beginning of DC indicating that the leading edge (LE) of the dorsal most epidermal (DME) cells is in contact with the AS. Arrows in (**G,H**) indicate LE of the DME (row of cells marked by brackets). The arrowheads indicate where the corresponding protein is absent from the LE (I-J'). The asterisks mark LE membranes positive for Crb (J) and $D$Patj (J') in foscrb$_{Y10A}$ mutant. Scale bar: 10 μm. Representative images from 8–12 different embryos for each genotype.

*Figure 1 continued on next page*

*Figure 1 continued*

The following figure supplement is available for figure 1:

**Figure supplement 1.** DC in *foscrb*$_{Y10F}$ embryos.

understanding of the mechanisms by which Crb regulates these morphogenetic processes, we performed detailed in vivo analyses of embryos expressing the different fosmid variants together with a *DE-cad::GFP* or a *DE-cad::mTomato* knock-in allele (*Huang et al., 2009*) in a *crb* null background (*crb*$^{GX24}$ or *crb*$^{11A22}$) (for simplicity, these are called *foscrb*, *foscrb*$_{Y10A}$ and *foscrb*$_{Y10F}$ from now on).

Because staging of embryos depends on morphological criteria, and *foscrb*$_{Y10A}$ mutant embryos show morphological defects, we imaged control and mutant embryos always in parallel, and stages were classified according to elapsed time after egg collection, i.e., after equal developmental times (see Materials and methods for details about staging and imaging). By the time *foscrb* embryos finish GB retraction (*Figure 1A*, *Video 1*), *foscrb*$_{Y10A}$ embryos (*Figure 1B*, *Video 2*) exhibit major defects in GB retraction, as revealed by a highly disorganised amnioserosa (AS) in which individual AS cells could hardly be followed. While *foscrb* embryos proceed through DC (*Figure 1C,E*, *Video 1*), those expressing the *foscrb*$_{Y10A}$ variant progressively lose the AS (*Figure 1D,F*) and ultimately fail to complete DC (*Video 2*). Embryos expressing the *foscrb*$_{Y10F}$ variant complete DC similar as *foscrb* embryos (*Figure 1—figure supplement 1*), indicating that the Y10A mutation specifically affects the progress of DC.

Various mechanisms have been documented to contribute to DC, including elongation of the dorsal most epidermal (DME) cells (*Riesgo-Escovar et al., 1996*). This elongation occurs normally in *foscrb* embryos, as revealed by phosphotyrosine (PY) staining associated with the ZA (*Figure 1G*). In contrast the DME cells of *foscrb*$_{Y10A}$ embryos do not elongate co-ordinately (*Figure 1H*). We analysed the localisation of Crb and *D*Patj at this stage. Both proteins are expressed at higher levels in the epidermis compared to the AS (*Figure 1I–J'*). In *foscrb* embryos, Crb (*Figure 1I*) and *D*Patj (*Figure 1I'*) are mostly absent from the leading edge (LE –*Figure 1I–I'* arrowheads) of the DME cells. In contrast, in *foscrb*$_{Y10A}$ embryos both Crb$_{Y10A}$ (*Figure 1J*, asterisks) and *D*Patj (*Figure 1J'*, asterisks) are detected at the LE, particularly in those cells that remain short, while both are removed in cells that elongate properly (*Figure 1J,J'*, arrowheads). Thus proper elongation of the DME cells fails in *foscrb*$_{Y10A}$ embryos.

## The FBM of Crb regulates filopodia formation and organisation of the supracellular actomyosin cable in the DME cells

Besides elongation of the DME cells, a complex actomyosin machinery is established at their LE. The DME cells extend filopodia and lamellipodia that are essential for correct 'zippering' (*Young et al., 1993*; *Edwards et al., 1997*; *Jacinto et al., 2000*; *Eltsov et al., 2015*). These filopodia, revealed by staining with an antibody against Stranded at Second [Sas (*Denholm et al., 2005*)], extend dorsally in *foscrb* embryos (*Figure 2A* arrow). In contrast, filopodia in *foscrb*$_{Y10A}$ embryos are disorganised and often absent (*Figure 2B*, empty arrowhead and arrowhead, respectively). This is confirmed by live imaging of embryos expressing a Venus-tagged Sas protein (*Video 3*). Filopodia of *foscrb*$_{Y10A}$ embryos are erratic, and some even appear to move out of the plane (*Video 3*, arrow in *foscrb*$_{Y10A}$ embryo), probably because of the loss of contact with the AS.

A key regulator of the number and length of filopodia during DC is the actin-elongation promoting protein Enabled (Ena) (*Gates et al., 2007*; *Nowotarski et al., 2014*; *Bilancia et al., 2014*; *Homem and Peifer, 2009*). Ena

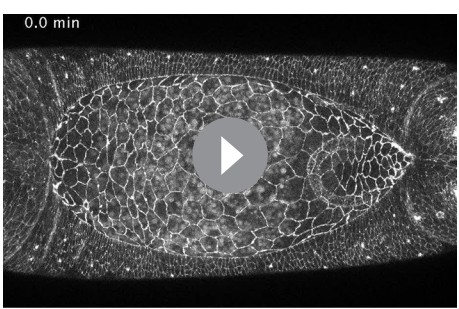

**Video 1.** Dorsal closure (DC) in a *w;foscrb,DE-cad::GFP;crb*$^{GX24}$ embryo. Note that the granules from the yolk are visible because of their strong autofluorescence in the green part of the spectrum. Time-lapse: 3.5 min; 12 fps.

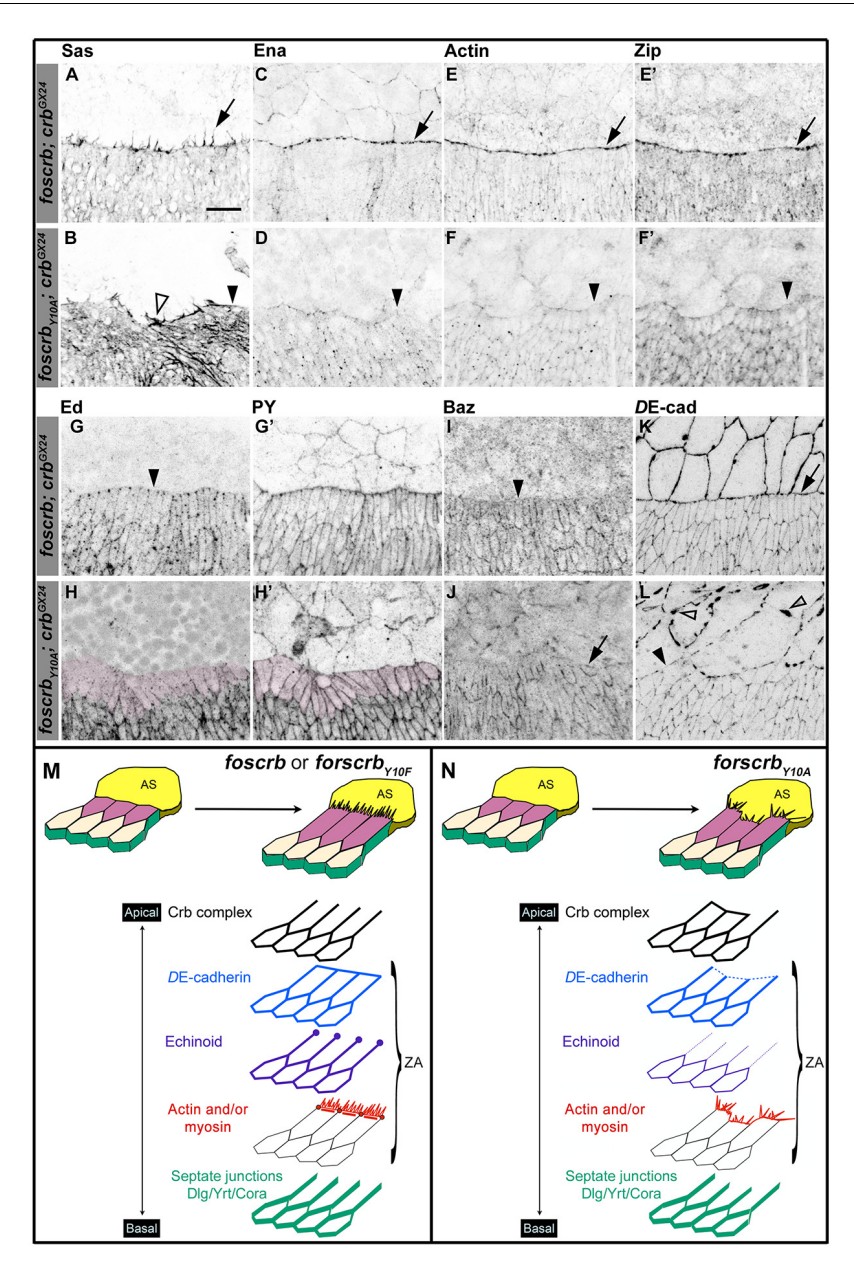

**Figure 2.** The FBM of Crb is important the establishment of the supracellular actomyosin cable at the LE of the DME cells during DC. (A-L) Localisation of Stranded at second (Sas, **A**,**B**), Enabled (Ena, **C**,**D**), Actin (**E**,**F**), Zipper (Zip, **E'**,**F'**), Echinoid (Ed, **G**,**H**), phosphotyrosine (PY, **G'**,**H'**), Bazooka (Baz, **I**,**J**), and *D*E-cadherin (*DE*-cad, **K**,**L**) at the beginning of stage 14. In all images the AS is at the top half, for the genotypes *w;foscrb;crb*$^{GX24}$ and *w;foscrb*$_{Y10A}$; *crb*$^{GX24}$. Filopodia extend dorsally in *foscrb* embryos (**A**, arrow), but in *foscrb*$_{Y10A}$ embryos filopodia are absent (**B**, arrowhead) or disorganised (**B**, empty arrowhead). Ena, Actin and Zip concentrate at the LE in *foscrb* embryos (**C**,**E** and **E'**, arrows), but these proteins are almost absent from the LE in *foscrb*$_{Y10A}$ embryos (**D**,**F** and **F'**, arrowheads). Ed is absent from the LE of *foscrb* embryos (**G**, arrowhead), but the DME cells of *foscrb*$_{Y10A}$ embryos show an important decrease of the protein (**H**, magenta overlay) though the PY staining is still clearly associated with the ZA in the same cells (**H'**, magenta overlay). Similarly, Baz decreases at the LE of *foscrb* embryos (**I**, arrowhead), but in *foscrb*$_{Y10A}$ embryos, the cells that do not elongate keep Baz at the LE (**J**, arrow), while other DME cells show a reduction of Baz (**J**, and **Figure 2—figure supplement 3**). *D*E-cad (mTomato signal) localises at all cell-cell contacts in *foscrb* embryos (**K**). However, in *foscrb*$_{Y10A}$, the *D*E-cad localisation is affected in both the dorsal epidermis (**L**, solid arrowhead) and the AS (**L**, empty arrowheads). Scale bar: 10 μm. (**M**) Schematic representation of the changes in DME cells at the beginning of DC in embryos expressing either fosCrb or fosCrb$_{Y10F}$. The elongation of the DME cells is accompanied by the removal of the Crb protein complex, Ed, Baz and the septate junction components from the LE. At the LE a supracellular actomyosin cable is established and filopodia extend dorsally and attach to the AS cells. Representative images from 8–12 different embryos for each genotype. (**N**) Schematic representation of the defects in the DME cells of embryos expressing the fosCrb$_{Y10A}$ variant. At the beginning of DC, the DME cells do not elongate uniformly. In the cells that do not elongate, the Crb protein

*Figure 2 continued on next page*

*Figure 2 continued*

complex and Baz remain at the LE. Reduced *D*E-cad suggest defects in the ZA function. Ed is dramatically reduced in DME cells, probably contributing to the absence of the supracellular actomyosin cable. Also, the DME cells exhibit disorganised filopodia. Nevertheless, the septate junction components are properly removed from the LE. The Crb protein complex is apical to the ZA, but Ed and the actomyosin cable are associated with the ZA.

The following figure supplements are available for figure 2:

**Figure supplement 1.** Localisation of Pyd, Dia and DAAM in *foscrb* and *foscrb*$_{Y10F}$ embryos.

**Figure supplement 2.** The FBM of Crb is important for the establishment of the supracellular actomyosin cable.

**Figure supplement 3.** Reduction of Baz in DME cells of *foscrb*$_{Y10A}$ embryos.

**Figure supplement 4.** Distribution of septate junction components in DME cells.

**Figure supplement 5.** Distribution of actomyosin and junctional components in DME cells of *foscrb*$_{Y10F}$ embryos.

---

concentrates at the LE of DME cells in *foscrb* embryos (*Figure 2C*, arrows). In contrast, Ena is strongly reduced at the LE of *foscrb*$_{Y10A}$ embryos (*Figure 2D*, arrowhead). Localisation of Ena at the LE depends on the ZA–associated protein Polychaetoid (Pyd) (*Choi et al., 2011*). However, Pyd localisation at the ZA shows no major difference in *foscrb* and *foscrb*$_{Y10A}$ embryos (*Figure 2—figure supplement 1A–B′′′*). The localisation of the formins Dia and DAAM, both involved in the growth of actin-based protrusions (*Matusek et al., 2006*; *Homem and Peifer, 2008*; *Liu et al., 2010*), is also similar in *foscrb* and *foscrb*$_{Y10A}$ embryos (*Figure 2—figure supplement 1C–F*). This suggests that different regulators of Ena are affected in *foscrb*$_{Y10A}$ mutant embryos.

In addition to filopodia, forces produced by a supracellular actomyosin cable at the LE contribute to DC (*Franke et al., 2005*; *Hutson et al., 2003*; *Kiehart et al., 2000*; *Jacinto et al., 2002*; *Young et al., 1993*). This supracellular cable, which contains actin (*Figure 2E*) and the non-muscle myosin II Zipper (Zip, *Figure 2E′*), is correctly formed in *foscrb* embryos (*Figure 2E,E′* arrows). However, it is virtually absent in *foscrb*$_{Y10A}$ embryos (*Figure 2F,F′*, arrowheads). Live imaging experiments using a *zipper::GFP* protein trap line (*Buszczak et al., 2007*; *Morin et al., 2001*) reveal that Zip::GFP appears homogenously along the LE in *foscrb* embryos. In contrast, it randomly concentrates in some segments along the LE of *foscrb*$_{Y10A}$ embryos (*Figure 2—figure supplement 2*). Together, these results show that

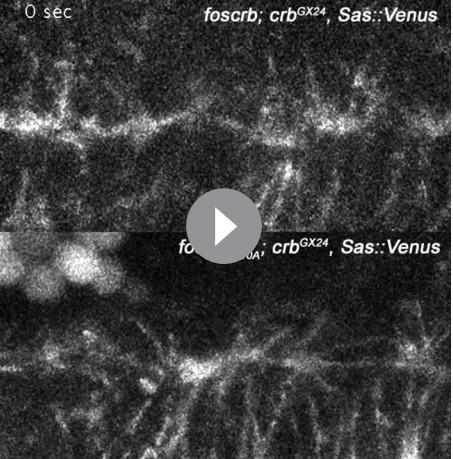

**Video 3.** Filopodia movement at the leading edge (LE) of the dorsal most epidermal (DME) cells in *w;foscrb*; *crb*$^{GX24}$,*Sas::Venus* (top) and *w;foscrb*$_{Y10A}$;*crb*$^{GX24}$,*Sas:: Venus* (bottom) embryos. The filopodia at the DME cells were followed for 5 min and the movie loops 6 times. Note that the filopodia in the *foscrb*$_{Y10A}$ embryo move randomly and some filopodia, like the one label with the arrow (bottom embryo), appear to detach and move out of the plane. Time-lapse: 10 sec; 8 fps.

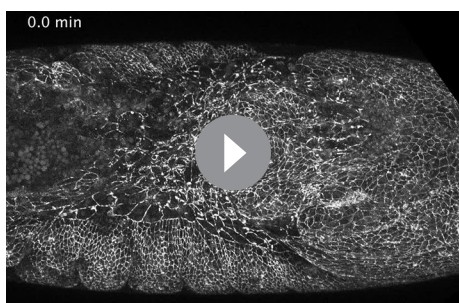

**Video 2.** Defective germ band (GB) retraction and DC phenotype in a *w;foscrb*$_{Y10A}$,*DE-cad::GFP;crb*$^{GX24}$ embryo. Time-lapse: 3.5 min; 12 fps.

the FBM of Crb is important for the generation and maintenance of actin-based protrusions and the correct organisation of the supracellular actomyosin cable at the LE.

The formation of the actomyosin cable at the LE depends on the removal of the adhesion protein Echinoid (Ed) from the LE and the AS cells (*Laplante and Nilson, 2011*; *Lin et al., 2007*). As expected, Ed in *foscrb* embryos is distributed as in wild type embryos (*Figure 2G*, arrowheads mark Ed absence at the LE). However, in *foscrb$_{Y10A}$* embryos, Ed levels are strongly reduced in the DME cells (*Figure 2H*, magenta overlay), even though the DME cells are still in contact with the AS, as revealed by PY staining (*Figure 2H'*). It has been suggested that the asymmetric distribution of Ed is essential to exclude the polarity protein Bazooka (Baz) away from the LE (*Laplante and Nilson, 2011*; *Pickering et al., 2013*). We found that, in contrast to *foscrb* embryos (*Figure 2I*, arrowhead), *foscrb$_{Y10A}$* embryos preserve Baz at the LE of those cells that fail to elongate (*Figure 2J*, arrow). In addition, there is a general reduction of Baz at the junctions of the DME cells of *foscrb$_{Y10A}$* embryos (*Figure 2—figure supplement 3*). Together, these results suggest that the FBM of Crb is important for Ed stability and hence Baz redistribution and amount in DME cells.

The asymmetric distribution of different proteins in the DME cells reflects the planar cell polarity of these cells, a feature that also includes the removal of septate junction (SJ) components from the LE (*Kaltschmidt et al., 2002*). We found that removal of Coracle (Cora), Discs Large (Dlg) and Yurt (Yrt) from the LE appears normal in the different fosmid variants (*Figure 2—figure supplement 4*), suggesting that not all aspects of the planar polarisation of the DME cells are affected in embryos expressing the *foscrb$_{Y10A}$* variant.

Ed, Baz and *D*E-cadherin (*D*E-cad) are all proteins associated with the ZA, which is essential in maintaining adhesion between the dorsal epidermis and the AS and for transmitting the forces generated during DC (*Gorfinkiel and Arias, 2007*; *Heisenberg and Bellaiche, 2013*; *Lecuit et al., 2011*). In *foscrb, D*E-cad localises at all cell-cell contacts, including the LE (*Figure 2K*, arrow). In *foscrb$_{Y10A}$* embryos, however, the *D*E-cad signal is strongly reduced at the LE (*Figure 2L*, solid arrowhead). Moreover, disruption of *D*E-cad suggests a discontinuous adhesion belt in the AS cells of these embryos (*Figure 2L*, empty arrowheads). The loss of *D*E-cad from the LE in the *foscrb$_{Y10A}$* embryos at this early stage is different from the normal redistribution of *D*E-cad that occurs at late stages during the zippering phase (*Gorfinkiel and Arias, 2007*). As expected, in *foscrb$_{Y10F}$* embryos, all proteins mentioned above localise as in *foscrb* embryos (*Figure 2—figure supplement 5*).

Taken together, these results show that the DC phenotype in *foscrb$_{Y10A}$* embryos is accompanied by defects in the establishment of the complex actomyosin apparatus at the LE of the DME cells and by the disturbance or even loss of different components of the ZA (schematised in *Figure 2M,N*).

## The FBM of Crb is essential for adhesion of the AS

As described above, GB retraction is defective and the AS is strongly disorganised in *foscrb$_{Y10A}$* embryos (*Figure 1F*). Because the AS is required during GB retraction (*Lamka and Lipshitz, 1999*; *Lynch et al., 2013*; *Scuderi and Letsou, 2005*), we analysed by live imaging whether the AS is affected before GB retraction.

In *foscrb* and *foscrb$_{Y10A}$* embryos, at the beginning of stage 11, AS cells are elongated along the antero-posterior axis (*Figure 3A,D*), highlighted by *D*E-cad::mTomato along the ZA (*Figure 3B,E*, arrows). In *foscrb$_{Y10A}$* embryos, however, the continuity of *D*E-cad::mTomato is frequently disrupted (*Figure 3E*, arrowhead) and *D*E-cad::mTomato additionally appears in large intracellular clusters of unknown identity (*Figure 3E*, concave arrowheads), which are never observed in *foscrb* embryos. As GB retraction proceeds, fragmentation of the ZA continues in the AS of *foscrb$_{Y10A}$* embryos and the tissue disintegrates (*Figure 3F* arrowheads and *Video 4*; and for a dorsal view of a different set of embryos see *Video 5*), while the dorsal aspect of *foscrb* embryos is covered by a continuous epithelial sheet (*Figure 3C*).

The defects of the AS in *foscrb$_{Y10A}$* embryos become very obvious in scanning electron micrographs (*Figure 3—figure supplement 1*). At stage 14, the AS forms a flat monolayer of epithelial cells in *foscrb* embryos (*Figure 3—figure supplement 1A,A'*). In contrast, in *foscrb$_{Y10A}$* embryos developed for the same period of time, the AS is completely disorganised. Large processes form, some of which extend over the caudal end of the embryos (*Figure 3—figure supplement 1B,B'*, arrow). Some isolated cells are visible over the epidermis (whether these are detached AS cells or migrating haemocytes was not determined –*Figure 3—figure supplement 1B*, arrowhead), while

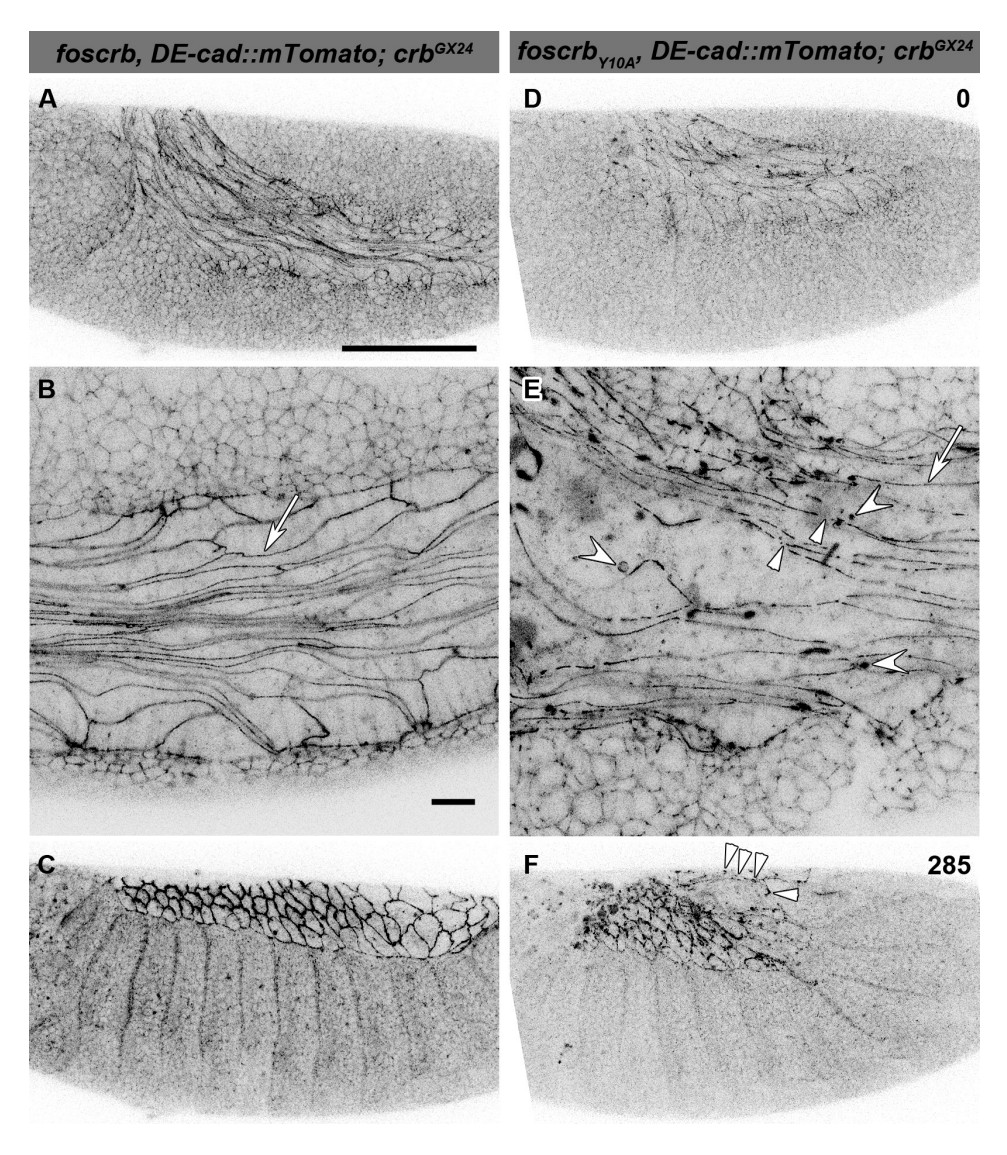

**Figure 3.** The FBM of Crb is important for the maintenance of the AS. (**A-F**) Stills from lateral views of live imaging of $D$E-cad::mTomato knock-in at the beginning of germ band (GB) retraction (**Video 4**). In all images the anterior part is towards the left, for the genotypes $w;foscrb,DE$-cad::mTomato;$crb^{GX24}$ and $w;foscrb_{Y10A},DE$-cad::mTomato; $crb^{GX24}$. All embryos were collected at the same time (1 hr collection), incubated at 28°C for 5 hr and imaged together. The numbers in (**D,F**) indicate the time in min. for the corresponding row. At stage 11 (**A,B,D,E**), the AS cells are elongated along the AP-axis, and $D$E-cad::mTomato localises along the ZA (**B,E**, arrows); in $foscrb_{Y10A}$ mutant, the continuity of $D$E-cad::mTomato along the ZA is lost (**E**, arrowhead) and $D$E-cad::mTomato is also found in large clusters (**E**, white concave arrowhead). At the end of GB retraction the AS covers the dorsal aspect of $foscrb$ embryos (**E**), but in $foscrb_{Y10A}$ (**F**), GB retraction is impaired and $D$E-cad::mTomato signal is fragmented in the AS (**F**, arrowheads). Scale bar: 100 µm, except for (**B,E**) 10 µm.

The following figure supplement is available for figure 3:

**Figure supplement 1.** The FBM of Crb is important for the integrity of the AS.

others have the appearance of apoptotic cells (**Figure 3—figure supplement 1B'**, concave arrowhead).

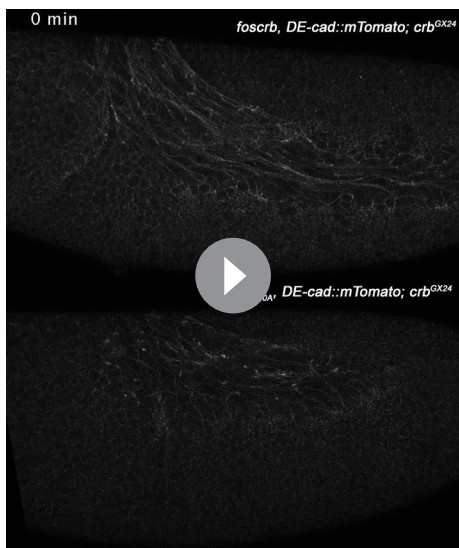

**Video 4.** Lateral views during germ band (GB) retraction in *w;foscrb,DE-cad::mTomato;crb^GX24* (top) and *w;foscrb_Y10A,DE-cad::mTomato;crb^GX24* (bottom) embryos. Time-lapse: 10 min; 8 fps.

Together, these observations suggest that cell-cell adhesion in the AS is strongly disrupted in *foscrb_Y10A* embryos, and define the FBM of Crb as an important regulator of cytoskeletal organisation and cell-cell adhesion of the AS.

## The FBM of Crb is essential for the integrity of the AS

Our scanning electron microscopy analyses suggest that the AS of *foscrb_Y10A* embryos undergo apoptosis. In order to determine whether apoptosis contributes to the disruption of the AS, we used the apoptotic reporter Apoliner, an RFP-GFP fusion protein localising at cell membranes of live cells. Caspase activation releases the GFP moiety, which is relatively unstable after cleavage, so dying cells have a stronger red appearance (*Bardet et al., 2008*; *Kolahgar et al., 2011*). Apoliner expression in the AS (specifically driven by the line GAL4^332.3) of *foscrb* embryos (*Video 6*) revealed some apoptotic cells at the posterior canthus at the end of GB retraction (*Figure 4A*, arrow). In *foscrb_Y10A* embryos developed for the same period of time, more apoptotic cells are visible, some of which detach (*Figure 4B*, arrowheads), while others remain attached to the posterior edge of the remaining AS (*Figure 4B*, arrow). As DC progresses in *foscrb* embryos, some apoptotic cells delaminate from the AS and are easily distinguished (*Video 6*, blinking arrows –some of these cells could be hemocytes with engulfed apoptotic debris, as reported previously [*Bardet et al., 2008*]). At this stage, almost all AS cells in *foscrb_Y10A* embryos are apoptotic (*Video 6*, compare embryos at 210 min). Finally, at the end of DC, the internalised AS cells are localised in a central rod-like structure in *foscrb* embryos and subsequently die by apoptosis (*Figure 4C*) [as has been reported for wild type embryos (*Reed et al., 2004*; *Shen et al., 2013*)], while in *foscrb_Y10A* embryos at this time point the remaining AS cells are completely disaggregated (*Figure 4D*). To summarise, the AS in *foscrb_Y10A* embryos breaks apart and undergoes premature apoptosis (*Video 6*), supporting the conclusion that an intact FBM is required for maintaining the integrity of the AS.

Several other processes are required for proper DC and integrity of the AS. At early stages, specification of the AS requires the U-shaped-group of genes (*hindsight –hnt, tail-up –tup, u-shaped –ush*, and *serpent –srp*), mutations in which produce phenotypes similar to those observed in *foscrb_Y10A* embryos (*Frank and Rushlow, 1996*; *Lamka and Lipshitz, 1999*; *Yip et al., 1997*; *Scuderi and Letsou, 2005*; *Lynch et al., 2013*). Hnt shows a strong and comparable expression pattern in the AS of *foscrb* and *foscrb_Y10A* embryos at early and late stages

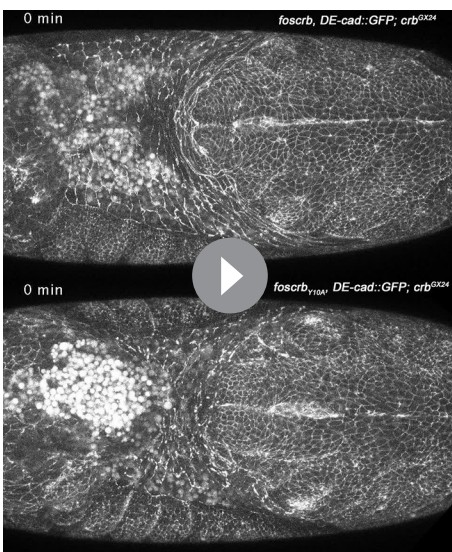

**Video 5.** Dorsal views during GB retraction and the beginning of DC in *w;foscrb,DE-cad::GFP;crb^GX24* (top) and *w;foscrb_Y10A,DE-cad::GFP;crb^GX24* (bottom) embryos. Note that the yolk aggregates are clearly visible because they have an intense autofluorescence in the green part of the spectrum. Time-lapse: 10 min; 8 fps.

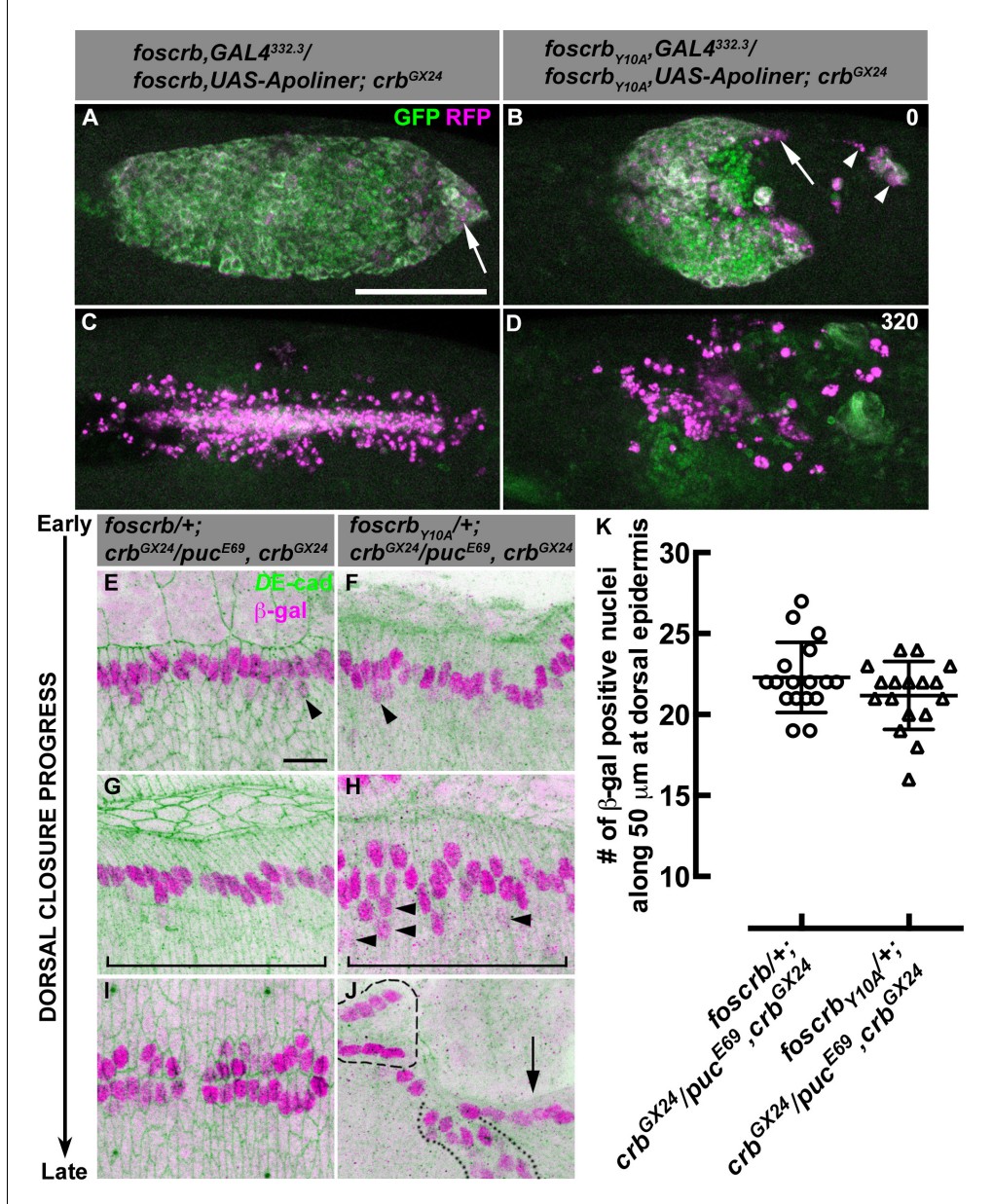

**Figure 4.** AS detachment in *foscrb_{Y10A}* embryos is accompanied by premature apoptosis. (A-D) Stills from dorsal views of live imaging of embryos in which the apoptotic reporter Apoliner is driven in the AS with the line GAL4^{332.3} (*Video 6*). Apoptotic cells in magenta appear more intense than their neighbours. In all images the anterior part is towards the left for the genotypes *w;foscrb,GAL4^{332.3}/foscrb,UAS-Apoliner;crb^{GX24}*, and *w;foscrb_{Y10A}, GAL4^{332.3}/foscrb_{Y10A},UAS-Apoliner;crb^{GX24}*. All embryos were collected at the same time (1 hr collection), incubated at 28°C for 7 hr and imaged together. The numbers in (B,D) indicate the time in minutes for the corresponding row. After GB retraction in *foscrb* embryos (A), some apoptotic cells are found mainly at the posterior canthus (A, arrow). In comparison, in *foscrb_{Y10A}* embryos, some of the cells that have detached from the AS (B, arrowheads), as well as those in the posterior edge of the AS (B, arrow), are apoptotic. As DC is completed in *foscrb* embryos (C), a significant portion of the internalised AS cells are apoptotic, while the remaining internalised cells are still localised in a rod-like structure along the dorsal part of the embryo. In contrast, in *foscrb_{Y10A}* embryos (D) all the remaining AS cells are apoptotic cells (the GFP signal in (D) does not belong to the AS). Scale bar: 100 μm. Representative images from 8–12 different embryos for each genotype. (E-K) Activation of the JNK pathway in the DME cells analysed with the enhancer trap *puc^{E69}* (β–galactosidase staining). *DE-cad* staining is in green. In all images anterior is to the left for the genotypes *w;foscrb/+;crb^{GX24}/puc^{E69},crb^{GX24}* and *w;foscrb_{Y10A}/+;crb^{GX24}/puc^{E69},crb^{GX24}*. From the beginning to the end of DC, Puc expression is normally induced on each side of the embryo in the single row of DME cells in both genotypes, and few positive β–gal nuclei appear below the row of DME cells (E,F, arrowheads). In *foscrb_{Y10A}* embryos at middle DC some β–gal positive cells appear below the DME cells (H, arrowheads). When DC is completed in *foscrb* embryos (I), a single row of cells on each side of the embryo is β–gal positive, even in *foscrb_{Y10A}* embryos, independently of whether the epidermis contacted the corresponding segment of the epidermis on the dorsal midline (J, dashed line), bunched on the same side of the embryo (J, dotted line) or fail to touch

*Figure 4 continued on next page*

Flores-Benitez and Knust. eLife 2015;4:e07398. DOI: 10.7554/eLife.07398

*Figure 4 continued*

the complementing segment (**J**, arrow). Scale bar: 10 μm. (**K**) No significant difference in the number of $\beta$–gal positive nuclei at middle DC along 50 μm at the dorsal epidermis (indicated by the brackets in **G,H**), mean ± SD, n= 17 embryos per genotype.

The following figure supplements are available for figure 4:

**Figure supplement 1.** Hindsight expression in *foscrb* and *foscrb*$_{Y10A}$ embryos.

**Figure supplement 2.** Localisation of integrin $\beta_{PS}$ in the AS of *foscrb* and *foscrb*$_{Y10A}$ embryos.

**Figure supplement 3.** Localisation of *D*Patj and Yrt in the dorsal epidermis.

**Figure supplement 4.** JNK signalling is normal in *foscrb*$_{Y10F}$ embryos.

(*Figure 4—figure supplement 1*), even in the detached AS cells of *foscrb*$_{Y10A}$ embryos (*Figure 4—figure supplement 1D*, arrowhead). This indicates that fate specification is not affected in *foscrb*$_{Y10A}$ embryos.

AS integrity also requires integrin-mediated attachment to the yolk sac membrane (*Reed et al., 2004*). Therefore, we analysed the localisation of integrin-$\beta_{PS}$, and found no major differences between *foscrb* and *foscrb*$_{Y10A}$ embryos (*Figure 4—figure supplement 2A,B*).

Yrt function is also important during DC, and zygotic *yrt* mutants have DC defects (*Hoover and Bryant, 2002*), similar to the ones observed upon Crb over-expression in the AS (*Harden et al., 2002*; *Wodarz et al., 1995*). Because Yrt is a FERM protein that negatively regulates Crb by directly interacting with its FBM (*Laprise et al., 2006*), Yrt appeared as a likely candidate in mediating the *foscrb*$_{Y10A}$ mutant phenotype. Yrt localises at the lateral domain and concentrates towards the apical aspect in a Crb-dependent manner from stage 13 onwards (*Laprise et al., 2006*). We found that independently of the fosmid genotype, Yrt concentrates correctly towards the apical aspect of the cells (*Figure 4—figure supplement 3*). Moreover, embryos expressing *foscrb* and lacking zygotic *yrt* show defects in DC

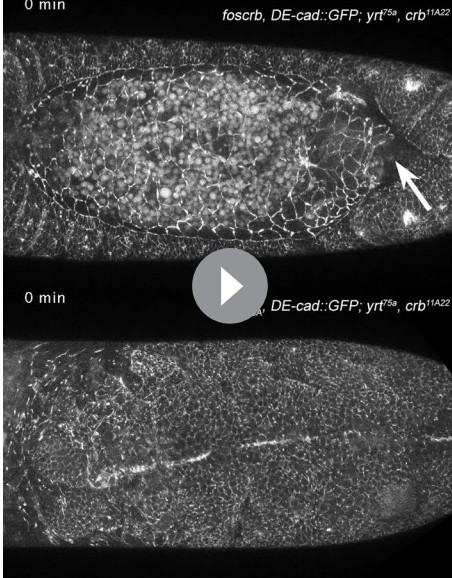

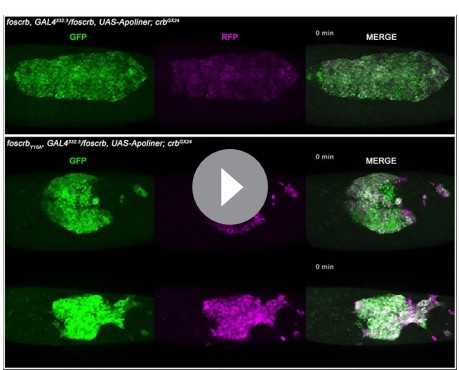

**Video 6.** Dorsal views during DC in *w;foscrb,GAL4*$^{332.3}$/*foscrb,UAS-Apoliner;crb*$^{GX24}$ (first row), and two examples of *w;foscrb*$_{Y10A}$*,GAL4*$^{332.3}$/*foscrb*$_{Y10A}$*,UAS-Apoliner;crb*$^{GX24}$ (second and third rows) embryos. Apoliner GFP signal is on the left (green), the RFP signal on the middle (magenta), and the merge on the right. At the time 210 min, the blinking arrows in the merge of the *foscrb* embryo indicate some apoptotic AS cells separated clearly. Time-lapse: 10 min; 8 fps.

**Video 7.** DC in *yrt*$^{\Delta75a}$ zygotic mutants expressing the different fosmids. *w;foscrb,DE-cad::GFP;yrt*$^{\Delta75a}$*crb*$^{11A22}$ (top) and *w;foscrb*$_{Y10A}$*,DE-cad::GFP;yrt*$^{\Delta75a}$*crb*$^{11A22}$ (bottom) embryos. The arrow in the top embryo marks the characteristic defects in the posterior canthus observed during DC in *yrt*$^{\Delta75a}$ zygotic mutants. In the *w;foscrb*$_{Y10A}$*,DE-cad::GFP;yrt*$^{\Delta75a}$*crb*$^{11A22}$ embryo the GB retraction and the DC phenotypes are comparable to the ones in the *w;foscrb*$_{Y10A}$*,DE-cad::GFP;crb*$^{GX24}$ (*Video 2*). Time-lapse: 6 min; 12 fps.

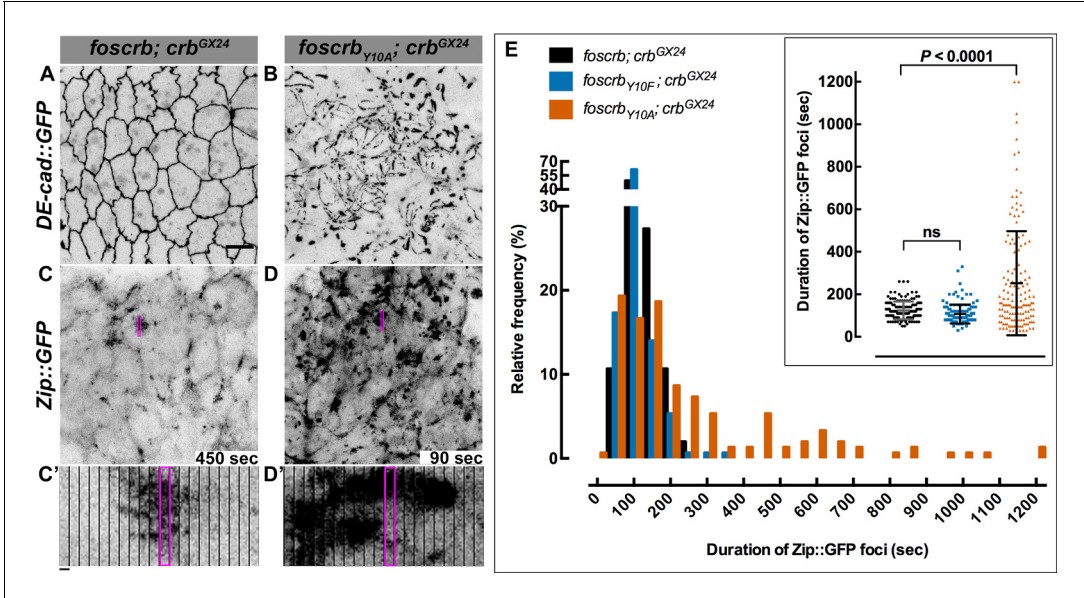

**Figure 5.** The FBM of Crb is essential for the regulation of actomyosin activity in the AS. Stills from views of the AS in live imaging of embryos expressing *DE*-cad::GFP knock-in (A,B, *Video 8*) or Zip::GFP (C-D', *Video 9*). In all images the anterior part is towards the left. Scale bar: 10 μm. (A) *w; foscrb,DE-cad::GFP;crb*$^{GX24}$. (B) *w;foscrb$_{Y10A}$,DE-cad::GFP;crb*$^{GX24}$. (C) *w;foscrb/Zip::GFP;crb*$^{GX24}$. (D) *w;foscrb$_{Y10A}$/Zip::GFP;crb*$^{GX24}$. The embryos were collected during 30 min, incubated at 28°C for 7 hr and imaged under the same conditions. The numbers in (C,D) indicate the time in seconds for the corresponding frame in *Video 9*. In *foscrb* embryos (A), *DE*-cad::GFP is localised at cell-cell junctions; but in *foscrb$_{Y10A}$* (B) embryos *DE*-cad::GFP continuity is strongly disturbed. (C',D') Kymographs of the Zip::GFP foci in the magenta box in (C,D). Scale bar in (C') 10 sec. (E) Histogram of the relative frequency of Zip::GFP foci duration during the pulsed contractions of the AS in *w;foscrb/Zip::GFP;crb*$^{GX24}$, *w;foscrb$_{Y10F}$/Zip::GFP;crb*$^{GX24}$ and *w; foscrb$_{Y10A}$/Zip::GFP;crb*$^{GX24}$ embryos. The graph in the insert shows all data points collected, and indicates the mean ± SD. ANOVA test followed by a Dunnett's multiple-comparison test; ns-not significant difference. *n* = 150 foci collected from each of the three different embryos.

The following figure supplement is available for figure 5:

**Figure supplement 1.** The FBM of Crb regulates the actomyosin activity in the AS.

mainly after GB retraction, when a failure in the zippering at the posterior canthus is patent (*Video 7*, arrow in the upper embryo). Despite this, the overall AS integrity is preserved during DC and most of the zippering is completed, leaving a hole only at the posterior canthus. This phenotype is completely different from the phenotype of *foscrb$_{Y10A}$* embryos described above (*Video 2*). Significantly, embryos with both the zygotic *yrt* mutant allele and the *foscrb$_{Y10A}$* variant do not show amelioration of the *foscrb$_{Y10A}$* phenotype (*Video 7*, bottom embryo). These embryos show strong defects in GB retraction, and the integrity of the AS is lost as development progresses. These results show that the DC phenotype of *foscrb$_{Y10A}$* embryos starts earlier in development and is more complex than that in *yrt* mutants, as the former fail in germ band retraction, lose the AS and do not progress on the zippering process. Thus, Yrt seems not to be involved in the phenotype of *foscrb$_{Y10A}$* embryos.

The AS regulates aspects of DME differentiation (*Stronach and Perrimon, 2001*) and embryos carrying mutations in components of the JNK signalling pathway show defective elongation of DME cells and fail to establish the supracellular actomyosin cable at the LE (*Riesgo-Escovar et al., 1996*; *Martín-Blanco et al., 1998*; *Ricos et al., 1999*; *Glise et al., 1995*; *Hou et al., 1997*; *Kockel et al., 1997*; *Reed et al., 2001*; *Ríos-Barrera and Riesgo-Escovar, 2013*). The mutant phenotype described here is characterised by defects in both the AS and the DME cells. To assess whether defects in the DME observed in *foscrb$_{Y10A}$* embryos are the result of impaired JNK signalling, we used the reporter line *puc-lacZ* (*Martín-Blanco et al., 1998*; *Ring and Martinez Arias, 1993*). At the beginning of DC, the DME cells of *foscrb* and *foscrb$_{Y10A}$* embryos are *β*-gal positive (*Figure 4E,F*), with few *lacZ*-positive nuclei in the row of cells ventral to DME cells (*Figure 4E,F*, arrowheads). At advanced DC, *foscrb* embryos still show a single row of *β*-gal positive cells (*Figure 4G*), while in

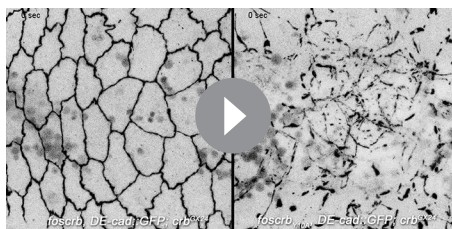

**Video 8.** Dorsal views during the pulsed contractions of AS cells in *w;foscrb,DE-cad::GFP;crb^GX24* (left) and *w;foscrb_{Y10A},DE-cad::GFP;crb^GX24* (right). Time-lapse: 10 sec; 15 fps.

*foscrb_{Y10A}* embryos β-gal positive nuclei can also be found at positions more ventral to the DME cells (*Figure 4H*, arrowheads). However, given that there is no significant difference in the number of β-gal positive nuclei along 50 μm of the dorsal epidermis between these genotypes (*Figure 4G,H*, brackets and 4K), we suggest that this phenotype is the result of aberrant elongation of the DME cells in *foscrb_{Y10A}* embryos (see for example *Figure 1H*). Accordingly, at the time when *foscrb* embryos complete DC, these embryos (*Figure 4I*) and *foscrb_{Y10A}* embryos exhibit a single row of β-gal positive cells on each side of the dorsal epidermis (*Figure 4J*). This is independent of whether the epidermis fuses on the dorsal midline (*Figure 4J*, encircled by dashed line), closes on the same side of the epidermis, thus causing bunching of the tissue (*Figure 4J*, encircled by dotted line) or does not touch any contra-lateral epidermis (*Figure 4J*, arrow). A normal activation of JNK signalling is also observed in *foscrb_{Y10F}* embryos (*Figure 4—figure supplement 4*), showing that JNK signalling appears to be normal in the DME cells of *foscrb_{Y10A}* embryos.

Taken together, these results support the conclusion that the FBM of Crb is an important regulator of the integrity and morphogenesis of the AS without affecting its specification during development.

## The FBM of Crb controls actomyosin dynamics in the AS

It has been previously shown that perturbing actomyosin dynamics of the AS cells interferes with normal DC (*Solon et al., 2009*; *Gorfinkiel et al., 2009*; *Fischer et al., 2014*). These dynamics, which are evident in stage 13 *foscrb* embryos (*Video 8*) similar as in wild-type embryos, is characterised by pulsed contractions of the AS cells. In *foscrb_{Y10A}* embryos, however, the pulsed contraction are difficult to follow, since individual cells can hardly be distinguished due to the highly disrupted ZA (*Video 8*, compare *Figure 5A and 5B*). Pulsed-contraction of wild-type AS cells has been correlated with a regular appearance and disappearance of medial actomyosin foci (*Blanchard et al., 2010*; *David et al., 2010*; *Solon et al., 2009*). These actomyosin foci are observed in *foscrb* embryos as revealed by Zip::GFP (*Video 9* and *Figure 5C*). Kymographs show that these foci are transient and disassemble after contraction (*Figure 5C',D'*). In contrast, the AS of *foscrb_{Y10A}* embryos shows more Zip::GFP foci (*Figure 5D*), some of which are more prominent (*Figure 5D'*, and *Figure 5—figure supplement 1* and *Video 10*). A similar behaviour was observed for F-actin (labelled with Utrophin::GFP (*Rauzi et al., 2010*) -data not shown). Importantly, analysis of the periodicity of foci formation shows that *foscrb* and *foscrb_{Y10F}* embryos have similar pulsed contractions, while *foscrb_{Y10A}* embryos have aberrant contractions, in that foci are more persistent (*Figure 5E*). These observations support the hypothesis that the AS of embryos

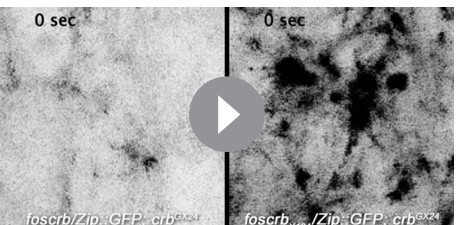

**Video 10.** Magnifications of a small group of cells shown in the *Video 11* to see in more detail the medial foci accumulation of Zip::GFP during the cell contraction. These magnifications (2X from original) were created using a bicubic algorithm in Fiji. *w;foscrb/Zip::GFP;crb^GX24* (left) and *w;foscrb_{Y10A}/Zip::GFP; crb^GX24* (right). Time-lapse: 10 sec; 15 fps.

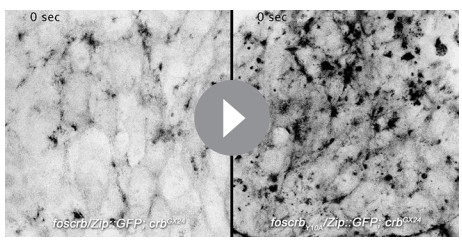

**Video 9.** Dorsal views during the pulsed contractions of AS cells in *w;foscrb/Zip::GFP;crb^GX24* (left) and *w;foscrb_{Y10A}/Zip::GFP;crb^GX24* (right). Time-lapse: 10 sec; 15 fps.

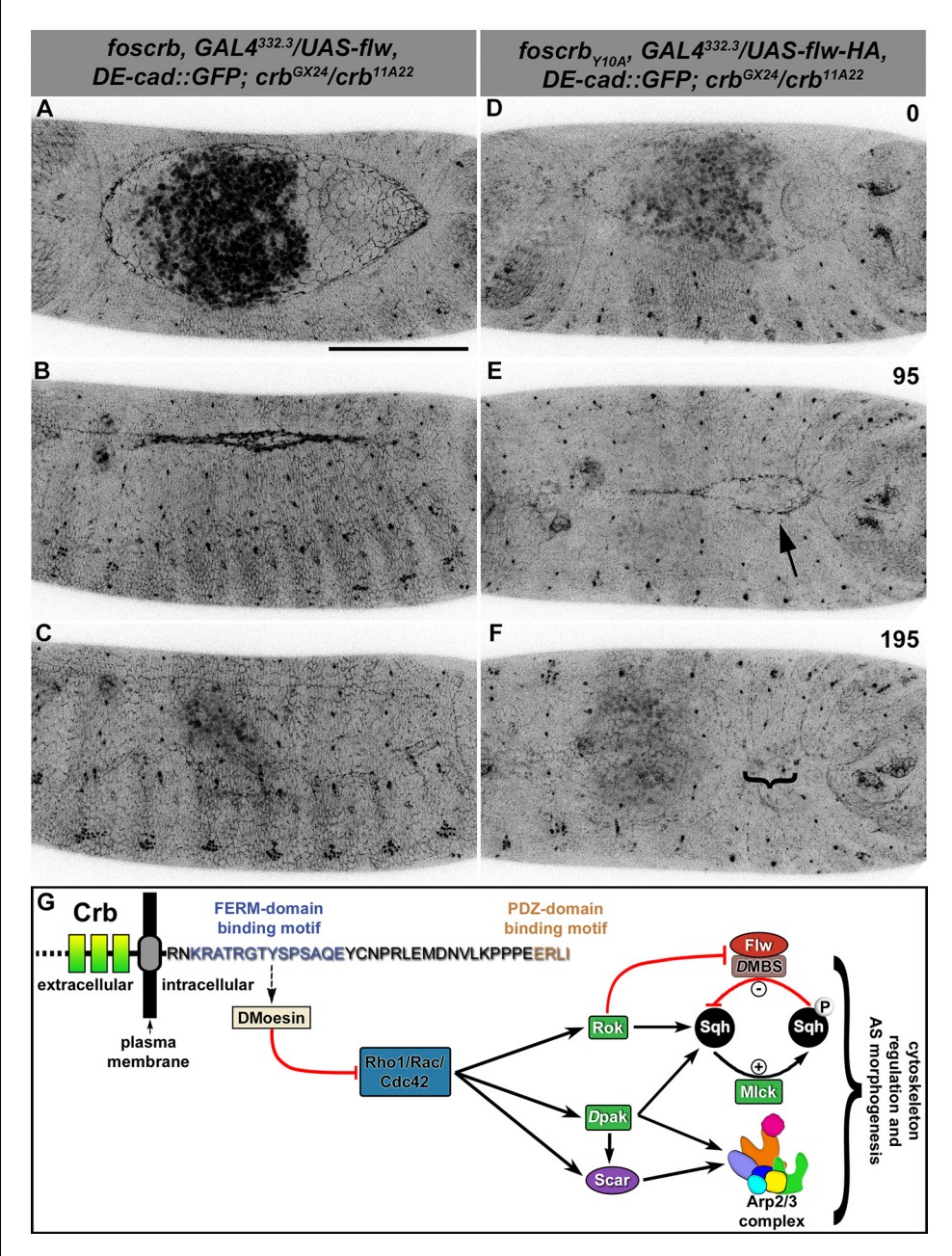

**Figure 6.** Expression of the myosin phosphatase Flapwing in the AS of *foscrb_Y10A* embryos suppresses the DC defects. (A-F) Stills from dorsal views of live imaging of embryos expressing *D*E-cad::GFP knock-in and Flw-HA in the AS cells under the control of the GAL4[332.3] driver (*Video 11*), for the genotypes *w;foscrb,GAL4[332.3]/UAS-flw-HA,DE-cad::GFP;crb[GX24]/crb[11A22],UAS-Act::RFP* and *w;foscrb_Y10A,GAL4[332.3]/UAS-flw-HA,DE-cad::GFP;crb[GX24]/crb[11A22],UAS-Act::RFP*. All embryos were collected at the same time (1 hr collection), incubated at 28°C for 7 hr and imaged together. The numbers on (D-F) indicate the time in minutes for the corresponding row. The overexpression of Flw-HA in the AS cells does not produce any obvious phenotype in *foscrb* (A-C) embryos, and it suppresses the DC defects in *foscrb_Y10A* (D-F) embryos; some defects found include an irregular zippering at the posterior canthus (E, arrow) as well as bunching of the dorsal epidermal (F, bracket). Scale bar: 100 μm. Representative images from 6–9 different embryos for each genotype. (G) Scheme of the possible pathways regulated by the FBM of Crb in the AS. Crb: Crumbs; Rok: Rho-kinase; *D*pak: *Drosophila* p21-activated kinase; Flw: Flapwing; *D*MBS: *Drosophila* myosin-binding-subunit; Sqh: spaghetti-squash; Mlck: myosin-light chain kinase. DOI: 10.7554/eLife.07398.031

The following figure supplement is available for figure 6:

*Figure 6 continued on next page*

*Figure 6 continued*

**Figure supplement 1.** Normal DC after Flapwing expression in the AS of *foscrb~Y10F~* embryos.

expressing the Crb~Y10A~ variant is under both constant and uncoordinated contraction.

The activity of non-muscle myosin-II (Zip) is mainly regulated by the phosphorylation state of the myosin-regulatory light chain [reviewed in (*Tan et al., 1992*)], encoded by the gene *spaghetti squash (sqh)*. Thus, if over-active actomyosin is responsible for the DC defects of *foscrb~Y10A~* embryos, we expect that expressing Flapwing (*flw*), the major *Drosophila* Sqh phosphatase (*Vereshchagina et al., 2004*), may suppress the DC defects. In fact, UAS-driven expression of Flw in the AS of *foscrb~Y10A~* embryos leads to a suppression of the DC phenotype (*Figure 6D–F*, *Video 11*), while it does not produce any evident dominant phenotype in *foscrb* or *foscrb~Y10F~* embryos (*Figure 6A–C*, and *Figure 6—figure supplement 1*). Interestingly, Flw over-expression also suppresses the disruption of the ZA in the AS (*Video 12*, compare B vs. D). This result supports our hypothesis that the FBM of Crb negatively regulates actomyosin activity in the AS.

Rho GTPases have been shown to stimulate myosin contraction by activating Rho-kinase (Rok) or the p21-activated kinase (*D*Pak), and are required for proper DC (*Mizuno et al., 1999*; *Harden et al., 1999*; 1996; *Conder et al., 2004*; *Magie et al., 1999*; 2002). To test whether Rho-GTPases are involved in the Crb-mediated DC phenotype, we expressed different versions of established Rho family effectors (see working model in *Figure 6G*) and examined their effects on DC in the embryonic cuticle, a suitable read-out of DC. We grouped the embryos according to their cuticle phenotype into two major categories (*Figure 7A*): 1) embryos with "DC-defect", which exhibit a range of defects from extensive dorsal opening (in which the mouthparts are exposed), to embryos with complete DC, which, however, still failed to hatch; and 2) embryos with "WT-like"

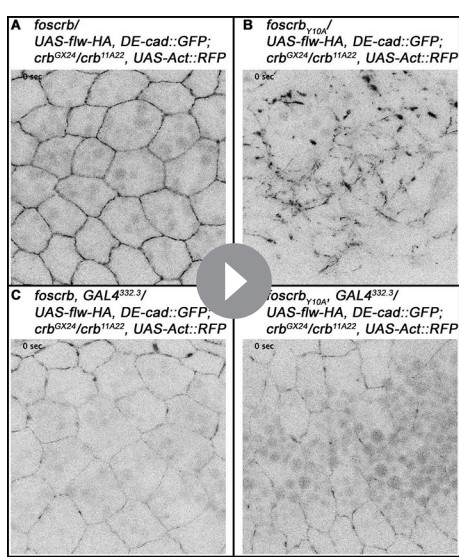

**Video 12.** Flw expression in the AS of *foscrb~Y10A~* embryos suppresses the disruption of the ZA. Dorsal views during the pulsed contractions of AS cells. The signal from the *UAS-Actin::RFP* is not shown. (A,B) Embryos that do not express the Flw and are trans-heterozygous for *D*E-cad::GFP; (A) *w;foscrb/UAS-flw-HA,DE-cad::GFP;crb^GX24^/crb^11A22^,UAS-Act::RFP* and (B) *w;foscrb~Y10A~/UAS-flw-HA,DE-cad::GFP;crb^GX24^/crb^11A22^, UAS-Act::RFP*. (C,D) Embryos that express Flw in the AS cells under the control of the GAL4^332.3^ driver; (C) *w; foscrb,GAL4^332.3^/UAS-flw-HA,DE-cad::GFP;crb^GX24^/ crb^11A22^,UAS-Act::RFP* and (D) *w;foscrb~Y10A~/ UAS-flw-HA,DE-cad::GFP;crb^GX24^/crb^11A22^,UAS-Act:: RFP*. Time-lapse: 10 sec; 15 fps.

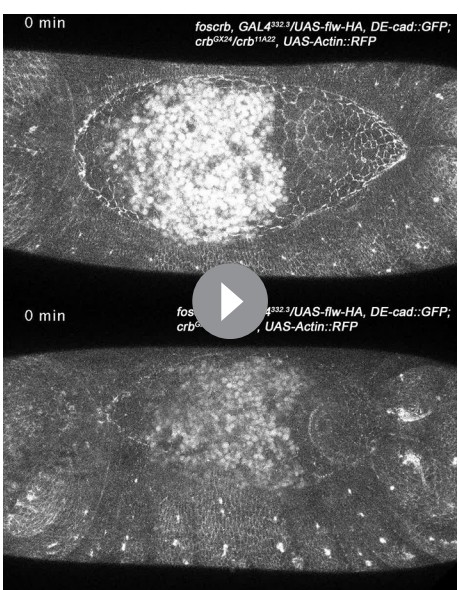

**Video 11.** Dorsal views during DC in embryos expressing the phosphatase Flw in the AS cells under the control of the GAL4^332.3^ driver. The signal from the *UAS-Actin::RFP* is not shown. *w;foscrb,GAL4^332.3^/UAS-flw-HA,DE-cad::GFP;crb^GX24^/crb^11A22^,UAS-Act::RFP* (top) and *w;foscrb~Y10A~,GAL4^332.3^/UAS-flw-HA,DE-cad:: GFP;crb^GX24^/crb^11A22^,UAS-Act::RFP* (bottom). Time-lapse: 5 min; 12 fps.

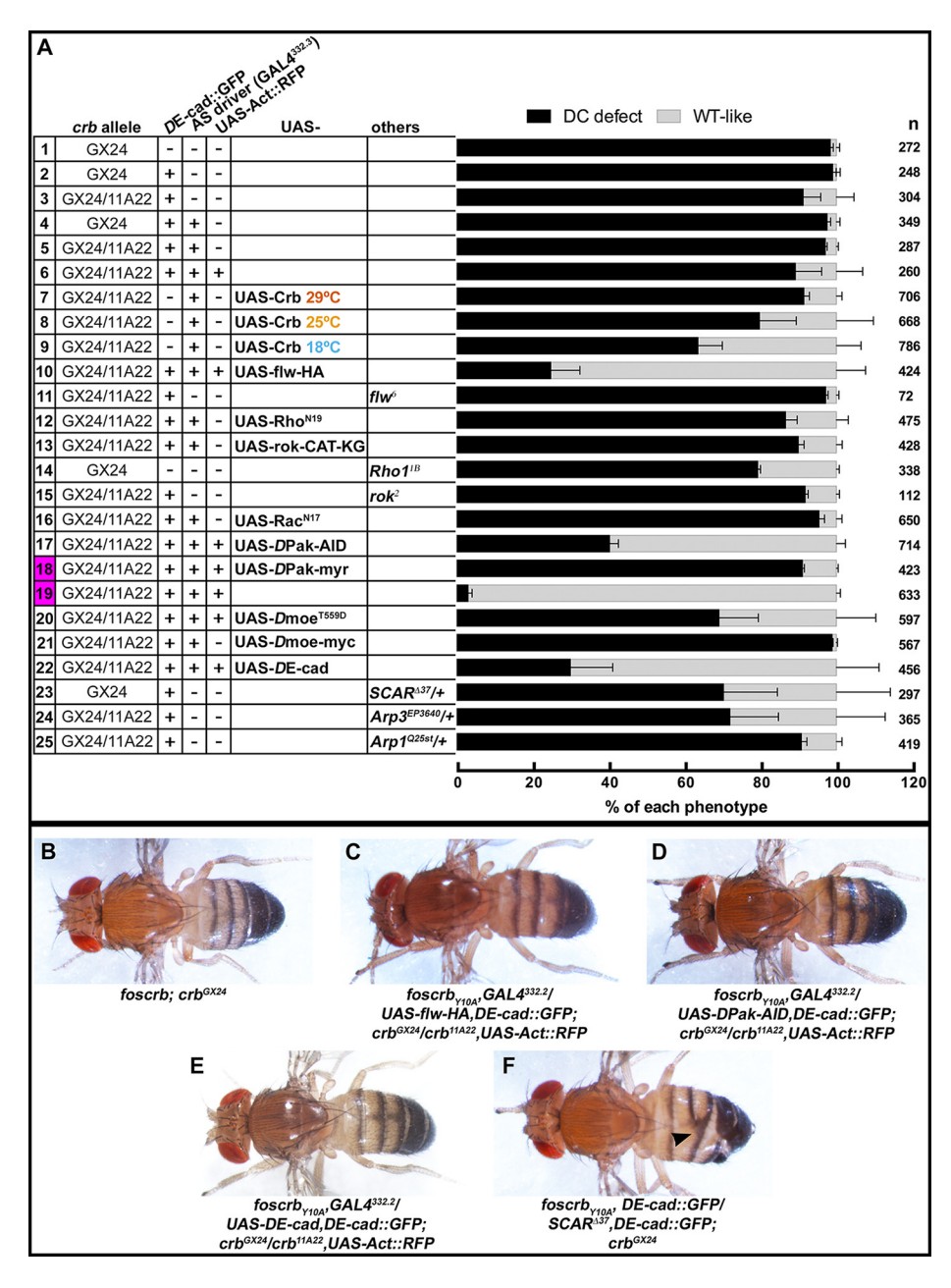

**Figure 7.** Reduction in actomyosin activity suppresses the DC defects in embryos expressing the *foscrb*<sub>Y10A</sub> variant. (**A**) Quantification of the defects observed in cuticle preparations from the genotypes indicated in the graph. For the complete genotype see *Figure 7—figure supplement 1*. The category "DC defect" includes a range of defects ranging from cuticles of embryos that completed DC but do not hatch, to cuticles with large DC openings. The category "WT-like" includes all larvae that hatch. For details about the classifications see *Figure 7—figure supplement 1*. Note that all the genotypes have the *foscrb*<sub>Y10A</sub> background, except the ones highlighted in magenta, numbers 18 and 19, that have the *foscrb* background. mean ± SD from 2–4 independent crosses. n = total number of cuticles counted for the indicated genotype. Note that suppression of the DC phenotype in *foscrb*<sub>Y10A</sub> embryos is particularly evident upon expression of Flw-HA (10), Pak-AID (17), and *D*E-cad (22). (**B-F**) Adult flies of the indicated genotypes. In (**F**), the arrowhead marks the defects in the dorsal abdomen.

The following figure supplements are available for figure 7:

**Figure supplement 1.** Reduction in the actomyosin activity suppresses the DC defects in embryos expressing the *foscrb*<sub>Y10A</sub> variant.

**Figure supplement 2.** Phosphorylated *D*Moesin levels are reduced in embryos expressing the *foscrb*<sub>Y10A</sub> variant.

*Figure 7 continued on next page*

*Figure 7 continued*

**Figure supplement 3.** Weak head phenotype of embryos expressing the *foscrb*$_{Y10A}$ variant.

cuticle, which includes all those that hatch (for more details about the different categories and phenotypes see *Figure 7—figure supplement 1*). Depending on the *crb* allelic combination, 89–98% of embryos expressing the *foscrb*$_{Y10A}$ variant fall into the "DC-defect" category (*Figure 7A*, 1st-6th black bars).

**Table 1.** Statistical analyses of the results shown in the *Figure 7—figure supplement 1*.

|  |  | Open cuticle | Dorsal hole | Closed but not hatched | Kinked larvae | WT-like |
|---|---|---|---|---|---|---|
| 1 | *foscrb*$_{Y10A}$;*crb*$^{GX24}$ |  |  |  |  |  |
| vs 14 | *foscrb*$_{Y10A}$/*Rho1*$^{1B}$;*crb*$^{GX24}$ | ** | ns | ns | * | *** |
|  |  |  |  |  |  |  |
| 2 | *foscrb*$_{Y10A}$,*DE-cad::GFP*;*crb*$^{GX24}$ |  |  |  |  |  |
| vs 23 | *foscrb*$_{Y10A}$,*DE-cad::GFP*/*SCAR*$^{Δ37}$,*DE-cad::GFP*;*crb*$^{GX24}$ | **** | ns | ns | **** | **** |
| vs 25 | *foscrb*$_{Y10A}$,*DE-cad::GFP*/*Arpc1*$^{Q25st}$,*DE-cad::GFP*;*crb*$^{GX24}$ | ns | ns | ns | ns | ns |
|  |  |  |  |  |  |  |
| 3 | *foscrb*$_{Y10A}$,*DE-cad::GFP*/+;*crb*$^{11A22}$/*crb*$^{GX24}$ |  |  |  |  |  |
| vs 11 | *flw*$^6$/*Y*/*w*$^*$;*foscrb*$_{Y10A}$,*DE-cad::GFP*/+;*crb*$^{11A22}$/*crb*$^{GX24}$ | **** | ** | ** | ns | ns |
| vs 15 | *rok*$^2$/*Y*/*w*$^*$;*foscrb*$_{Y10A}$,*DE-cad::GFP*/+;*crb*$^{11A22}$/*crb*$^{GX24}$ | * | ns | ns | ns | ns |
| vs 24 | *foscrb*$_{Y10A}$,*DE-cad::GFP*/+;*crb*$^{11A22}$,*Arp3*$^{EP3640}$/*crb*$^{GX24}$ | ns | ns | ns | ns | ** |
|  |  |  |  |  |  |  |
| 5 | *foscrb*$_{Y10A}$,*GAL4*$^{332.3}$/*DE-cad::GFP*;*crb*$^{GX24}$/*crb*$^{11A22}$ |  |  |  |  |  |
| vs 7 | 29°C *foscrb*$_{Y10A}$,*GAL4*$^{332.3}$/*UAS-Crb*$^{full\ length}$;*crb*$^{GX24}$/*crb*$^{11A22}$ | ns | ns | ns | ns | ns |
| vs 8 | 25°C *foscrb*$_{Y10A}$,*GAL4*$^{332.3}$/*UAS-Crb*$^{full\ length}$;*crb*$^{GX24}$/*crb*$^{11A23}$ | ns | ns | ns | ns | **** |
| vs 9 | 18°C *foscrb*$_{Y10A}$,*GAL4*$^{332.3}$/*UAS-Crb*$^{full\ length}$;*crb*$^{GX24}$/*crb*$^{11A24}$ | ns | ns | ns | ns | **** |
| vs 12 | *foscrb*$_{Y10A}$,*GAL4*$^{332.3}$/*DE-cad::GFP*;*crb*$^{GX24}$/*crb*$^{11A22}$,*UAS-Rho*$^{N19}$ | ** | ns | ns | ns | ** |
| vs 13 | *foscrb*$_{Y10A}$,*GAL4*$^{332.3}$/*DE-cad::GFP*;*crb*$^{GX24}$/*crb*$^{11A22}$,*UAS-rok.CAT-KG* | ns | ns | ns | ns | * |
| vs 16 | *foscrb*$_{Y10A}$,*GAL4*$^{332.3}$/*DE-cad::GFP*;*crb*$^{GX24}$/*crb*$^{11A22}$,*UAS-Rac*$^{N17}$ | ns | ns | ns | ns | ns |
| vs 21 | *foscrb*$_{Y10A}$,*GAL4*$^{332.3}$/*DE-cad::GFP*;*crb*$^{GX24}$/*crb*$^{11A22}$,*UAS-Dmoe-myc* | ns | ns | ns | ns | ns |
| vs 22 | *foscrb*$_{Y10A}$,*GAL4*$^{332.3}$/*UAS-DE-cad*,*DE-cad::GFP*;*crb*$^{GX24}$/*crb*$^{11A22}$ | **** | *** | ns | ns | **** |
|  |  |  |  |  |  |  |
| 6 | *foscrb*$_{Y10A}$,*GAL4*$^{332.3}$/*DE-cad::GFP*;*crb*$^{GX24}$/*crb*$^{11A22}$,*UAS-Act::RFP* |  |  |  |  |  |
| vs 10 | *foscrb*$_{Y10A}$,*GAL4*$^{332.3}$/*UAS-flw-HA*,*DE-cad::GFP*;*crb*$^{GX24}$/*crb*$^{11A22}$,*UAS-Act::RFP* | ns | **** | ns | * | **** |
| vs 17 | *foscrb*$_{Y10A}$,*GAL4*$^{332.3}$/*UAS-DPak-AID*,*DE-cad::GFP*;*crb*$^{GX24}$/*crb*$^{11A22}$,*UAS-Act::RFP* | ns | **** | ns | ns | **** |
| vs 20 | *foscrb*$_{Y10A}$,*GAL4*$^{332.3}$/*UAS-Dmoe*$^{T559D}$,*DE-cad::GFP*;*crb*$^{GX24}$/*crb*$^{11A22}$, *UAS-Act::RFP* | ns | * | ns | ns | **** |
|  |  |  |  |  |  |  |
| 18 | *foscrb*,*GAL4*$^{332.3}$/*UAS-Dpak-myr*,*DE-cad::GFP*;*crb*$^{GX24}$/*crb*$^{11A22}$,*UAS-Act::RFP* |  |  |  |  |  |
| vs 19 | *foscrb*,*GAL4*$^{332.3}$/*DE-cad::GFP*;*crb*$^{GX24}$/*crb*$^{11A22}$,*UAS-Act::RFP* | ** | *** | **** | ns | **** |

One-way-ANOVA analysis followed by a Dunnet's multiple comparisons test between the indicated categories of the different genotypes. Statistical significant difference indicated as follows: **ns** p>0.05; *p≤0.05; **p≤0.01; ***p≤0.001; ****p≤0.0001.

Using this read-out, we confirm that over-expression of the myosin phosphatase Flw in the AS strongly suppresses the DC defects of $foscrb_{Y10A}$ embryos. In fact, >75% hatch (*Figure 7A*, 10th vs. 6th bars) and even some $foscrb_{Y10A}$ adults eclose with no obvious defect (*Figure 7C*). Interestingly, cuticles from $foscrb_{Y10A}$ and hemi- or homozygous for the $flw^6$ allele show an enhanced DC phenotype in comparison with the $foscrb_{Y10A}$ with a wild type *flw* allele (*Figure 7A*, 3rd vs. 11th black bars: 91.2% to 97.1%; and *Figure 7—figure supplement 1*, 3rd vs. 11th black bars, completely open cuticle from 27.7% to 73.5%). These results support the conclusion that the FBM of Crb regulates the AS actomyosin dynamics by regulating myosin activity.

In line with this conclusion we found that over-expression of dominant-negative Rho (Rho$^{N19}$) or a kinase-dead Rok (Rok-CAT-KG) in the AS of $foscrb_{Y10A}$ increases the number of hatched larvae (*Figure 7A*, 5th vs. 12th and 13th gray bars: from 2.9% to 13.4% and 10.0%, respectively), and the proportion of embryos with open cuticles is reduced (*Figure 7—figure supplement 1*, 5th vs 12th and 13th black bars, from 52.7% to 23.6% and 28.5%, respectively). Moreover, $Rho1^{1B}$ hemizygosity effectively suppresses the DC defects of $foscrb_{Y10A}$ embryos (*Figure 7A*, 14th bar vs. 1st black bars, 79.2 vs. 98.3%). In contrast, $foscrb_{Y10A}$ embryos hemi- or homozygous for $rok^2$ show no suppression of the DC phenotypes (*Figure 7A*, 15th vs. 3rd bars), which suggests that *rok* deficiency may be deleterious in the $foscrb_{Y10A}$ background and that other morphological processes dependent on Rok could be affected (*Simões et al., 2010*; *Krajcovic and Minden, 2012*; *Mason et al., 2013*; *Bertet et al., 2004*). Similarly, over-expression of dominant-negative Rac1 (Rac1$^{N17}$) in the AS of $foscrb_{Y10A}$ embryos does not suppress the DC phenotype (*Figure 7A*, 16th vs. 5th bars) and even appears to increase the proportion of embryos with open cuticles (*Figure 7—figure supplement 1*, 5th vs. 16th black bars, from 52.7% to 72.9%). We assume that the phenotypic enhancement is due to an additive effect, since over-expression of Rac1$^{N17}$ in wild-type embryos results in DC defects (*Harden et al., 2002*).

An important regulator of cytoskeleton activity downstream of Rho GTPases is *D*Pak (*Hofmann et al., 2004*). Interestingly, over-expression of the auto-inhibitory domain of *D*Pak [*D*Pak-AID -(*Conder et al., 2004*)] in the AS of $foscrb_{Y10A}$ embryos leads to a very strong suppression of the DC phenotype, as 59% of those embryos hatch (*Figure 7A*, 17th vs. 6th bars), and even adult flies eclose (*Figure 7D*). Accordingly, over-expression of constitutive active *D*Pak (*D*Pak-myr) in the AS of otherwise viable *foscrb* embryos leads to embryonic lethality with >90% of embryos with a DC-defect (*Figure 7A*, 18th vs. 19th bars). These results indicate that unregulated activation of *D*Pak in the AS is sufficient to produce defects in DC, and that this kinase plays a major role in the defects observed in the $foscrb_{Y10A}$ embryos.

*D*Moe has been shown to antagonise the activity of the Rho pathway (*Speck et al., 2003*; *Neisch et al., 2010*; *Hipfner et al., 2004*). The participation of *D*Moe in the process under discussion here is supported by the fact that the FBM of Crb can recruit *D*Moesin (*D*Moe) to the membrane (*Médina et al., 2002*) and physically interacts with it (*Wei et al., 2015*), and that phosphorylated-*D*Moe (P-*D*Moe) is reduced in stage 11 $foscrb_{Y10A}$ embryos (*Klose et al., 2013*). This reduction in P-*D*Moe persists during DC (*Figure 7—figure supplement 2*). In line with this, over-expression of the phosphomimetic form *D*Moe$^{T559D}$ in the AS of $foscrb_{Y10A}$ embryos notably increases the number of larvae that hatch (*Figure 7A*, 6th vs. 20th gray bars, from 10.8% to 30.9%), while over-expression of *D*Moe does not ameliorate the DC defects in those embryos (*Figure 7A*, 21st bar). This suggests that the regulation of the cytoskeleton dynamics by Crb is mediated in part by the active form of *D*Moe. Together these results let us to conclude, that the FBM of Crb regulates actomyosin dynamics in the AS during DC by down-regulating the activity of the Rho1 pathway.

We wanted to exclude the possibility that the phenotypes observed are due to a dominant effect of the Y10A mutation. In fact, over-expression of full-length Crb$^{WT}$ in the AS of wild-type embryos leads to premature contraction of the AS and a DC phenotype (*Harden et al., 2002*; *Wodarz et al., 1995*). Driving the expression of UAS-Crb$^{WT}$ in the AS of $foscrb_{Y10A}$ embryos leads to a suppression of the DC phenotype, as >36% hatch at 18°C (*Figure 7A*, 8th and 9th bars vs. 5th gray bars), while inducing a stronger over-expression by maintaining embryos at 29°C does not ameliorate the $foscrb_{Y10A}$ phenotype (*Figure 7A*, 5th vs. 7th bars). These results show that the DC phenotype of $foscrb_{Y10A}$ embryos is due to loss of Crb function.

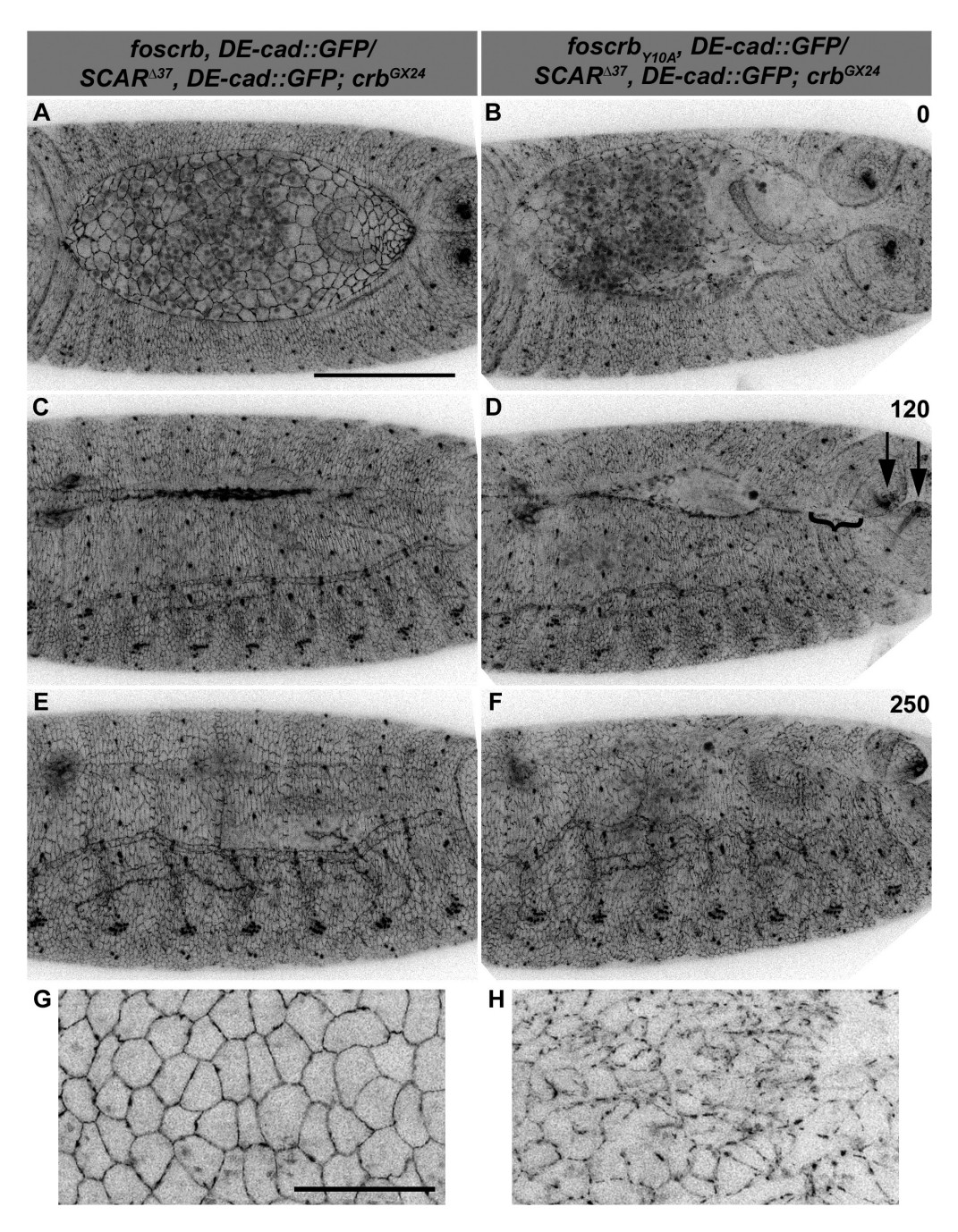

**Figure 8.** Reduction of the SCAR-Arp complex activity suppresses the DC defects and ameliorates the loss of *D*E-cadherin in the AS of embryos expressing the *foscrb_{Y10A}* variant. (**A**-**F**) Stills from dorsal views of live imaging of embryos expressing *D*E-cad::GFP knock-in and heterozygous for the *SCAR^{Δ37}* loss of function allele (*Video 13*). In all images the anterior is to the left, for the genotypes *w;foscrb,DE-cad::GFP/SCAR^{Δ37},DE-cad::GFP; crb^{GX24}* and *w;foscrb_{Y10A},DE-cad::GFP/SCAR^{Δ37},DE-cad::GFP;crb^{GX24}*. All embryos were collected at the same time (1 hr collection), incubated at 28°C for 7 hr and imaged together. The numbers in (**B,D,F**) indicate the time in minutes for the corresponding row. DC occurs normally in *foscrb* (**A,C,D**) embryos heterozygous for the *SCAR^{Δ37}* allele, and DC defects are suppressed in *foscrb_{Y10A}* (**B,D,F**) embryos; some defects still visible include the impaired GB retraction (compare **B** with **A**), asymmetric position of the posterior spiracles (**D**, arrows), and bunching of the dorsal epidermis (**D**, bracket). Scale bar: 100 μm. (**G,H**) Magnified views of AS from (**A,B**, respectively). Note that, in order to make the localisation of *D*E-cad::GFP more perceptible, the autofluorescence of the yolk (visible in **A,B**) was removed from the original stack by hand using Fiji. Scale bar: 100 μm. Representative images from 6–9 different embryos for each genotype.

## The FBM of Crb is essential for the stability of *DE*-cadherin in the AS

Besides an over-active actomyosin network, *foscrb*$_{Y10A}$ embryos exhibit interruptions in *DE*-cad distribution (**Figures 2L**, **3F** and **5B**). In addition some embryos show weak head-involution defects (**Figure 7—figure supplement 3**), a phenotype reminiscent to that of weak alleles of *shotgun* (*shg*) (the gene encoding *DE*-cad) (**Tepass et al., 1996**), *armadillo* (*arm*) (the gene encoding β-catenin) (**McEwen et al., 2000**) or *α-Cat* (**Sarpal et al., 2012**). Therefore we asked whether the DC phenotype of *foscrb*$_{Y10A}$ embryos could be rescued by restoring a functional adhesion belt. Over-expression of *DE*-cad in the AS of these embryos indeed can suppress the DC phenotype, as 70% of the larvae hatched (**Figure 7A**, 22$^{nd}$ vs. 6$^{th}$ bars), and even adult animals are obtained (**Figure 7E**).

A likely candidate of *DE*-cad regulation is the Arp2/3 complex, which has been shown to regulate endocytosis of *DE*-cad (**Georgiou et al., 2008**; **Leibfried et al., 2008**). In addition, reducing the activity of the Arp2/3 complex suppresses the DC phenotype of *α-Cat* mutants (**Sarpal et al., 2012**). Therefore, we tested the effects of removing one copy of *SCAR*, *Arp3* or *Arpc1* on the DC phenotype of *foscrb*$_{Y10A}$ embryos. Strikingly, *foscrb*$_{Y10A}$ embryos that are heterozygous for *SCAR*$^{Δ37}$ exhibit only minor defects in GB retraction (**Figure 8B**), partially restore *DE*-cad::GFP localisation in the AS (compare **Figure 8H** with **Figure 5B**) and completed DC (**Figure 8F**, **Video 13**). In fact, ~28% of these larvae hatch, as revealed by the cuticle phenotype (**Figure 7A**, 23$^{rd}$ vs. 2$^{nd}$ bar), and even some of the *w;foscrb*$_{Y10A}$,*DE*-cad::GFP/SCAR$^{Δ37}$,*DE*-cad::GFP;crb$^{GX24}$ develop into adult flies that exhibit defects in abdominal development (**Figure 7F**, arrowhead). A similar suppression was obtained in *foscrb*$_{Y10A}$ embryos heterozygous for *Arp3*$^{EP3640}$ (**Video 14**) (**Figure 7A**, 24$^{th}$ vs. 3$^{rd}$ bar). *foscrb* embryos heterozygous for *SCAR*$^{Δ37}$ or *Arp3*$^{EP3640}$ show normal DC (**Figure 8E** and **Video 14**).

In summary we could demonstrate that the DC phenotype of embryos expressing Crb$_{Y10A}$ is due to enhanced Rho-mediated actomyosin activity and reduced adhesion. Whether these two processes are linked or independent functions downstream of Crb remains to be discussed.

## Discussion

Dorsal closure is an ideal model to study how coordinated behaviour of epithelial sheets controls morphogenesis. Here we present data to show that a mutation in the FERM-domain binding motif of

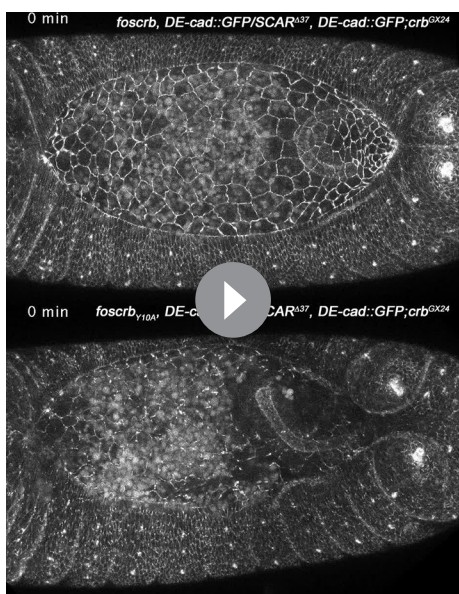

**Video 13.** Dorsal views during DC in embryos heterozygous for the *SCAR*$^{Δ37}$ allele. *w;foscrb,DE-cad::GFP/SCAR*$^{Δ37}$,*DE-cad::GFP;crb*$^{GX24}$ (top) and *w; foscrb*$_{Y10A}$,*DE-cad::GFP/SCAR*$^{Δ37}$,*DE-cad::GFP;crb*$^{GX24}$ (bottom). Time-lapse: 10 min; 8 fps.

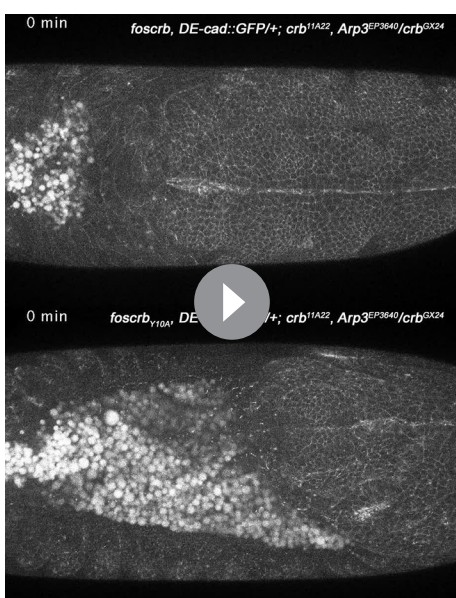

**Video 14.** Dorsal views during DC in embryos heterozygous for the *Arp3*$^{EP3640}$ allele. *w;foscrb,DE-cad::GFP/+;crb*$^{11A22}$,*Arp3*$^{EP3640}$/crb$^{GX24}$ (top) and *w; foscrb*$_{Y10A}$,*DE-cad::GFP/+;crb*$^{11A22}$,*Arp3*$^{EP3640}$/crb$^{GX24}$ (bottom). Time-lapse: 10 min; 8 fps.

the polarity determinant Crb affects major steps during DC, namely elongation of the DME cells, proper formation of the actomyosin cable at the LE, and regulated constriction of the AS cells. In addition, impaired *D*E-cad localisation suggest impaired adhesion. Overall, our results define a novel role of the FBM of Crb as an essential negative regulator of actomyosin dynamics in the AS during DC in *Drosophila*. This function is not allele-specific, since embryos carrying a *crb* allele, in which Y10, P12 and E16 in the FBM are replaced by alanines (*Huang et al., 2009*) develop a similar DC phenotype as *foscrb*$_{Y10A}$ embryos (data not shown). Genetic interaction studies revealed that this function of the FBM is mediated by *D*Moesin, members of the Rho family, the p21 activated kinase *D*Pak, and the SCAR-Arp2/3 complex (*Figure 6G*).

One phenotype observed upon complete loss of function of *crb* is a failure to maintain an intact ZA, a phenotype associated with the loss of polarity of many embryonic epithelia (*Tepass et al., 1990*; *Tepass and Knust, 1990*; *1993*; *Grawe et al., 1996*; *Tepass, 1996*). In fact, the AS is the tissue that is affected earliest (late stage 7/early stage 8) in *crb* mutant embryos (*Tepass, 1996*). However, *foscrb*$_{Y10A}$ embryos exhibit disrupted *D*E-cad staining in the AS only from stage 11 onward. Therefore, we suggest that the way how Crb controls maintenance of ZA integrity in the AS at later stages is different from its early function, which depends on a functional PBM (*Wodarz, et al., 1993*; *Klose et al., 2013*) and its interactions with the Par complex (*Morais-de-Sá et al., 2010*; *Harris and Peifer, 2005*). However, whether Crb, and in particular its FBM, regulates ZA integrity during DC by a different mechanism, or whether defects in the ZA are a secondary consequence of impaired actomyosin activity, remains to be determined.

Several of our results are compatible with the assumption that Crb regulates actomyosin dynamics, but since *foscrb*$_{Y10A}$ mutant embryos show defects both in the AS and the DME cells, we cannot distinguish in which of the tissues Crb activity is primarily required and whether defects observed in the DME of *foscrb*$_{Y10A}$ mutant embryos are secondary consequences of excessive contraction of the AS cells. Previous results clearly show that the activity of one tissue affects the behaviour of the respective other (*Kiehart et al., 2000*; *Hutson et al., 2003*; *Gorfinkiel et al., 2009*; *Solon et al., 2009*). For example, *zip* mutants have DC and head involution defects, and restoring *zip* function in either the dorsal epidermis or the AS is sufficient to rescue dorsal-open phenotypes (*Franke et al., 2005*). Similarly, expression of Pak-AID in the AS of *foscrb*$_{Y10A}$ mutants is sufficient to recover proper elongation of the DME (data not shown). However, the multitude of phenotypes observed in the DME cells of *foscrb*$_{Y10A}$ mutant embryos, such as persistence of Crb$_{Y10A}$, *D*Patj and Baz proteins and decrease of Ed expression at the LE, as well as disruption of the supracellular actomyosin cable and disorganised filopodia, suggest that Crb performs also specific functions in the DME. One possibility is that Crb influences actomyosin activity and filopodia formation in the DME cells by regulating the stability and localisation of Ena, the major regulator of protrusive activity at the LE (*Nowotarski et al., 2014*). Another possibility is that Crb regulates the LE actomyosin by modulating the localisation of Baz. In wild-type embryos, the removal of Baz from the LE (*Laplante and Nilson, 2011*) allows the relocation of the lipid phosphatase Pten, which, in turn, results in a localised accumulation of phosphatidylinositol-3,4,5-trisphosphate at the LE, promoting the formation of filopodia along the LE (*Pickering et al., 2013*).

## Crb regulates actomyosin dynamics

The most prominent phenotype of *foscrb*$_{Y10A}$ embryos is the over-contraction of AS cells, most likely mediated by *D*Pak. In fact, cortical localisation of *D*Pak in the AS of *foscrb*$_{Y10A}$ embryos appears to be increased in some cells (data not shown). In addition, over-expression of Pak-AID in the AS of *foscrb*$_{Y10A}$ suppresses the GB retraction and DC phenotypes. A similar degree of suppression was observed upon over-expression of Flw, a negative regulator of Sqh. Members of the Rho GTPase family are well-established upstream regulators of actomyosin dynamics. Our data suggest that Rho1 plays a crucial role downstream of Crb, since heterozygosity of *Rho1*$^{1B}$ partially suppresses the DC phenotype of *foscrb*$_{Y10A}$ embryos. Previous data showed that over-expression of the constitutively active or dominant-negative form of Rac1 in the AS of wild-type embryos results in AS disruption (*Harden et al., 2002*). Our observation that the phenotype of *foscrb*$_{Y10A}$ embryos is enhanced upon expression of a dominant negative form of Rac1 in the AS of *foscrb*$_{Y10A}$ embryos suggests that Rac1 may act upstream of Crb or in a parallel pathway. Since the effects of dominant negative Cdc42$^{N17}$ could not be studied due to technical difficulties (see Materials and methods), we cannot

exclude any contribution of Cdc42 in this process. Therefore, our data so far support a role of Rho1 in the Crb-mediated control of actomyosin dynamics in the AS (*Figure 6G*).

The FERM protein *D*Moe is a likely candidate to link the FBM of Crb to Rho1 activity. *Dmoe* mutant imaginal epithelial cells lose epithelial markers and intercellular adhesion, become motile and show invasive behaviour (*Speck et al., 2003*). In addition, lack of *D*Moe activates the Rho1-Rok-myosin cascade and JNK-mediated apoptosis in imaginal discs (*Warner et al., 2010*; *Neisch et al., 2010*). In fact, the FBM of Crb can recruit Moe to the cell membrane, a process that fails upon replacement of Tyr10 or Arg7 by Ala in the FBM of Crb (*Neisch et al., 2010*; *Médina et al., 2002*). Similarly, mutating Tyr10 in the FBM of the intercellular adhesion molecule (ICAM)-2 or the equivalent Tyr residue in the FBM of the neural cell adhesion molecule L1 impairs interaction with the FERM proteins radixin and ezrin, respectively (*Hamada et al., 2003*; *Cheng et al., 2005*). Moreover, it has been shown recently that the FBM of Crb is necessary for organising *D*Moe, aPKC and the actin cytoskeleton at the marginal zone in the developing follicular epithelium (*Sherrard and Fehon, 2015*). And in cervical carcinoma cells, over-expression of the mammalian CRB3 protein restores an epithelial-like morphology by organising a cortical actomyosin network through the regulation of the p114RhoGEF-RhoA-ROCK1/2 pathway via the FERM protein Ehm2 (*Loie et al., 2015*). Finally, recent works documented direct binding between Moesin and Crb, which was abolished upon Y10A substitution (*Wei et al., 2015*).

It is unlikely that one of the other two established binding partners of the FBM of Crb, Ex and Yrt (*Ling et al., 2010*; *Robinson et al., 2010*; *Laprise et al., 2006*), mediates the Crb function in the AS. So far, no role of Ex during DC has been reported, and *ex* mutant embryos reach stage 16 of development without showing major morphogenetic defects (*Marcinkevicius and Zallen, 2013*). Yrt is expressed in the AS and the epidermis, but this is not affected in *foscrb*$_{Y10A}$ embryos. In addition, the DC phenotype of zygotic *yrt*$^{\Delta 75a}$ mutants is less severe than the one observed in *foscrb*$_{Y10A}$ embryos. Finally, we do not observe increased Crb protein levels in *foscrb*$_{Y10A}$ embryos, which would be expected if the interaction between Yrt and Crb is impaired (*Laprise et al., 2006*).

Further support for a more direct role of Crb in regulating the actomyosin network comes from the observation that Crb co-localises with *D*Par-6, aPKC and Baz at the medial actomyosin foci in the AS (*David et al., 2010*; *2013*). Given the known interactions between members of the Crb complex with members of the Par complex [reviewed in (*Bulgakova and Knust, 2009*; *Tepass, 2012*; *Rodriguez-Boulan and Macara, 2014*)], David et al. (*David et al., 2010*) suggest that Crb in apical medial foci provides an anchor for PAR proteins. They go on to show that Baz and Par6-aPKC have opposite effects on foci duration, in that Baz promotes and Par6-aPKC complex inhibits the duration of foci. The interplay between these polarity complexes and the actomyosin system seems to establish a delayed negative feedback that promotes the cyclic contractions in the AS (*David et al., 2010*; *2013*). In fact, Crb::GFP also exhibits a similar pulsation as Zip::GFP in the AS (own unpublished observations), so it will be important to analyse whether Crb$_{Y10A}$::GFP mutant proteins have different dynamics in comparison to the wild type Crb.

## Crb–a regulator of ZA integrity via actomyosin dynamics?

Given the observation that at early stages of embryonic development the PBM is required for ZA stability, and that the Crb$_{Y10A}$ mutant protein has an intact PBM, it is possible that during DC, Crb-mediated regulation of actomyosin dynamics impacts on ZA stability. Interestingly, *D*Pak is not only a regulator of actomyosin dynamics, but is also involved in supporting ZA stability, both in *Drosophila* and in mammalian cells (*Lozano et al., 2008*; *Braga et al., 2000*; *Akhtar and Hotchin, 2001*; *Pirraglia et al., 2010*; *Menzel et al., 2007*; *2008*). The role of *D*Pak itself in DC morphogenesis is still controversial. Previous work showed that cell shape changes in the AS occur normally in embryos lacking maternal and zygotic *Dpak* and that inhibition of *D*Pak in the AS does not prevent apical constriction of amnioserosa cells (*Conder et al., 2004*). However, wild-type embryos expressing Pak-AID in the AS show defects in head involution and DC, which are stronger than those of embryos devoid of maternal and zygotic *D*Pak. This led the authors to suggest that Pak-AID may also affect the activity of a second kinase, Pak3, in the AS (*Conder et al., 2004*). Thus, whether inhibition of *D*Pak, Pak3 or both upon expression of Pak-AID in *foscrb*$_{Y10A}$ embryos accounts for the rescuing effect of the DC phenotype, including rescue of the ZA, remains to be clarified.

How can *D*Pak regulate ZA integrity? ZA remodelling is essential for morphogenesis, and this remodelling is driven by the endocytosis and recycling of junctional components (*Harris, 2012*;

*Matsubayashi et al., 2015*). *D*Pak can activate the Arp2/3 complex directly or via the *Drosophila* WAVE homolog SCAR (*Lecuit et al., 2011*; *Kurisu and Takenawa, 2009*; *Zallen et al., 2002*). Arp2/3, in turn, has been implicated in the regulation of ZA stability, e.g. in the *Drosophila* notum, where it maintains ZA stability by regulating the endocytosis of junctional components (*Watanabe et al., 2009*; *Quiros and Nusrat, 2014*; *Lecuit et al., 2011*; *Georgiou et al., 2008*; *Leibfried et al., 2008*). Moreover, reducing the activity of the Arp2/3-complex suppresses the DC phenotype of $\alpha$-*Cat* mutants (*Sarpal et al., 2012*), and the Arp2/3–WAVE/SCAR complexes associate with E-cad clusters and regulate their endocytosis (*Verma et al., 2012*; *Kovacs et al., 2002*; *Lecuit and Yap, 2015*). In fact, *D*E-cad endocytosis is enhanced in a Rho1-dependent manner when junctions are under stress and *D*E-cad clusters are also down-regulated via inhibition of Par3 by Rok (*Levayer et al., 2011*; *Lecuit and Yap, 2015*). Our results are in agreement with a role of Arp2/3 in regulating ZA stability in the AS. Heterozygosity of $SCAR^{\Delta 37}$, $Arp1^{Q25st}$ or $Arp3^{EP3640}$ not only partially restored *D*E-cad::GFP localisation at the ZA in the AS of $foscrb_{Y10A}$ embryos and suppressed DC defects, but even rescued the lethality of $foscrb_{Y10A}$ flies. Fusion of abdominal segments in adult escapers suggest that Crb may also be involved in histoblast fusion during metamorphosis (*Madhavan and Madhavan, 1980*; *Ninov et al., 2007*). Myosin-II activity itself has also been shown to be essential for the maintenance of AJs in some cases. Mice ablated for NMHC II-A die by E7.5 due to massive defects in cell-cell contacts and epithelial multi-layering accompanied by loss of E-cad and $\beta$-catenin from adhesion sites (*Conti et al., 2004*). Similarly, ZA stability in the *Drosophila* embryonic ectoderm depends on myosin-II contractility and requires interactions with actin (*Engl et al., 2014*; *Truong Quang et al., 2013*). Finally, Rok and myosin-II activities participate in ZA remodelling in the *Drosophila* pupal eye by regulating the formation of *D*E-cad recycling endosomes (*Yashiro et al., 2014*). Because the SCAR-Arp2/3 complex is an important enhancer of actin protrusions (*Wood et al., 2002*; *Abreu-Blanco et al., 2012*; *Georgiou and Baum, 2010*), it is also plausible that reducing its activity in $foscrb_{Y10A}$ embryos stabilises the ZA indirectly.

On the other hand, misregulation of actomyosin activity is not always associated with defects in ZA stability and integrity of the AS. Expressing a constitutively active form of MLCK to increase myosin II activity or over-expression of RhoGEF2, an activator of Rho1, results in an increase in the number and density of actin foci without affecting the integrity of the AS (*Azevedo et al., 2011*; *Fischer et al., 2014*), which could be due to the use of a weak GAL4 driver. Alternatively, the difference to our results could be explained by the fact that these authors performed the over-expression in a background with more than two copies of E-cad (using a ubi-*D*E-cad::GFP line), while we performed the experiments in a knock-in *D*E-cad::GFP line (*Huang et al., 2009*; *2011*), which thus may represent a more sensitive background.

## Crb–an organiser of a platform to link the ZA with the actomyosin network?

Another possibility to interpret our results is that Crb, or an interacting protein, couples the actomyosin network and the ZA. During gastrulation in *C. elegans* a molecular clutch has been postulated to connect the myosin network with the adhesion sites to transmit the force generated by the actomyosin contractions (*Roh-Johnson et al., 2012*). In *Drosophila*, the actomyosin contractions in the AS are initially uncoupled from apical contractions and hence the ZA (*Solon et al., 2009*; *Gorfinkiel et al., 2009*; *Blanchard et al., 2010*). Successive rows of amnioserosa cells are then sequentially stabilised in a contracted state, driving further contraction of the tissue. The surface stabilization mechanism is not known, but is likely to involve an increase in cellular stiffness [reviewed in (*Paluch and Heisenberg, 2009*)]. In $foscrb_{Y10A}$ embryos the actomyosin foci in the AS emerge prematurely before the onset of germ band retraction, whereas in wild-type these foci are more abundant after the end of germ band retraction (*Figure 2—figure supplement 2* and data not shown). Thus, the early over-contraction of the actomyosin in $foscrb_{Y10A}$ embryos may induce a premature coupling to the ZA, thus disrupting germ band retraction and DC. An interesting candidate for this coupling is the protein Canoe, which binds to $\alpha$-catenin (*Sawyer et al., 2009*; *Pokutta et al., 2002*), and whose absence results in a DC phenotype (*Jürgens et al., 1984*; *Takahashi et al., 1998*; *Boettner et al., 2003*; *Choi et al., 2011*). Absence of Canoe induces the detachment of the actomyosin apparatus from cell-cell junctions during *Drosophila* mesoderm invagination (*Sawyer et al., 2009*; 2011).

**Table 2.** List of fly stocks used in this study.

| Fly stock | Description |
|---|---|
| $w$ | All stocks have the $w*$ or $w^{1118}$ background |
| $w;foscrb$ $w;foscrb_{Y10F}$ $w;foscrb_{Y10A}$ | Flies expressing fosmid variants of crb under the control of the endogenous promoter and inserted into the landing site attP40 on 2nd chromosome; described in (**Klose et al., 2013**) |
| $w;;crb^{11A22}/TTG$ | crb null allele; BSC 3448 |
| $w;;crb^{GX24}/TTG$ | crb null allele (**Huang et al., 2009**) |
| $w;yrt^{\Delta75a}crb^{11A22}/TTG$ | yrt protein null allele recombined with the $crb^{11A22}$ allele (**Laprise et al., 2006**) |
| $w;;puc^{E69}/TTG$ | lacZ enhancer trap in the puc locus, a read-out of JNK signalling (**Ring and Martinez Arias, 1993**; **Martín-Blanco et al., 1998**) |
| $w;SCAR^{\Delta37}/CTG$ | Loss of function allele (**Zallen et al., 2002**); BSC 8754 |
| $w;;Arp3^{EP3640}/TTG$ | generated by Berkeley Drosophila Genome Project (**Hudson and Cooley, 2002**); BSC 17149 |
| $w;ex^{697}/CTG$ | lacZ enhancer trap in the ex locus; kindly provided by Nick Tapon |
| $w;nub^{1}Arpc1^{Q25st}$ $FRT40A/CTG$ | Nonsense mutation at Gln25 (CAG→TAG); behaves as a null mutant (**Hudson and Cooley, 2002**); BSC 9135 |
| $w\ flw^{6}/FTG$ | Amorphic allele (**Raghavan et al., 2000**); BSC 23693 |
| $y\ w\ rok^{2}\ FRT19A/FTG$ | Encodes the first 21 amino acids of rok followed by a 35 aa random peptide and a stop codon (**Winter et al., 2001**); BSC 6666 |
| $w;Rho1^{1B}/CTG$ | Rho1 loss of function allele; BSC 9477 |
| $w;DE\text{-}cad::GFP$ | DE-cadherin fused with GFP knock-in allele; homozygous viable (**Huang et al., 2009**) |
| $w;DE\text{-}cad::mTomato$ | DE-cadherin fused with mTomato knock-in allele; homozygous viable (**Huang et al., 2009**) |
| $w;Zipper::GFP$ | Protein trap line: Zipper fused with GFP under endogenous promoter; homozygous viable; BSC 51564. |
| $w;sqh::Utrophin::GFP$ | Actin binding domain of human Utrophin fused with GFP under the control of the sqh promoter (**Rauzi et al., 2010**). |
| $w;;Sas::Venus$ | On 3rd; Stranded at Second fused with Venus under tubulin promoter (**Firmino et al., 2013**) |
| $w;\ GAL4^{332.2}$ | On 2nd; expresses GAL4 in amnioserosa; BSC 5398 |
| $w;\ UAS\text{-}Apoliner$ | On 2nd; engineered apoptotic reporter (**Bardet et al., 2008**); BSC 32122 |
| $w;\ UAS\text{-}flw\text{-}HA$ | On 2nd; HA-tagged flw protein under UAS control; BSC 23703 |
| $w;;\ UAS\text{-}Rho1^{N19}$ | On 3rd; dominant negative Rho1 under the control of UAS; BSC 7328 |
| $w;;\ UAS\text{-}Rac^{N17}$ | On 3rd; dominant negative Rac under the control of UAS; BSC 6292 |
| $w;\ UAS\text{-}Cdc47^{N17}$ | On 2nd; negative Cdc42 under the control of UAS; BSC 6288. The stock $w;DE\text{-}cad::GFP,UAS\text{-}Cdc42^{N17}/(CTG);crb^{11A22},UAS\text{-}Actin::RFP/TM6B\text{-}YFP$ or TTG was not possible to obtain, probably because the expression of $Cdc^{N17}$, induced by the GAL4 from the balancer chromosome is detrimental. |
| $w;;\ UAS\text{-}moe^{T559D}\text{-}myc$ | On 2nd; phosphomimetic Moesin under the control of UAS; BSC 8630 |
| $w;;\ UAS\text{-}moe\text{-}myc$ | On 3rd; myc-tagged Moesin under the control of UAS; BSC 52236 |
| $w;\ UAS\text{-}Pak\text{-}myr$ | On 2nd: constitutively-active, membrane-bound Pak under UAS control; BSC 8804 |
| $w;\ UAS\text{-}Pak\text{-}AID$ | On 2nd; Pak autoinhibitory domain under UAS control; kindly provided by Nicholas Harden (**Conder et al., 2004**) |
| $w;;\ UAS\text{-}Act::RFP$ | On 3rd; RFP-tagged Act5C under UAS control; BSC 24779 |
| $w;;\ UAS\text{-}rok\text{-}CAT\text{-}KG$ | On 3rd; a kinase-dead rok under UAS control; BSC 6671 |
| $FTG$ | Balancer on 1st FM7c, twi-GAL4 UAS-EGFP; from BSC 6873 |
| $CTG$ | Balancer on 2nd CyO, twi-GAL4 UAS-EGFP; from BSC 6662 |
| $TTG$ | Balancer on 3rd TM3, twi-GAL4 UAS-EGFP $Sb^{1}$ $Ser^{1}$; from BSC 6663 |
| $TM6B\text{-}YFP$ | Balancer on 3rd TM6B, Dfd-EYFP, $Sb^{1}$ $Tb^{1}$ $ca^{1}$; from BSC 8704 |

BSC - Bloomington stock center; DGRC - Drosophila Genetic Resource Center.

In conclusion, we show a novel function of the FBM of Crb as an essential regulator of cytoskeleton dynamics and tissue integrity during DC. Different lines of evidence show that Crb regulation of AS morphogenesis involves DMoesin, Rho-GTPases, class-I Pak, and the SCAR-Arp2/3 complex. Further work will determine at which level Crb regulates actomyosin dynamics and why it is just the

**Table 3.** Antibodies and probes employed.

| | Dilution | Source |
|---|---|---|
| Phalloidin Alexa Fluor 555 | 1:500 | Invitrogen |
| Alexa Fluor 488-, 568-, and 647-conjugated | 1:500 | Invitrogen |
| | | |
| *Rat antibodies* | | |
| anti-Crb2.8 | 1:500 | (*Richard et al., 2006*) |
| anti-*D*E-cadherin | 1:20 | DSHB DCAD2 |
| anti-Yurt | 1:100 | (*Laprise et al., 2006*) |
| | | |
| *Mouse antibodies* | | |
| anti-$\alpha$-Spectrin | 1:25 | DSHB 3A9 |
| anti-$\beta$-galactosidase | 1:200 | DSHB 40-1a |
| anti-Coracle | 1:25 | DSHB C566.9 |
| anti-Crb-Cq4 | 1:300 | DSHB Cq4 |
| anti-Disc large | 1:100 | DSHB 4F3 |
| anti-Enabled | 1:100 | DSHB 5G2 |
| anti-GFP | 1:500 | Roche 11814460001 (Mannheim, Germany) |
| anti-Hindsight | 1:100 | DSHB 1G9 |
| anti-Integrin $\beta_{PS}$ | 1:2 | DSHB CF.6G11 |
| anti-Phosphotyrosine | 1:100 | BD Transduction Laboratories cat. no. 610000 |
| anti-SCAR | 1:25 | DSHB P1C1 |
| | | |
| *Rabbit antibodies* | | |
| anti-Bazooka | 1:500 | kindly provided by A. Wodarz |
| anti-DAAM | 1:3000 | kindly provided by József Mihály (unpublished) |
| anti-Diaphanous | 1:5000 | kindly provided by Steven A. Wasserman (*Afshar et al., 2000*) |
| anti-*D*Patj | 1:1000 | (*Richard et al., 2006*) |
| anti-Echinoid | 1:5000 | kindly provided by Laura Nilson (*Laplante and Nilson, 2006*) |
| anti-Expanded | 1:300 | (*Boedigheimer and Laughon, 1993*) |
| anti-GFP | 1:500 | Invitrogen |
| anti-*D*Pak | 1:8000 | kindly provided by Nicholas Harden (*Harden et al., 1996*) |
| anti-Polychaetoid | 1:5000 | kindly provided by Sarah Bray (*Djiane et al., 2011*) |
| anti-Phospho-Moesin | 1:100 | Cell Signaling Technology 3150 (Danvers, Massachusetts, USA) |
| anti-Stranded at second | 1:500 | kindly provided by E. Organ and D. Cavener |

Invitrogen, Molecular Probes (Eugene, Oregon, USA); DSHB - Developmental Studies Hybridoma Bank (Iowa city, Iowa, USA)

morphogenesis of the AS that depends on the FBM of Crb, while all other embryonic epithelia are not affected.

## Materials and Methods

### Fly stocks (see *Table 2*)

Flies were maintained at 25°C on standard food. All the mutant alleles where balanced over fluorescent balancers to identify the homozygous mutants in fixed embryos or live imaging microscopy (see below). All crosses and analyses were carried in a *crb* null background (*crb*$^{GX24}$ or *crb*$^{11A22}$, homozygous or trans-heterozygous), so the expression of the different variants of Crb is exclusively provided

by the fosmid (*Klose et al., 2013*). The different UAS-lines where recombined with the *D*E-cad::GFP knock-in allele or the null *crb*[11A22] allele. The driver line GAL4[332.3] was recombined with each of the different fosmid alleles.

## Embryo collection and antibody staining

Embryo stage refers to the *foscrb;crb*[GX24] genotype morphology accordingly to (*Campos-Ortega and Hartenstein, 1985*). All genotypes (*foscrb;crb*[GX24], *foscrb*[Y10F];*crb*[GX24] and *foscrb*[Y10A]; *crb*[GX24]) were collected under the same conditions, at the same time and during the same period (indicated in the respective figure legend). In this way, the comparison between *foscrb* or *foscrb*[Y10F] and *foscrb*[Y10A] mutant phenotypes show the differences observed at a specific time after egg laying. Embryos were collected on apple juice plates at 25°C and then incubated for the appropriate times at 25°C or 28°C, dechorionated in 3% sodium hypochlorite for 3 min, fixed for 20 min in 4% formaldehyde in phosphate-buffered saline (PBS) solution/heptane V/V 1:1. Vitelline membrane was removed by strong shaking in heptane/methanol v/v 1:1, except for the staining of actin in which the vitelline membrane was removed by strong shaking in 80% ethanol. Embryos were blocked for 2 hr at room temperature in PBT (PBS + 0.1% Triton X-100) + 5% normal horse serum (Sigma-Aldrich H1270, St. Louis, Missouri, USA). Embryos were incubated for 2 hr at room temperature or overnight at 4°C with primary antibodies (see *Table 3*). For analysis of Zipper localisation, we used the protein trap line *Zipper::GFP* (see *Table 2*) and the staining was done using the anti-GFP antibody. Incubations with the appropriate secondary antibodies were performed for 1 hr at room temperature. Stained embryos were mounted in glycerin propyl gallate (75% glycerol, 50 mg/mL propyl gallate) and visualized using a Zeiss LSM 780 NLO confocal microscope (ZEISS Microscopy, Jena, Germany) with a C-Apochromat 40x/1.2W Corr objective with the correction collar at 0.18 (at this position the brightness and contrast was enhanced). To distinguish homozygous embryos, in all the stainings an anti-GFP antibody was included to stain for the balancer-provided GFP. All images for a given marker in different genotypes were taken under the same settings for laser power, PMT gain and offset. Maximal projections, merging and LUT-pseudocolor assignment was performed using Fiji (*Schindelin et al., 2012*). For the FIRE-LUT pseudocolor 0 is black and 255 is white. Mounting was done in Adobe Photoshop CC 2015.0.1 and when brightness and contrast was adjusted, the modifications were equally applied to all the set of images for a given marker.

## Cuticle preparation

Embryos were collected overnight on apple juice plates at 25°C and then incubated for > 6 hr at 28°C. All the GFP or YFP positive eggs (the GFP or YFP is provided by the balancer) were removed and the remaining eggs where maintained at 25°C. The next day, the plates were screened again to remove remaining GFP/YFP positive eggs/larvae. Thus, all the remaining eggs/larvae had a *crb* null background (*crb*[GX24] or *crb*[11A22], homozygous or trans-heterozygous). These eggs/larvae were collected, dechorionated in 3% sodium hypochlorite for 3 min, mounted on Hoyer's medium (gum arabic 30 g, chloral hydrate 200 g, glycerol 20 g, $H_2O$ 50 ml), and the slide was incubated overnight at 60°C. In this way, all the eggs laid in the plate were at least >28 hr at 25°, enough time to let the larvae hatch when they are viable. The preparations were analysed by phase contrast with a Zeiss Axio Imager.Z1 microscope with an EC Plan-NEOFLUAR 10X/0.3 objective.

## Scanning electron microscopy (SEM)

Embryos were collected on apple juice plates for 1 hr at 25°C and then incubated for 8 hr at 28°C, dechorionated in 3% sodium hypochlorite for 2 min 30 sec, and fixed for 30 min in 25% glutaraldehyde/heptane v/v 1:1. Devitellinization was done by hand in 25% glutaraldehyde. Then, the embryos were postfixed in modified Karnovsky (2% paraformaldehyde/2% glutaraldehyde in 50 mM HEPES) followed by 1% osmium tetroxide in PBS, dehydrated in a graded series of ethanol, transferred to microporous capsules (78 μm pore size, Plano Cat. 4614) and critical point dried using the Leica CPD 300 (Leica Microsystems GmbH, Wetzlar, Germany). Embryos were mounted on 12 mm aluminium stubs and sputter coated with gold using a Leica Baltec SCD 050. Samples were analysed with a Jeol JSM 7500F cold field emission SEM (JEOL Ltd, Tokyo, Japan) at 10 kV acceleration voltage.

## Live imaging

Embryos were collected and incubated as describe above (see 'Embryo collection and antibody staining'). In the analysis of pulsed contractions in the AS, sequential collections of 30 min interspaced by 1 hr between each genotype allowed us to analyse 2–3 embryos of each genotype on the same session, so the acquisition conditions for all the genotypes were identical. To eliminate $crb^{GX24}$ or $crb^{11A22}$ heterozygous embryos, all GFP or YFP positive embryos were removed. The remaining eggs were dechorionated by hand or in 3% sodium hypochlorite for 2 min, mounted and oriented in a bottom glass Petri dish (MatTek P35G-1.5.14-C, Ashland, Massachusetts, USA). Previously, the glass was cover with a thin layer of glue (adhesive dissolved from double sided tape in heptane). The embryos were covered with water and visualized by multi-position scanning using a Zeiss LSM 780 NLO confocal microscope with a W Plan-Apochromat 40x/1.0 objective. Excitation was performed with 488 nm for GFP or YFP, and 561 nm for RFP or mTomato from an Argon Multiline Laser. The pinhole was adjusted for faster acquisition, so the step sizes correspond to 2.01 μm (*Videos 1*, *2*, *7*, *11*, *13*, *14*), 2.3 μm (*Videos 4*, *5*, *6*), 1.2 μm (*Video 8*, *12*), 1.46 μm (*Videos 3* and *9*). 4D-Hyperstacks were processed with Fiji (*Schindelin et al., 2012*) and the movies were rendered with Adobe Photoshop CC 2015.0.1. Under these conditions we observed that *w;foscrb,DE-cad::GFP;crb^{GX24}* embryos imaged for >7 hr at 5 min time lapse hatched and survived without showing any obvious damage (data not shown).

## Statistical analyses

Statistical analyses were performed with GraphPad Prism 6. Results are expressed as means ± SD. Statistical significance was evaluated in a one-way analysis of variance (ANOVA) followed by a Dunnett's multiple-comparison test. In the analysis of the statistical significance of the data presented in the *Figure 7—figure supplement 1*, the percentages were first converted to arcsin values and then analysed by a one-way-ANOVA followed by a Dunnet's multiple comparisons test.

# Acknowledgements

Stocks obtained from the Bloomington Drosophila Stock Center (NIH P40OD018537) were used in this study. Antibodies obtained from the Developmental Studies Hybridoma Bank, created by the NICHD of the NIH and maintained at The University of Iowa, Department of Biology, Iowa City, IA 52242. We thank Thomas Kurth (Biotec Dresden) for the scanning electron microscopy analysis and the LMF facility of MPI-CBG for microscopy support. We thank Sarita Hebbar and Johanna Lattner for critical reading of the manuscript, and Perla Zerial for her advice on the statistical analysis. The work was supported by funding from the Max-Planck Society.

# Additional information

### Funding

| Funder | Grant reference number | Author |
| --- | --- | --- |
| Max-Planck-Gesellschaft | | Elisabeth Knust |
| Max-Planck-Gesellschaft | Postdoc Stipend | David Flores-Benitez |

The funders had no role in study design, data collection and interpretation, or the decision to submit the work for publication.

### Author contributions

DF-B, Conception and design, Acquisition of data, Analysis and interpretation of data, Drafting or revising the article; EK, Conception and design, Analysis and interpretation of data, Drafting or revising the article

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
