## [Decision Letter]

Thank you for submitting your work entitled "Crumbs is a negative regulator of amnioserosa contractions during dorsal closure in *Drosophila*" for peer review at *eLife*. Your submission has been favorably evaluated by K VijayRaghavan (Senior editor, who also served as Reviewing editor and reviewer) and three other reviewers.

The reviewers have discussed the reviews with one another and the Reviewing editor has drafted this decision to help you prepare a revised submission.

Summary of paper:

In this manuscript, Flores-Benitez and Knust examine the specific phenotypes associated with a crumbs allele, *foscrb_Y10A_*, which leads to aberrant dorsal closure. The authors examine the role of this allele in relation to the major known mechanical and signaling pathways for dorsal closure: the DME supracellular actin cable, DME leading edge protrusions, contractions of the amnioserosa, and the JNK pathway. The authors find that JNK signaling is essentially intact, while the DME supracellular actin cable, and DME protrusions are disrupted, with mostly expected changes to proteins known to mediate these structures. In regards to the amnioserosa itself, the authors find that this structure has disrupted zonula adherens and, through a series of rescue experiments, conclude that the major defect is due to excessive Rho-mediated amnioserosa contractions.

Overview of referee comments:

The reviewers agree that that this is potentially good work and in revised form could be of interest to a large audience. For a substantially revised manuscript to make a valuable contribution to *eLife* there are several major issues, including important controls that need to be addressed.

The authors need to improve the delivery if they want this paper to have the impact they want it to have. As presented, it is difficult to read and experimentally incomplete. Part of the problem is that paragraphs are not very well structured and topic sentences don't adequately describe the topics addressed in the paragraph. Thus the reader is expecting to learn about one thing but data and details are presented regarding many others. Part of the problem may also be that the authors have presented a lot of material (too much material?) and as a consequence, the presentation is not as clear or complete as it might be. There are in addition some technical problems with the data as presented. The authors should focus on presenting their findings more clearly and more completely, potentially removing some findings in order to better document and provide appropriate controls on other findings.

The domain-specific phenotypes are the major thrust of the paper. Therefore the authors must with clarity establish that the phenotypes are due to the mutant domain (the authors still need to verify that the protein is properly folded and trafficked to conclude that the phenotypes are domain- and not simply crb- specific). Better showing the localization of the mutant protein at higher resolution and in all tissues would be valuable. Presumably, in tissues that are unaffected, it should look precisely like the wild type or the Y10F variant.

Having done this, the authors need to carefully discuss direct- and indirect- phenotypes, making sure that they are not over-interpreted. The rescue experiments with Rho and Flw suggest that there could be a more direct effect on actomyosin activity on top of the adhesion defects. In this context it is important that the major conclusion that Crumbs is negatively regulating Rho1 should be tested directly (see specific comments below). The "meat" of the manuscript lies on the set of rescue experiments of dorsal closure, and particularly of amnioserosa morphogenesis. However, the authors make the analysis of the defects at the level of dorsal epidermal cells an important part of the work which then seems to be set aside. The inclusion of the section describing AS cells becoming migratory is not salient to the authors' major points, but is a bold statement with the data presented. They should remove this section from the paper or provide additional control experiments to show these cells are not escaping hemocytes.

Specific Comments:

Reviewer 1:

This paper provides convincing phenotypic evidence for the crumbs allele *foscrb_Y10A_*causing defects to the DME supracellular actin cable, DME protrusions, and amnioserosa over contractions. However, a major concern is that there is little evidence that all of these strikingly documented phenotypes are not secondary to the adhesion defects found here and expected in a crumbs allele. As such, some of the conclusions are overstatements of the data presented.

1) While the images and movies provided are often striking, quantification would be helpful especially in DME phenotypes. For instance, in Figure 20 the authors highlight cells, which masks the ability to judge expression independently; a line profile across these cells and adjacent cells would helpful. Similarly, the differences in cadherin clustering in Figure 3 are hard to gauge in the picture shown and would benefit from quantification.

2) The conclusion that Yurt is uninvolved because its phenotype is qualitatively less severe could be explained by it only being involved in a subset of activities. Combining *foscrb_Y10A_* with a yurt allele could easily test this possibility.

3) The finding that amnioserosa cells are becoming motile is very striking. However, these cells resemble hemocytes, especially in the SEM pictures. If the amnioserosa has holes in it, as well as prematurely apoptotic cells, it would be expected for hemocytes to be responding to clear this cellular debris. The authors should confirm that the mobile cells are not hemocytes or tone down this section.

4) The major conclusion that Crumbs is negatively regulating Rho1 should be tested directly, especially considering some of the surprising results in the rescue experiments where Rok, loss of function Rho1 and Rho1 dominant negative alleles do not give similar results. Is Rho1 or active Rho1 increased in the *foscrb_Y10A_* background?

*Reviewer 2:*

This is a compelling, interesting and rigorously crafted work that reports novel and important information that will be of interest to a large audience. I enjoyed very much reading this work which opens up several research avenues to unravel the molecular mechanisms underlying amnioserosa cell apical contraction. I only have a couple of questions:

1) The authors say that Crumbs is a negative regulator of actomyosin dynamics and report an increase in the number of myosin (Zipper) foci, which persist longer and are more disorganized. I wonder if actin localization also follows this aberrant dynamics. It is very likely that it does so, but it may be good to check.

2) Regarding the defects of epidermal cells that are in contact with the amnioserosa and the disorganization of the actin cable and filopodia in *foscrb_Y10A_* embryos, the authors seem to favor the idea that they are a secondary effect due to defects in the contraction of the amnioserosa. Although I agree that some of the observed defects can be attributed to a failure in amnioserosa morphogenesis (i.e. *D*E-Cadherin localization), I think that some of the defects could be due to a specific function of Crumbs in these cells. In this sense, I wonder whether the authors have observed any differences in the organization of the dorsal epidermis in the different rescue experiments performed, i.e. is the actin cable, filopodia formation or Ena localization differentially restored when Flw is ectopically expressed in the AS or when one copy of *SCAR, Arp3* or *Arpc1* is removed from fo *foscrb_Y10A_* embryos?

*Reviewer 3:*

A substantially revised manuscript with some additional controls may well make an excellent contribution to *eLife*.

1) A previous finding was that a fosmid covering the entire crumbs (*crb*) locus, *foscrb* or a variant that replaces a conserved tyrosine with phenylalanine (*foscrb_Y10F_*), but not alanine (*foscrb_Y10AI_*) could completely rescue the embryonic lethality, germband retraction, dorsal closure and head involution defects that characterize homozygous, *crb* null animals. They evaluate *crb* function in detail by analyzing *foscrb* and variant rescue of *crb* null specimens using live imaging of a knockin allele of *D*E-cadherin, *D*E-cadGFP. The *foscrb_Y10A_* animals show major defects in germ band retraction, have a highly disorganized amnioserosa which is lost over time. Ultimately dorsal closure fails. The authors do not make clear why they have chosen to analyze two variants, Y10F, which in all cases mimics the wild type Y10 variant and Y10A, which in all cases behaves quite differently. They should explain why they are evaluating Y10F and might save considerable space by simply reporting that the Y10F variant behaves like the wild type Y10 in all cases (perhaps I missed where it did not?).

2) Next, they examine a role for Yrt, a candidate FERM domain protein that may act in concert with Crb. Normally, Crb, *D*Patj and Yrt are expressed at higher levels in the epidermis than the amnioserosa, but are largely absent from the leading edge of the dorsal most cells. In the Y10A animals, the DMEs fail to elongate properly and Crb and *D*Patj are associated with the leading edge, particularly in cells that remain short. In contrast, Yurt is absent from the leading edge in normal and Y10A animals. In addition, the authors find that the *yrt* mutant phenotype is different than the Y10A mutant phenotype, which they state starts earlier and is more complex than the *yrt* phenotype. They conclude the *yrt* seems not to be involved with the phenotype of Y10A embryos (this is not convincing, see specific comments below).

3) Next the authors demonstrate that a key feature of the Y10A phenotype is the absence of filopodia at the leading edge and conclude that *crb* function is required for normal microvilli.

4) The authors then turn their attention to how *crb* functions in the localization of echinoid and the formation of the actomyosin rich cable in the DME cells. They describe changes in echinoid distribution and report alterations in Baz distribution that are not particularly convincing. In addition they found that *yrt*, coracle and Dlg distributions were normal in all fosmid variants and conclude that *crb* is not responsible for specifying their localization.

5) Next, the authors evaluate *D*E-cadherin localization in the *foscrb* variants and report that in *foscrb* and Y10F *D*E-cad localizes at all cell-cell contacts, including at the LE, but fail to discuss the normal loss of *D*E-cad from the leading edge later in closure. Have they evaluated *D*E-cad distributions throughout closure (it changes dramatically)? The authors report that *D*E-cad at the LE and in the amnioserosa is reduced it the Y10A animals. Their conclusion that Y10A animals lack the actomyosin apparatus of the LE is certainly justifiable, but they should refine their conclusions about adhesion based on better characterization of the *D*E-cad distributions at early and late stages of closure.

6) The authors also evaluate the state of the amnioserosa in Y10A mutants and report that there are discontinuities in the distribution of *D*E-cad on AS borders and that there are large intracellular clusters of *D*E-cad in the Y10 animals (however, see comments below). Because germband retraction fails in these embryos, it is not clear that the defects reported are due to defects in dorsal closure per se and the authors should rethink their conclusions as a consequence.

7) Next the authors evaluate the integrity of the amnioserosa. They do not really explain why they used imaging of the microtubule binding protein Jupiter to do so (no comment on microtubule distribution or function is present, so the reader is left wondering). Nevertheless, they find that while extensions of the amnioserosa over the epidermis characterize all three variants of *foscrb, wt* and Y10F extensions are resolved during closure, but the in Y10A animals they are not. Again, a reasonable possibility seems to be that the amnioserosa is falling apart and because germ band retraction and dorsal closure have not progressed in Y10A animals that the AS cells take on the random migration phenotype. The SEMs presented confirm that aberrant nature of the apical surfaces of the AS cells that is consistent with their entering apoptosis. This may be because they are prematurely apoptotic or instead that they are undergoing apoptosis at an "appropriate" time, but germ band retraction and dorsal closure is delayed. Can the authors reference other developmental stages that confirm or refute this interpretation?

8) They next evaluate apoptosis and conclude that apoptosis is premature in Y10A embryos. For reasons detailed below, their data are not convincing on this point.

9) They also evaluate JNK signaling using *puc-LacZ* which is a biomarker for JNK signaling. They show convincingly that *puc-LacZ* localization is aberrant in Y10A embryos, most likely as a consequence of the abnormal elongation of DME cells. However, because there is the same number of *puc-lacZ* positive nuclei in the Y10A animals, the authors appear to conclude that JNK signaling is normal. Nevertheless, they never explicitly say that JNK signaling is normal in the Y10A embryos.

10) The authors go onto evaluate pulsed contractions of the amnioserosa cells. Videos and stills confirm that the individual cells in Y10A embryos are hard to discern. Nevertheless, there are pulsed contractions that are occurring in *D*E-cad labeled embryos. In addition, the zipGFP labeled embryos show clear, pulsed formation of myosin foci that subsequently disperse. The authors note that the period of the formation and dispersal of foci is very irregular in the Y10A embryos and the movies and kymographs are consistent with that observation. However, such movies and kymographs (of selected foci) do not give a sufficiently quantitative overview of the effect. The authors should quantify periods for randomly chosen foci in the wt, Y10F and Y10A cells. At least tens of foci should be analyzed, if not a hundred or so. Histograms of observed periods for foci formation and dissolution would be essential for drawing the conclusions that they subsequently draw as to the role of *crb* FBM in regulating actomyosin.

11) The authors show that over expression of Flapwing, a phosphatase for phospho-sqh, suppresses the Y10A failure in germ band retraction and dorsal closure. This is a very interesting finding and provides the most compelling evidence that *crb* mediates its effects through regulation of myosin contractility. Nevertheless, it is somewhat surprising that the authors did not evaluate better the effects of flapwing expression on the integrity of the amnioserosa cells (as assayed in *D*E-cad-GFP expressing embryos). Thus, Figure 5 panels M-O are at very low magnification and quality such that the integrity of the AS cells and the *D*E-cad-GFP adhesion junctions cannot be ascertained. The authors should present movies comparable to Video 10 with flapwing over expression and should also evaluate quantitatively the period of pulsed contractions in the flapwing over expressing embryos (for wt, Y10F and Y10A embryos).

12) To understand the mechanisms by which myosin is phosphorylated, the authors turn their attention to three effectors of small GTPases in the Rho/Rac/Cdc42 family, Rok, *D*Pak and Scar. They present data in a table that is part of Figure 6. It is not easy to follow is only poorly referenced in the text. First, there are no error bars on the numbers presented. 71.3% or 70.5% really different than one another (probably not)? Are they different from 59.7 or 56.5 (without error bars, not possible to know)? My interpretation of the first 3 rows of the table is that the expression of *UAS-Actin-RFP*, under the control of an amnioserosal driver (I shouldn't have to reference another table to figure out where Gal4-332.2 is expressed) substantially suppresses the most severe dorsal open phenotype (i.e., from 59.7 to 18.5), but the authors don't comment on that. Perhaps they want us only to focus on the penetrance of the wt-like phenotype? If so they should explicitly say so and they should consider pooling the defects into a single dorsal open category, perhaps relegating the more detailed panel to supplementary material). Moreover, where they do explicitly draw our attention to changes, the comparisons seem flawed. They ask us to compare row 4 to row 3 to show that *UAS-crb* expression increases the most severe dorsal hole, but the embryos in row 4 don't express *UAS-actin-RFP* and the expression of *UAS-actin-RFP* greatly affects the number of animals in the most severe category (compare row 3 to rows 1 and 2). Similarly, rows 9 and 10 are compared to 3, but 9 and 10 don't express *UAS-actin-RFP*. Similarly, are differences seen with *UAS-moe* (without actin) vs. UAS phosphomimetic moe (with actin) simply due to the expression of actin? If I understand Figure 6 correctly, experiments are not being compared to appropriate controls.

13) Nevertheless, some of the data show intriguing trends. Over expression of the autoinhibitory domain of *D*Pak suppresses the DC phenotype of Y10A animals, whereas over expression of *D*Pak induces a DC phenotype in otherwise normal embryos. Is it correct that autoinhibition of *D*Pak is intra-molecular? If so isn't it surprising that the autoinhibitory domain is effective when presented as a separate protein? How overexpressed is the domain?

Other key data are the suppression of Y10A DC phenotypes by the overexpression of *D*E-Cad.

14) Subsequent analysis of the role of Arp2/3 is focused on Arp2/3's role in endocytosis and is interpreted vis-à-vis the endocytosis of *D*E-cad. Lamelipodia and filopodia are a key feature of both the leading edge and amnioserosal tissues and Arp2/3 may have key roles in their function independent of their role in endocytosis. Why do the authors attribute the effects of Arp2/3 solely on endocytosis?

15) Finally, the authors conclude that *crb* function impacts both actomyosin dynamics and adhesion, and that the two effects may be inter-related, but the title of their paper suggests only a role in regulation of actomyosin dynamics. It is not clear why this is appropriate. While they use the Discussion to argue their point, it seems that a revised Results text, with some experimental revision may alter the conclusions made in the Discussion substantially. At this point their title should be revised to more accurately reflect the firm conclusions made in the Results and leave the speculation in the Discussion clearly marked as such.

In addition to these general comments, I make the following specific points for the Introduction and part of the Results. I incorporated comparable specific comments into the more general comments above for the remainder of the results but found that detailed comments were becoming both too numerous and too complicated.

In the subsection “The FBM of Crb is essential for dorsal closure”, first paragraph: The authors conclude the FERM binding domain of *crb* is responsible for the effects that they see because a point mutation (Y10A) in the FBD fails to rescue *crb* mutant function. Do they know that the alanine variant is made and correctly trafficked to the plasma membrane? If so, they should say how they know that and provide a reference. Otherwise conclusions regarding FBD function (vs. *crb* function as a whole) are erroneous.

Certainly dorsal closure fails in these Y10A animals (Figure 1), but germ band retraction is far from complete. Moreover, in normal embryos, the amnioserosa undergoes apoptosis upon completion of closure. Are the defects in closure observed due to the fact that at the posterior end the canthus does not form properly because of the incompletely retracted germ band and because the amnioserosa undergoes apoptosis thereby blocking closure? This puts a different spin on the defects observed.

In the subsection “The FBM of Crb is essential for dorsal closure”, the authors state that "Of these protein species, Yrt appeared as a likely candidate responsible for the defects in the *foscrb_Y10A_* mutant, since Yrt negatively regulates Crb (Laprise et al., 2006), zygotic *yrt* mutant embryos show DC defects (Hoover and Bryant, 2002), and over-expression of Crb in the amnioserosa results in DC defects (Harden et al., 2002; Wodarz et al., 1995)”. I don't understand how "over-expression of Crb in the amnioserosa results in DC defects" makes Yrt a likely candidate? The manuscript is characterized by a number of statements such as this one that distracts the reader and detracts from the message that the authors are trying to deliver.

In the subsection “The FBM of Crb is essential for dorsal closure”, fifth paragraph: The authors should show on the schematic in Figure 1 the orientation of the cross sections shown in Figure 1 panels M, N and O. Neither the text, nor the figure legend adequately describes the orientation of the section, and how it was obtained. Thus it is hard for the reader to understand what these panels are trying to show.

In the subsection “The FBM of Crb is essential for dorsal closure”, the authors state: "Moreover, in *foscrb* and *foscrb_Y10F_* lacking zygotic *yrt* DC fails only after GB retraction, which is qualitatively different from the phenotype of *foscrb_Y10A_* or *foscrb_Y10A_* embryos lacking zygotic *yrt* (Video 4)". The authors should state explicitly how the phenotypes are qualitatively different and explicitly state whether or not the Y10A phenotype differs in the presence or absence of zygotic *yrt*. In addition the next sentence "These results suggest that the DC phenotype of *foscrb_Y10A_* embryos starts earlier and is more complex than that in *yrt* mutants" requires the authors to give us more details about the *yrt* phenotypes, which are only referenced and not adequately described. I found this paragraph very hard to get through. The authors should consider reworking it for clarity.

In the subsection “The FBM of Crb regulates filopodia formation and organisation of the supracellular actomyosin cable“, first paragraph: The authors should provide a reference or other explanation of why Sas stains filopodia. Also, given that the core of microvilli consists of actin filaments, I don't understand why microvilli are not visible in the *foscrb* and Y10F animals (Figure 2). (Under appropriate conditions, ena might also be visible in the microvilli, but previous reports suggest it is visible primarily if ena is over expressed).

In the subsection “The FBM of Crb regulates filopodia formation and organisation of the supracellular actomyosin cable”, the authors state: "The formation of the actomyosin cable at the LE depends on the asymmetric accumulation of the adhesion protein Echinoid (Ed) in the DME cells (Laplante and Nilson, 2011; Lin et al., 2007)”. As written only readers familiar with the referenced work will realize that the asymmetry of Ed protein in the DME cells is not normal and that it is asymmetry in the distribution of Ed between the epidermis and the amnioserosa that is relevant. They also state that "Removal of Ed from the LE is evident in *foscrb* and *foscrb_Y10F_* embryos (Figure 2, arrowheads mark Ed absence at the LE)", but Ed is also not present at the leading edge of the DME cells in Y10A animals. Moreover, the data presented in Figure 2 is not at all convincing. Baz is present at the leading edge of only one Y10A cell and it is possible that the dark line shown is background, like the line just above the arrow, deeper in the amnioserosa.

In the subsection “The FBM of Crb regulates filopodia formation and organisation of the supracellular actomyosin cable“, sixth paragraph: Gorfinkiel and Arias 2007 show very convincingly that *D*E-cad initially localizes to the LE, but later during closure is lost from the leading edge (see Gorfinkiel and Arias Figure 1). The authors need to verify that their observations on the fosmid rescue embryos are consistent with the literature and consider how clearing the leading edge of *D*E-cad is consistent with the mechanisms they propose here. Moreover, the quality of the *D*E-cad stained images lacks contrast (compare Gorfinkial and Arias, Figure Aiii and Biii to this manuscripts Figure 2). Better stained preparations might give better insight into *D*E-cad distributions in the Y10A animals.

In the subsection “The FBM of Crb is essential for the morphogenesis of the AS”, second paragraph: The authors claim that there are discontinuities in *D*E-cad localization in Y10A animals which appears to be the case, although if the structure of the AS is considerably different than wildtype. How do the authors know that the *D*E-cad junctions haven't simply gone out of the plane of section? Moreover, the authors cite large aggregates of *D*E-Cad-GFP in the cytoplasm of the Y10A cells, but there seem to be a few such aggregates even in *foscrb* animals and large numbers of such aggregates in the Y10F animals (i.e., numbers comparable to those seen in the Y10A cells, see 3A' and 3B', respectively).

In the subsection “The FBM of Crb is essential for the morphogenesis of the AS”, fifth paragraph, the authors state: "Apoliner expression in the AS of *foscrb* and *foscrb_Y10F_* embryos (Video 9) revealed some apoptotic cells at the posterior canthus at the end of GB retraction (Figure 4, arrows). In *foscrb_Y10A_* embryos of a similar age, some detached cells (Figure 4, arrowheads) as well as those at the posterior edge of the remaining AS (Figure 4, arrow) appear as apoptotic cells."

First, the authors miss a key paper on apoptosis in the amnioserosa (Shen et al. 2013 PLoS ONE 8:e60180), which presents data in form that is much easier to interpret (see Shen et al. Figure 5). To me, the data presented here suggest that despite substantial differences in AS morphology, that the level of apoptosis in Y10A animals is about the same as in *foscrb* and Y10F embryos (there may be some additional apoptosis at lesions in the AS). The authors go on to write "At the end of DC, the internalised AS cells are localised in a central rod-like structure in *foscrb* and foscrbY10F embryos and subsequently die by apoptosis (Figure 4) [as has been reported for wild type embryos (Reed et al., 2004)], while in *foscrb_Y10A_* embryos of similar age, the remaining AS cells have completely disaggregated (Figure 4)." What do they authors mean by "a similar age"? To me this is completely consistent with the interpretation that morphogenesis is delayed and the programed cell death of the AS occurs "normally" in the animals with all three *foscrb* variants. The authors go on to state "To summarise, the AS in *foscrb_Y10A_* embryos breaks apart and undergoes premature apoptosis". I agree that the AS breaks apart, but it is not clear from the data as described that the apoptosis is "premature".

They next write "suggesting that an intact FBM is required for maintaining the elasticity of the AS". There is nothing in this paper that evaluates the "elasticity" of the AS. Again, they need to know that *crb* is folding properly to draw conclusions about a defective FBM (vs. a key role for crb) and they can at best replace "elasticity" with "integrity".

*Reviewer 4:*

We know from previous work that the Crumbs FERM domain-binding motif (FBM) directly interacts with the FERM-domain of Yurt (Yrt) and Expanded (Ex) and recruits Moesin to the apical membrane. And we know that mutation of this domain has no effect on the many other tissues where Crumbs has a role, but has lethal phenotype in the embryo, which this study shows to be due to an overactive actomyosin network. The live-imaging of Cadherins using *D*E-cad::GFP knock-in allele in a *crb* null background with formed introduced variants of crumbs is well-executed, systematic and key part of the study from which the conclusions are derived. The authors cleanly demonstrate a Yrt independent role and then show that *foscrb_Y10A_* show defects in the establishment of the actomyosin apparatus cells in a situation where components of the Zonula Adherens are disturb or missing. Next the authors go on to show that the FBM of Crb is an important regulator of cytoskeletal activity and cell-cell adhesion of the amnioserosa and required for its proper morphogenesis. The next set of experiments are the 'meat' of the paper and they comprehensively address the role of Crumbs FBM in actomyosin dynamics and show that the FBM of Crb regulates actomyosin dynamics in the amnioserosa during dorsal closure by down-regulating the Rho1 pathway. Finally the authors show that the dorsal-closure phenotype of CrbY_10A_ embryos also due to reduced adhesion. The authors have carefully dissected out Crumb function through the FBM. As they point out we do not yet know if the actomyosin phenotype and the reduced adhesion ones are linked or not. The phenotypes could result from defects in amnioserosa differentiation or in its attachment to the basal lamina. The authors raise this point themselves and have 'data not shown' to negate such conclusions. It will be good to see this data. While the work convincingly demonstrates a novel function of the FBM in cytoskeleton dynamics and also show that Crb regulation of amnioserosa morphogenesis involves *D*Moesin, Rho- GTPases, class-I Pak, and the SCAR-Arp2/3 complex, two questions remain. How does this regulation take place (three ways are suggested) and why other epithelia which require Crumbs does not seem to require this domain (while requiring the downstream partners). The key question then is whether the role of the FBM show is a peculiarity that details out an aspect of Crumbs or if it is telling us something broader about its role in epithelial morphogenesis. It will be good to see a discussion on this.

[Editors' note: further revisions were requested prior to acceptance, as described below.]

Thank you for resubmitting your work entitled "Crumbs is an essential regulator of cytoskeletal dynamics and cell-cell adhesion during dorsal closure in *Drosophila*" for further consideration at *eLife*. Your revised article has been favorably evaluated by K VijayRaghavan (Senior editor, who also served as Reviewing editor) and three reviewers. The manuscript has been improved but there are some remaining issues that need to be addressed before acceptance, as outlined below:

The reviewer reports are appended below. The authors will see that all the issues, but one, are readily addressed by editing the text. Reviewer 3's first comment is an important one, which is speedily addressable experimentally. *eLife* usually does not ask for multiple rounds of experiments but we were not convinced by the rebuttal in this case, and need to be assured with a micrograph of quality that the interpretation presented is valid.

*Reviewer 1::*

In their revised manuscript, the authors have included new data and analyses, and have clarified the text to adequately address most of the considerable reviewers' comments. While some of the mechanistic questions raised are still unanswered, in particular in the sections relating to the role of Rho GTPases, the authors did try to address these with the available reagents. The overall flow of the paper is much improved.

*Reviewer 2:*

I think this new version of the manuscript has very much improved. The text and figures have been simplified which benefits the delivery of the main message. I also think the Discussion better integrates different interpretations of the cause of the phenotype, both at the level of the tissue contributions (dorsal epidermis and AS) and of the cellular processes involved (actomyosin contractility and cell-cell adhesion).

More specific issues:

I particularly like Figure 7—figure supplement 1 more than new Figure 7 find the genotypes are easier to follow and think the subtleties of the DC defects are important. The addition of error bars is a good idea but in this case I would also add statistical significance of the differences observed.

I suggest Figure 5–figure supplement 2 goes to the main text: it's a very important piece of data and captures very nicely the behaviour of actomyosin foci.

*Reviewer 3:*

This is a vastly improved manuscript that is now nearly ready for publication in *eLife*.

I still have 6 concerns that should be corrected before publication.

The authors assert that Baz is retained at the leading edge of Y10A embryos. The micrograph they present in 2J is simply not compelling and the rebuttal they provide is simply not convincing. If Baz is preserved at the leading edge in Y10A embryos surely a micrograph that provides more compelling evidence should be easy to obtain? Compare their localization of Baz in wild type to that presented by Laplante and Nilson, 2011, which they cite.

Figure 2 show schematics to help the reader figure out what the main features of the Y10A phenotypes are. There is little or no evidence for the presence of a conventional myosin in filopodia. Interestingly, although the Sas data show filopodia, neither the actin nor the myosin panels show filopodia very well (there seem to be some filopodia in the actin panel E). The schematic that shows "actomyosin" should be removed. Actin might be shown in filopodia the actomyosin cable, myosin should be shown only in the actomyosin cable.

Gorfinkiel and Martinez-Arias (2007) showed that while Cadherin is initially localized to the leading edge of the DME cells, that it is cleared with time. In my first review, I was afraid that the phenotype shown for loss of cadherin in Y10A animals was due to the normal clearing of cadherin from the leading edge later in closure. The authors provide a convincing rebuttal that they are comparing early stage *foscrb* animals to early stage Y10A animals. They should simply add a sentence to the revised manuscript to indicate that they know that cadherin is cleared at later stages and that this comparison is between different genotypes during early closure stages. The readers need to know this as much as do the reviewers.

While pulsing or oscillating amnioserosa cells characterize dorsal closure in wild type embryos and are very interesting, it is not clear to me that oscillations "are required" for closure. Thus, oscillations can be suppressed and closure still occurs. The sentence that begins “Besides elongation of the DME cells and integrity of the AS, DC requires pulsed contractions of the AS cells…” should be revised.

Figure 6 is referenced as a model for how the FDB of Crb might regulate cytoskeletal function. I am puzzled by some of the details. In particular, the model has Rok and *D*Pak as positive regulators of MLCK. My understanding is that MLCK, Rok and *D*Pak all phosphorylate sqh directly, so arrows from Rok and Pak to MLCK and not to sqh are at best misleading and more likely simply wrong (unless there are data in the literature that I simply don't know about). Second, Rok has an inhibitory connection to flapwing. Is that really the case? Does Rok phosphorylate Flapwing to inhibit its phosphatase activity or otherwise bind to Flapwing to sequester its activity?

---

## [Author Response]

*Overview of referee comments: The reviewers agree that that this is potentially good work and in revised form could be of interest to a large audience. For a substantially revised manuscript to make a valuable contribution to* eLife *there are several major issues, including important controls that need to be addressed. The authors need to improve the delivery if they want this paper to have the impact they want it to have. As presented, it is difficult to read and experimentally incomplete. Part of the problem is that paragraphs are not very well structured and topic sentences don't adequately describe the topics addressed in the paragraph. Thus the reader is expecting to learn about one thing but data and details are presented regarding many others. Part of the problem may also be that the authors have presented a lot of material (too much material?) and as a consequence, the presentation is not as clear or complete as it might be. There are in addition some technical problems with the data as presented. The authors should focus on presenting their findings more clearly and more completely, potentially removing some findings in order to better document and provide appropriate controls on other findings.* We substantially modified the text and thus improved clarity. In addition, we removed some of the data (or present them as supplementary material). For example, the Y10F variant was used as a control to show that mutating the Y10 is not, by itself, destructive and that the defects in the Y10A variant are due to a conformational defect rather than a putative phosphorylation defect. Indeed, it was shown recently by Wei et al., (J Biol Chem 2015 290:11384) that the Y10A mutation in the FBM of Crb ablates the interaction with Moesin. We removed all results from the Y10F variant from the main figures, and added them as complementary when necessary. Also, we removed all the movies with the results of the Y10F variant. We modified the text accordingly, to improve the flow of the text.

*The domain-specific phenotypes are the major thrust of the paper. Therefore the authors must with clarity establish that the phenotypes are due to the mutant domain (the authors still need to verify that the protein is properly folded and trafficked to conclude that the phenotypes are domain- and not simply crb- specific). Better showing the localization of the mutant protein at higher resolution and in all tissues would be valuable. Presumably, in tissues that are unaffected, it should look precisely like the wild type or the Y10F variant.*

The localisation of the different *foscrb* variants in different tissues has been documented in our previous work Klose et al., 2013 (G3 2013, 3:153). Nonetheless, in the first paragraph of the Results section we stress that the fosCrb_Y10A_ protein localises apically in most embryonic epithelia, which also appear to have normal cell polarity (like the epidermis, for example). Besides this, in our current manuscript you can see the localisation of Crb and Crb_Y10A_ in the embryonic epidermis (previously Figure 1; now Figure 1). The localisation of all these variants in the amnioserosa is very difficult to obtain in fixed samples, because it appears to be highly dynamic according to our observations with a *fosCrb::EGPF* version (D.F.B and E.K. unpublished observations).

*Having done this, the authors need to carefully discuss direct- and indirect- phenotypes, making sure that they are not over-interpreted. The rescue experiments with Rho and Flw suggest that there could be a more direct effect on actomyosin activity on top of the adhesion defects. In this context it is important that the major conclusion that Crumbs is negatively regulating Rho1 should be tested directly (see specific comments below). The "meat" of the manuscript lies on the set of rescue experiments of dorsal closure, and particularly of amnioserosa morphogenesis. However, the authors make the analysis of the defects at the level of dorsal epidermal cells an important part of the work which then seems to be set aside. The inclusion of the section describing AS cells becoming migratory is not salient to the authors' major points, but is a bold statement with the data presented. They should remove this section from the paper or provide additional control experiments to show these cells are not escaping hemocytes.*

We re-wrote some parts of the Discussion and carefully took the possibility into account that some of the observed defects are an indirect consequence of the over- contraction of the AS in the *foscrb_Y10A_* embryos (second and third paragraphs). Regarding the direct measurement of Rho1, we explain the reasons why this experiment is not feasible (see below). About the AS cells becoming migratory, we moved the scanning EM data into the supplement (now Figure 3—figure supplement 1) and toned down the conclusion as recommended by the Reviewer 1 in point #3 (see below) (subsection “The FBM of Crb is essential for adhesion of the AS”, third paragraph).

Specific Comments:

Reviewer 1:

*This paper provides convincing phenotypic evidence for the crumbs allele* foscrb_Y10A_
*causing defects to the DME supracellular actin cable, DME protrusions, and amnioserosa over contractions. However, a major concern is that there is little evidence that all of these strikingly documented phenotypes are not secondary to the adhesion defects found here and expected in a crumbs allele. As such, some of the conclusions are overstatements of the data presented.*

We re-wrote several of the conclusions and we have tried to be more sober and moderate in the statements. Nevertheless, regarding the adhesion defects observed in the *foscrb_Y10A_* embryos, those are not similar to the phenotypes described for *crb* loss of function, because i) theY10A phenotype is very specific to the AS, and ii) it develops later, and iii) it depends on the FBM. Take, for example, the fact that, the expression of *UAS-Pak-AID* with the GAL4^334.3^ is enough to rescue the lethality of the *foscrb_Y10A_* embryos, leading to adults that do not show any major or obvious defects in other tissues (Figure 7). This supports the conclusion that the *foscrb_Y10A_* allele is not alike to the well-documented loss-of-function *crb* alleles characterised in this and other labs. Whether some of the documented phenotypes are or are not secondary to the adhesion defects, are now discussed and we agree that is a conundrum to be solved (subsection “The FBM of Crb controls actomyosin dynamics in the AS”, sixth paragraph).

*1) While the images and movies provided are often striking, quantification would be helpful especially in DME phenotypes. For instance, in Figure 20 the authors highlight cells, which masks the ability to judge expression independently; a line profile across these cells and adjacent cells would helpful.*

Done. We now show the image without the highlight (Figure 2) and complemented it with the line profile along the DME cells using the signal from phosphotyrosine staining as a reference for the ZA. In the Figure 2—figure supplement 3, we show that Bazooka intensity in the DME cells is strongly reduced in the *foscrb_Y10A_* embryos.

*Similarly, the differences in cadherin clustering in Figure 3 are hard to gauge in the picture shown and would benefit from quantification.*

We provide better quality images of the AS at stage 11 (Figure 3). In those images it is possible to see the disruption in *D*E-cad continuity at the ZA of *foscrb_Y10A_* embryos (Figure 3). Moreover, because we now employed a *D*E-cad::mTomato knock-in allele (viable knock-in allele – Huang et al,. 2009 PNAS 106:8284– like the *D*E-cad::GFP allele used in the rest of the work), the auto-fluorescence from the yolk is very low, improving the signal-to-noise ratio. However, the quantification of those defects appeared to be quite arbitrary. Of course it is possible to gauge the intensity of *D*E-cad along the ZA in both *foscrb* and *foscrb_Y10F_* embryos (for example with a line profile). On the contrary, in *foscrb_Y10A_* embryos, the positioning of the line (i.e. ROI) along the ZA in the AS becomes arbitrary, as the signal is not continuous or even lost, so we are not absolutely confident of drawing the ROI along the ZA. This is even more problematic at later stages, when the ZA becomes extremely disrupted (see current Figure 5).

*2) The conclusion that Yurt is uninvolved because its phenotype is qualitatively less severe could be explained by it only being involved in a subset of activities. Combining* foscrb_Y10A_
*with a yurt allele could easily test this possibility.*

The experiment and the video were already presented in the first version of the manuscript (Video 4), now (Video 7). We now also explain in more detail the Yrt phenotype and change the referred paragraph to make clear that the *yrt* phenotype is different from the *foscrb_Y10A_* mutant, and also that there is no enhancement or suppression of the *foscrb_Y10A_* phenotype when combined with the zygotic *yrt* mutant (subsection “The FBM of Crb is essential for the integrity of the AS”, third paragraph). The *foscrb_Y10F_* embryo was removed from the movie, because it is phenotypically identical to the *foscrb* when combined with the *yrt* mutant.

*3) The finding that amnioserosa cells are becoming motile is very striking. However, these cells resemble hemocytes, especially in the SEM pictures. If the amnioserosa has holes in it, as well as prematurely apoptotic cells, it would be expected for hemocytes to be responding to clear this cellular debris. The authors should confirm that the mobile cells are not hemocytes or tone down this section.*

Done, we toned down the conclusion and stated that we do not know the identity of these migratory cells, which can well be haemocytes escaping through the hole in the AS of the *foscrb_Y10A_* embryos (subsection “The FBM of Crb is essential for adhesion of the AS”, third paragraph).

*4) The major conclusion that Crumbs is negatively regulating Rho1 should be tested directly, especially considering some of the surprising results in the rescue experiments where Rok, loss of function Rho1 and Rho1 dominant negative alleles do not give similar results. Is Rho1 or active Rho1 increased in the* foscrb_Y10A_*background?*

Because this is one of the major points in the manuscript, we obviously wanted to know the precise change on Rho1 activity. Also, because the use of dominant negative or dominant active constructs can mask the effects of other small GTPases, we wanted to analyse the in vivo condition. Nevertheless, it is not possible to measure active Rho levels by conventional assays (like the Pull-down Activation Assay): the AS represents a very small proportion of the whole embryo, and even in the case that all the Rho protein is active (i.e. GTP loaded) in the AS of the *foscrb_Y10A_* embryos, we think that the other tissues of the embryo will mask the difference when processed for the assay.

However, we tried to approach this question by other strategies. First, we used the available lines that express Rho1::GFP under the control of the Rho1 promoter created in the lab of Susan Parkhurst, but none of them was strong enough and the signal-to-noise ratio was too low for analysis. We tried to generate recombinant stocks with the *Rho1^1b^* allele plus the Rho1::Rho1::GFP construct, plus the corresponding fosmid, plus the *crb* null allele (i.e. *w*; foscrb,Rho1^1b^,Rho1::Rho1::GFP/CyO, twi::GFP;crb^GX24^/TM3, twi::GFP*), but this approach was unsuccessful, as the intermediate stocks were very weak.

Another strategy consisted in the use of Rho family GTPase activity biosensors also generated by the Parkhurst lab (though they have not been characterised completely). Some of them are expressed constitutively and others are UAS constructs. We explored the use of different Rok^RBD^, Dia^RBD^, Capu^RBD^ and Pkn^RBD^ versions with different GAL4 drivers, too, and none of them gave a useful signal-to-noise ratio in our imaging settings. A new probe was reported by the Lecuit lab (Anillin::GFP, Munjal et al., (2015). Nature, DOI: 10.1038/nature14603), but the work was published on 27 July, so for obvious reasons we have not tested this probe.

Reviewer 2:

*This is a compelling, interesting and rigorously crafted work that reports novel and important information that will be of interest to a large audience. I enjoyed very much reading this work which opens up several research avenues to unravel the molecular mechanisms underlying amnioserosa cell apical contraction. I only have a couple of questions: 1) The authors say that Crumbs is a negative regulator of actomyosin dynamics and report an increase in the number of myosin (Zipper) foci, which persist longer and are more disorganized. I wonder if actin localization also follows this aberrant dynamics. It is very likely that it does so, but it may be good to check.*

We generated flies where the *D*E-cad::mTomato allele was recombined with the actin binding domain of Utrophin tagged with GFP and expressed under the control of the spaghetti squash promoter (sqh::Ut::GFP), and put in the *crb* null background. We have not analysed these embryos in detail. We provide a video (Author response video 1), where it is possible to appreciate the pulsed contractions in a *foscrb* embryo, and that the AS of the *foscrb_Y10A_* embryo is strongly disrupted (inside the dotted line in the merge of the *foscrb_Y10A_* embryo). The *D*E-cad::mTomato is disrupted in the “left over” of the AS, but nevertheless, it is possible to appreciate in the green channel clusters that resemble the ones observed with Zipper. Since this result is somewhat redundant, we mention it as “data not shown “in the text (subsection “The FBM of Crb controls actomyosin dynamics in the AS”, first paragraph).Author response Video 1.Dorsal closure (DC) in a *w;foscrb,DE-cad::GFP;crb^GX24^* embryo.Note that the granules from the yolk are visible because of their strong auto-fluorescence in the green part of the spectrum. Time-lapse: 3.5 min; 12 fps.**DOI:**
http://dx.doi.org/10.7554/eLife.07398.04410.7554/eLife.07398.044

*2) Regarding the defects of epidermal cells that are in contact with the amnioserosa and the disorganization of the actin cable and filopodia in foscrbY10A embryos, the authors seem to favor the idea that they are a secondary effect due to defects in the contraction of the amnioserosa. Although I agree that some of the observed defects can be attributed to a failure in amnioserosa morphogenesis (i.e.* D*E-Cadherin localization), I think that some of the defects could be due to a specific function of Crumbs in these cells. In this sense, I wonder whether the authors have observed any differences in the organization of the dorsal epidermis in the different rescue experiments performed, i.e. is the actin cable, filopodia formation or Ena localization differentially restored when Flw is ectopically expressed in the AS or when one copy of* SCAR, Arp3 *or* Arpc1 *is removed from foscrbY10A embryos?*

Indeed, we have not explored in detail the organisation of the DME cells in the different rescue experiments. We have made some observations that support the idea of a secondary effect: the elongation of the DME cells is restored in *foscrb_Y10A_* embryos expressing Pak-AID only in the AS. This was observed using *D*E-cad::GFP in a heterozygous combination, so the signal-to-noise ratio is not the optimal to show it. In Author response video 2, the upper panels show the max projections of the stacks obtained. As *D*E-cad::GFP is heterozygous, the contrast is very low. In the lower panels, part of the same stacks was used to remove the autofluorescence from the yolk and the vitelline in Fiji.Author response Video 2.Dorsal closure (DC) in a *w;foscrb,DE-cad::GFP;crb^GX24^* embryo.Note that the granules from the yolk are visible because of their strong auto-fluorescence in the green part of the spectrum. Time-lapse: 3.5 min; 12 fps.**DOI:**
http://dx.doi.org/10.7554/eLife.07398.04510.7554/eLife.07398.045

Thus, to obtain a reasonable number of observations and analyse these defects in detail, we need to generate other stocks in which the *DE-cad::mTomato* allele is recombined with the appropriate genotypes. However, we have not analysed the cytoskeleton dynamics of the DME cells in these conditions, so we cannot exclude the possibility that some of the defects in the DME cells of *foscrb_Y10A_* embryos are secondary to the defects in the AS. Accordingly, we discuss these aspects more carefully in the manuscript (Discussion, last paragraph).

On the other hand, in another set of unpublished experiments we used the engrailed-GAL4 or the patched-GAL4 drivers to express different constructs (*UAS-Ena::GFP, Uas-Pak-AID, UAS-Moesin, UAS-moesin^T559D^, UAS-flw*, etc.) in stripes on the epidermis of the *foscrb_Y10A_* embryos, but so far we did not find any suppression or rescue of the Y10A phenotype.

Reviewer 3:

*A substantially revised manuscript with some additional controls may well make an excellent contribution to* eLife. *1) A previous finding was that a fosmid covering the entire crumbs (*crb*) locus,* foscrb *or a variant that replaces a conserved tyrosine with phenylalanine (*foscrb_Y10F_*), but not alanine (*foscrb_Y10AI_*) could completely rescue the embryonic lethality, germband retraction, dorsal closure and head involution defects that characterize homozygous,* crb *null animals. They evaluate* crb *function in detail by analyzing* foscrb *and variant rescue of* crb *null specimens using live imaging of a knockin allele of* D*E-cadherin,* D*E-cadGFP. The* foscrb_Y10A_
*animals show major defects in germ band retraction, have a highly disorganized amnioserosa which is lost over time. Ultimately dorsal closure fails. The authors do not make clear why they have chosen to analyze two variants, Y10F, which in all cases mimics the wild type Y10 variant and Y10A, which in all cases behaves quite differently. They should explain why they are evaluating Y10F and might save considerable space by simply reporting that the Y10F variant behaves like the wild type Y10 in all cases (perhaps I missed where it did not?).*

Thanks for the suggestion. We now explain in the first paragraph of the Results that the *foscrb_Y10F_* allele is a control, in which embryonic development occurs as in *foscrb* or wild type embryos. We also explain that these results indicate that the phenotypes in *foscrb_Y10A_* embryos are due to the conformational distortion of the FERM-domain binding motif of Crb (as recently shown by Wei et al., 2015 J Biol Chem 290:11384), and not because of a putative interference with the phosphorylation of Crb in this tyrosine residue. Accordingly, we decided to remove all the results from the *foscrb_Y10F_* variant from the main figures and videos, and added them as figure supplements when we considered necessary, which not only saves space, but also improves the flow of reading.

*2) Next, they examine a role for Yrt, a candidate FERM domain protein that may act in concert with Crb. Normally, Crb,* D*Patj and Yrt are expressed at higher levels in the epidermis than the amnioserosa, but are largely absent from the leading edge of the dorsal most cells. In the Y10A animals, the DMEs fail to elongate properly and Crb and* D*Patj are associated with the leading edge, particularly in cells that remain short. In contrast, Yurt is absent from the leading edge in normal and Y10A animals. In addition, the authors find that the* yrt *mutant phenotype is different than the Y10A mutant phenotype, which they state starts earlier and is more complex than the* yrt *phenotype. They conclude the* yrt *seems not to be involved with the phenotype of Y10A embryos (this is not convincing, see specific comments below).*

We have addressed these issues (please see below).

*3) Next the authors demonstrate that a key feature of the Y10A phenotype is the absence of filopodia at the leading edge and conclude that* crb *function is required for normal microvilli. 4) The authors then turn their attention to how* crb *functions in the localization of echinoid and the formation of the actomyosin rich cable in the DME cells. They describe changes in echinoid distribution and report alterations in Baz distribution that are not particularly convincing. In addition they found that yrt, coracle and Dlg distributions were normal in all fosmid variants and conclude that* crb *is not responsible for specifying their localization.*

As Reviewer 1 also pointed out, the defects in Bazooka distribution in the DME cells of *foscrb_Y10A_* embryos were not clear, and according to his request, we now show the image without the highlight (Figure 2) and complemented it with the line profile along the cells using the signal from phosphotyrosine staining as a reference for the ZA. In the Figure 2—figure supplement 3, it is possible to see that the Bazooka intensity in the DME cells is strongly reduced in the *foscrb_Y10A_* embryos.

*5) Next, the authors evaluate* D*E-cadherin localization in the* foscrb *variants and report that in* foscrb *and Y10F* D*E-cad localizes at all cell-cell contacts, including at the LE, but fail to discuss the normal loss of* D*E-cad from the leading edge later in closure. Have they evaluated DE-cad distributions throughout closure (it changes dramatically)? The authors report that DE-cad at the LE and in the amnioserosa is reduced it the Y10A animals. Their conclusion that Y10A animals lack the actomyosin apparatus of the LE is certainly justifiable, but they should refine their conclusions about adhesion based on better characterization of the* D*E-cad distributions at early and late stages of closure.*

This question also refers to your specific comment below regarding the normal loss of *D*E-cad from the leading edge, we refer to Figure 1 from Gorfinkiel and Arias 2007 (J Cell Sci 120:3289), which shows a late stage of DC (during the zippering phase), which is, however, never reached by the *foscrb_Y10A_* embryos. Our characterisation focuses on earlier stages of embryonic development (i.e. form the start of germ band retraction), because these are the time points when the Y10A phenotype starts.

Anyway, in Figure 9 you can see that, at late closure, the loss of *D*E-cad from the LE indeed occurs in *foscrb* and *foscrb_Y10F_* embryos (see the arrowheads). These observations are consistent with our conclusions, as these alleles rescue the lack of endogenous *crb* completely, and behave, in all analysed aspects, as wild type.

Author response image 1.**DOI:**
http://dx.doi.org/10.7554/eLife.07398.046

Please note that we call the attention to defects in *D*E-cad localisation in the LE from the beginning of DC in *foscrb_Y10A_* embryos. Thus, we think that it is not necessary to include these observations into the manuscript as the redistribution of *D*E-cad at late stages does not change our interpretation about the defects in the AS of *foscrb_Y10A_* embryos.

*6) The authors also evaluate the state of the amnioserosa in Y10A mutants and report that there are discontinuities in the distribution of* D*E-cad on AS borders and that there are large intracellular clusters of* D*E-cad in the Y10 animals (however, see comments below). Because germband retraction fails in these embryos, it is not clear that the defects reported are due to defects in dorsal closure per se and the authors should rethink their conclusions as a consequence.*

This comment is about the same topic as comment #7, so please refer to the answer below.

*7) Next the authors evaluate the integrity of the amnioserosa. They do not really explain why they used imaging of the microtubule binding protein Jupiter to do so (no comment on microtubule distribution or function is present, so the reader is left wondering). Nevertheless, they find that while extensions of the amnioserosa over the epidermis characterize all three variants of* foscrb, wt *and Y10F extensions are resolved during closure, but the in Y10A animals they are not. Again, a reasonable possibility seems to be that the amnioserosa is falling apart and because germ band retraction and dorsal closure have not progressed in Y10A animals that the AS cells take on the random migration phenotype. The SEMs presented confirm that aberrant nature of the apical surfaces of the AS cells that is consistent with their entering apoptosis. This may be because they are prematurely apoptotic or instead that they are undergoing apoptosis at an "appropriate" time, but germ band retraction and dorsal closure is delayed. Can the authors reference other developmental stages that confirm or refute this interpretation?*

We employed Jupiter::GFP as a probe to follow the morphogenesis of the whole embryo, because this marker 1) is expressed in all cells, 2) it is a Protein Trap, so its 3) expression is under the control of the endogenous promoter, 4) it is homozygous viable, and 5) no phenotype has been reported for this allele. This offers an advantage over other available markers, like the actin-binding-domain of Moesin::GFP, which is expressed under the control of the *sqh* promoter or UAS, or LifeAct, which was not suitable in our experiments. For more clarity, we have removed the Jupiter::GFP experiment completely.

On the other hand, we also thought about the possibility of a “delayed” germ band retraction in *foscrb_Y10A_* embryos, but conclude that AS apoptosis occurring at the “appropriate” time is very unlikely. For example, expressing flapwing or Pak-AID in the AS is enough to suppress the germ band retraction phenotype, which suggests that this phenotype is secondary to the defects in the AS.

Unfortunately, we cannot use other morphological characteristics, such as the elongation of the hindgut, or the branching and fusion of the tracheal system, because these characteristics are disrupted in the *foscrb_Y10A_* embryos, probably due to defects in germ band retraction (this work and Klose et al., 2013 G3, 3:153).

We are aware of the difficulty in staging *foscrb_Y10A_* embryos (and other mutant genotypes) (Klose et al., 2013 G3, 3:153). Thus, as we did in the previous work and as it is now explained in the current Results (subsection “The FBM of Crb is essential for dorsal closure”) and Material and methods (subsection “Embryo collection and antibody staining”), all genotypes are collected *at the same moment* and collections are limited to a short period of time (specified in each figure or movie). Then, the embryos are incubated the appropriate time and fixed or imaged *at the same time* (even in live imaging, the three genotypes are *in the same chamber* and imaged using a multiposition setting), and the stage refers to the *foscrb* morphology based on Campos-Ortega and Hartenstein 1985 (The Embryonic Development of *Drosophila melanogaster*). Accordingly, when we describe any defect in the *foscrb_Y10A_* embryos, we always compare it to the *foscrb* embryos of the same age, meaning that both have been analysed at the same time after egg collection.

Consequently, for example, if we report a defect in germ band retraction (take for instance Figure 3—figure supplement 1), it is because at this time point, the *foscrb* and the *foscrb_Y10F_* embryos (the control embryos) already finished retraction.

To disentangle whether the defects in the germ band are a direct or indirect consequence of the DC defects is something beyond the aim of our present work. This specific topic has been analysed by others, so the roles of the AS (mechanical and/or biochemical) during germ band retraction are still under intense research: Yip et al., 1997 Development 124:2129; Lamka and Lipshitz, 1999 Dev Biol 214:102; Schöck and Perrimon, 2002 Dev Biol 248:29; Schöck and Perrimon, 2003 Genes Dev 17:597; Kozlova and Thummel, 2003 Science 301:1911; Reed et al., 2004 Curr Biol 14:372; Zúñiga et al., 2009 BMC Biol 7:61; Lynch et al., 2013 Dev Biol 384;205; Lynch et al., 2014 New J Phys 16:055003.

*8) They next evaluate apoptosis and conclude that apoptosis is premature in Y10A embryos. For reasons detailed below, their data are not convincing on this point.*

We guess you are referring to our statement "Apoliner expression in the AS […] appear as apoptotic cells" as “reasons detailed below”, so please read the following response.

*9) They also evaluate JNK signaling using* puc-LacZ *which is a biomarker for JNK signaling. They show convincingly that* puc-LacZ *localization is aberrant in Y10A embryos, most likely as a consequence of the abnormal elongation of DME cells. However, because there is the same number of* puc-lacZ *positive nuclei in the Y10A animals, the authors appear to conclude that JNK signaling is normal. Nevertheless, they never explicitly say that JNK signaling is normal in the Y10A embryos.*

Done; the conclusion has been included (subsection “The FBM of Crb is essential for the integrity of the AS”, end of fifth paragraph).

*10) The authors go onto evaluate pulsed contractions of the amnioserosa cells. Videos and stills confirm that the individual cells in Y10A embryos are hard to discern. Nevertheless, there are pulsed contractions that are occurring in* D*E-cad labeled embryos. In addition, the zipGFP labeled embryos show clear, pulsed formation of myosin foci that subsequently disperse. The authors note that the period of the formation and dispersal of foci is very irregular in the Y10A embryos and the movies and kymographs are consistent with that observation. However, such movies and kymographs (of selected foci) do not give a sufficiently quantitative overview of the effect. The authors should quantify periods for randomly chosen foci in the wt, Y10F and Y10A cells. At least tens of foci should be analyzed, if not a hundred or so. Histograms of observed periods for foci formation and dissolution would be essential for drawing the conclusions that they subsequently draw as to the role of* crb *FBM in regulating actomyosin.*

Done; the histograms confirm our observations (see Figure 5–figure supplement 3).

*11) The authors show that over expression of Flapwing, a phosphatase for phospho-sqh suppresses the Y10A failure in germ band retraction and dorsal closure. This is a very interesting finding and provides the most compelling evidence that* crb *mediates its effects through regulation of myosin contractility. Nevertheless, it is somewhat surprising that the authors did not evaluate better the effects of flapwing expression on the integrity of the amnioserosa cells (as assayed in* D*E-cad-GFP expressing embryos). Thus, Figure 5 panels M-O are at very low magnification and quality such that the integrity of the AS cells and the* D*E-cad-GFP adhesion junctions cannot be ascertained. The authors should present movies comparable to Video 10 with flapwing over expression and should also evaluate quantitatively the period of pulsed contractions in the flapwing over expressing embryos (for wt, Y10F and Y10A embryos).*

We could not evaluate the integrity of the ZA in vivo in experimental conditions where flapwing is expressed, because the *D*E-cad::GFP is heterozygous (recombined on the same chromosome as *UAS-flw*), so the signal-to-noise ratio is very low.

In the movie the reviewer is referring to (previous Video 10, now Video 8), the *D*E-cad::GFP allele was recombined with the corresponding *foscrb* variant, so it is homozygous and the signal to noise ratio is enough for a detailed observation of the *D*E-cad::GFP, despite the strong auto-fluorescence of the yolk. Hence, to make a proper analysis of the ZA in vivo when flw is expressed in the AS, the *D*E-cad::GFP allele should be recombined also on the chromosome carrying the corresponding *foscrb* variant and the GAL4^332.3^ (i.e. analysis of the embryo with the genotype *w*;foscrb,DE-cad::GFP,GAL4^332.2^/UAS-flw-HA,DE-cad::GFP;crb^GX24^/crb^11A22^,UAS-Actin::RFP*), which was not feasible. Another strategy will be to recombine the *D*E-cad::mTomato with the *UAS-flw*, but these stocks were not ready on time for these experiments.

Nevertheless, we prepared movies (now Video 12, subsection “The FBM of Crb controls actomyosin dynamics in the AS”) with better resolution of the AS in embryos expressing flapwing and heterozygous for *D*E-cad::GFP, where it is possible to appreciate the restoration of the *D*E-cad::GFP at the ZA in the AS of *foscrb_Y10A_* embryos. Regarding your last point, the analysis of the period of pulsed contractions, the low signal-to-noise ratio makes this analysis quite laborious and even arbitrary, as increased laser power lead to bleach and other technical problems (like cell-cell adhesions not really distinguishable).

Although it is reasonable to expect that the pulsed contractions will be damped by flapwing expression, this information is not related to the main aim of our work. Please bear in mind that this experiment is a “proof of principle”, in which we asked whether the expression of the myosin phosphatase leads to a suppression of the DC phenotype of the *foscrb_Y10A_* embryos. Other approaches could have been the expression of the MBS (another subunit of the myosin phosphatase), for example. Thus, we don’t see the justification to analyse the period of pulsed contractions (or other mechanical properties) of this tissue in these settings. As stated above, those analyses require the generation of other tools.

Nevertheless, we made very preliminary and crude analysis (2 embryos/genotype) and the DC rate after expression of Flw seems to be unaffected in *foscrb* embryos and normal in *foscrb_Y10A_* embryos (Figure 2).

*12) To understand the mechanisms by which myosin is phosphorylated, the authors turn their attention to three effectors of small GTPases in the Rho/Rac/Cdc42 family, Rok,* D*Pakand Scar. They present data in a table that is part of Figure 6. It is not easy to follow is only poorly referenced in the text. First, there are no error bars on the numbers presented. 71.3% or 70.5% really different than one another (probably not)? Are they different from 59.7 or 56.5 (without error bars, not possible to know)? My interpretation of the first 3 rows of the table is that the expression of* UAS-Actin-RFP*, under the control of an amnioserosal driver (I shouldn't have to reference another table to figure out where Gal4-332.2 is expressed) substantially suppresses the most severe dorsal open phenotype (i.e., from 59.7 to 18.5), but the authors don't comment on that. Perhaps they want us only to focus on the penetrance of the wt-like phenotype? If so they should explicitly say so and they should consider pooling the defects into a single dorsal open category, perhaps relegating the more detailed panel to supplementary material). Moreover, where they do explicitly draw our attention to changes, the comparisons seem flawed. They ask us to compare row 4 to row 3 to show that* UAS-crb *expression increases the most severe dorsal hole, but the embryos in row 4 don't express* UAS-actin-RFP *and the expression of* UAS-actin-RFP *greatly affects the number of animals in the most severe category (compare row 3 to rows 1 and 2). Similarly, rows 9 and 10 are compared to 3, but 9 and 10 don't express* UAS-actin-RFP*. Similarly, are differences seen with* UAS-moe *(without actin) vs. UAS phosphomimetic moe (with actin) simply due to the expression of actin? If I understand Figure 6 correctly, experiments are not being compared to appropriate controls.*

Thanks for the suggestion of pooling the defects into a single dorsal open category. We modified the main figure as such and provided the detail graph as supplement. We also included 3 more control crosses (Figure 7 and Figure 1, Bars 3-5) to have appropriate comparisons. Expression of *UAS-Actin::RFP* does not suppress the *foscrb_Y10A_* phenotype (compare 3^rd^, 5^th^ and 6^th^ bars) Also, now the graphs present the mean ± SD, while before it presented the overall percentage of the pooled experiments. Finally, we explained in more detail the comparisons in the Results.

*13) Nevertheless, some of the data show intriguing trends. Over expression of the autoinhibitory domain of* D*Pak suppresses the DC phenotype of Y10A animals, whereas over expression of* D*Pak induces a DC phenotype in otherwise normal embryos. Is it correct that autoinhibition of* D*Pak is intra-molecular?*

No, it is inter-molecular. The inhibitory domain in complex with PAK1 is a dimer between the auto-inhibitory region and the kinase domains. If you’re interested into the details of the interaction these papers describe it: Lei et al., 2002 Cell 102, 387–397 and Parrini et al., 2002 Mol Cell 9, 73–83. Also, these reviews may be of interest: Hofmann et al., 2004 J Cell Sci 117, 4343–4354; Zhao and Manser, 2005 Biochem J 386, 201–214; Kumar et al., 2006 Nat Rev Cancer 6, 459–471.

*If so isn't it surprising that the autoinhibitory domain is effective when presented as a separate protein? How overexpressed is the domain?*

By “How overexpressed is the domain?” do you mean that you want to know the abundance? If so, we think that any kind of measurement seems to be arbitrary. It will compare a “0” expression level (CTL) vs a GAL4-induced expression. Even if we see any signal in immunofluorescence or Western blot, it will only be informative of whether the Pak-AID is expressed or not.

*Other key data are the suppression of Y10A DC phenotypes by the overexpression of* D*E-Cad. 14) Subsequent analysis of the role of Arp2/3 is focused on Arp2/3's role in endocytosis and is interpreted vis-à-visthe endocytosis of* D*E-cad. Lamelipodia and filopodia are a key feature of both the leading edge and amnioserosal tissues and Arp2/3 may have key roles in their function independent of their role in endocytosis. Why do the authors attribute the effects of Arp2/3 solely on endocytosis?*

We include this possibility in the Discussion (third paragraph and subsection “Crb – a regulator of ZA integrity via actomyosin dynamics?”).

*15) Finally, the authors conclude that* crb *function impacts both actomyosin dynamics and adhesion, and that the two effects may be inter-related, but the title of their paper suggests only a role in regulation of actomyosin dynamics. It is not clear why this is appropriate. While they use the Discussion to argue their point, it seems that a revised Results text, with some experimental revision may alter the conclusions made in the Discussion substantially. At this point their title should be revised to more accurately reflect the firm conclusions made in the Results and leave the speculation in the Discussion clearly marked as such.*

The major conclusion did not change. However, we appreciate the reviewer’s suggestion and modified the title.

*In addition to these general comments, I make the following specific points for the Introduction and part of the Results. I incorporated comparable specific comments into the more general comments above for the remainder of the results but found that detailed comments were becoming both too numerous and too complicated.*

*In the subsection “The FBM of Crb is essential for dorsal closure”, first paragraph: The authors conclude the FERM binding domain of* crb *is responsible for the effects that they see because a point mutation (Y10A) in the FBD fails to rescue* crb *mutant function. Do they know that the alanine variant is made and correctly trafficked to the plasma membrane? If so, they should say how they know that and provide a reference. Otherwise conclusions regarding FBD function (vs.* crb *function as a whole) are erroneous.*

As explained above in your first comment, and explained in the first paragraph of the Results, this was presented in our previous work: Klose, S., Flores-Benitez, D., Riedel, F., and Knust, E. (2013). Fosmid-based structure-function analysis reveals functionally distinct domains in the cytoplasmic domain of Drosophila crumbs. G3 (Bethesda) 3, 153–165.

*Certainly dorsal closure fails in these Y10A animals (Figure 1), but germ band retraction is far from complete. Moreover, in normal embryos, the amnioserosa undergoes apoptosis upon completion of closure. Are the defects in closure observed due to the fact that at the posterior end the canthus does not form properly because of the incompletely retracted germ band and because the amnioserosa undergoes apoptosis thereby blocking closure? This puts a different spin on the defects observed.*

This is the same question raised in your comments 6 and 7; please see the corresponding responses.

*In the subsection “The FBM of Crb is essential for dorsal closure”, the authors state that "Of these protein species, Yrt appeared as a likely candidate responsible for the defects in the* foscrb_Y10A_
*mutant, since Yrt negatively regulates Crb (Laprise et al., 2006), zygotic* yrt *mutant embryos show DC defects (Hoover and Bryant, 2002), and over-expression of Crb in the amnioserosa results in DC defects (Harden et al., 2002; Wodarz et al., 1995)”. I don't understand how "over-expression of Crb in the amnioserosa results in DC defects" makes Yrt a likely candidate? The manuscript is characterized by a number of statements such as this one that distracts the reader and detracts from the message that the authors are trying to deliver.*

We rephrased the third paragraph to make it clearer (subsection “The FBM of Crb is essential for the integrity of the AS”). The idea is that Yrt negatively regulates Crb by interacting with its FBM directly, and *yrt* mutants (which leads to Crb over expression) or *UAS-Crb* over-expression in the AS produces similar DC defects. Thus, it was possible that the interaction of Yrt and Crb is affected in the *foscrb_Y10A_* embryos. The epistasis analysis shows that this possibility is unlikely (Video 7).

*In the subsection “The FBM of Crb is essential for dorsal closure”, fifth paragraph: The authors should show on the schematic in Figure 1 the orientation of the cross sections shown in Figure 1 panels M, N and O. Neither the text, nor the figure legend adequately describes the orientation of the section, and how it was obtained. Thus it is hard for the reader to understand what these panels are trying to show.*

We added a reference axis in Figure 1 and Figure 4—figure supplement 3, as well as the corresponding explanatory text in the figure legends.

*In the subsection “The FBM of Crb is essential for dorsal closure”, the authors state: "Moreover, in* foscrb *and* foscrb_Y10F_
*lacking zygotic* yrt *DC fails only after GB retraction, which is qualitatively different from the phenotype of* foscrb_Y10A_
*or* foscrb_Y10A_
*embryos lacking zygotic* yrt *(Video 4)". The authors should state explicitly how the phenotypes are qualitatively different and explicitly state whether or not the Y10A phenotype differs in the presence or absence of zygotic* yrt*. In addition the next sentence "These results suggest that the DC phenotype of* foscrb_Y10A_
*embryos starts earlier and is more complex than that in* yrt *mutants" requires the authors to give us more details about the* yrt *phenotypes, which are only referenced and not adequately described. I found this paragraph very hard to get through. The authors should consider reworking it for clarity.*

Done, see subsection “The FBM of Crb is essential for the integrity of the AS”, third paragraph.

*In the subsection “The FBM of Crb regulates filopodia formation and organisation of the supracellular actomyosin cable“, first paragraph: The authors should provide a reference or other explanation of why Sas stains filopodia.*

This has been moved to the subsection “The FBM of Crb regulates filopodia formation and organisation of the supracellular actomyosin cable in the DME cells”. This is our observation (both, in stainings and in live imaging – Video 4). Anyway, after revising the literature, this staining has been seen before in another work (Figure 4 in Denholm et al., 2005 Development 132:2389), but not explicitly mention in the text or the figure.

*Also, given that the core of microvilli consists of actin filaments, I don't understand why microvilli are not visible in the* foscrb *and Y10F animals (Figure 2). (Under appropriate conditions, ena might also be visible in the microvilli, but previous reports suggest it is visible primarily if ena is over expressed).*

Indeed, the confocal was adjusted in a way that only the signal from the cable is visible. The filopodia are there, but to see them, the gain and power of the laser have to be enhanced and the cable will appear as a blob completely out of range.

*In the subsection “The FBM of Crb regulates filopodia formation and organisation of the supracellular actomyosin cable”, the authors state: "The formation of the actomyosin cable at the LE depends on the asymmetric accumulation of the adhesion protein Echinoid (Ed) in the DME cells (Laplante and Nilson, 2011; Lin et al., 2007)”.As written only readers familiar with the referenced work will realize that the asymmetry of Ed protein in the DME cells is not normal and that it is asymmetry in the distribution of Ed between the epidermis and the amnioserosa that is relevant. They also state that "Removal of Ed from the LE is evident in* foscrb *and* foscrb_Y10F_
*embryos (Figure 2, arrowheads mark Ed absence at the LE)", but Ed is also not present at the leading edge of the DME cells in Y10A animals.*

We re-wrote the paragraph to make clear that the distribution of Ed in the *foscrb* embryos is like the one reported for wild type embryos. Also, we point out that the phenotype in the *foscrb_Y10A_* embryos is that the overall level of Ed in the DME is reduced, and not that Ed is absent from the LE (subsection “The FBM of Crb regulates filopodia formation and organisation of the supracellular actomyosin cable in the DME cells”).

*Moreover, the data presented in Figure 2 is not at all convincing. Baz is present at the leading edge of only one Y10A cell and it is possible that the dark line shown is background, like the line just above the arrow, deeper in the amnioserosa.*

This is addressed also in your point #4 above, and besides the line profile, in the Figure 2—figure supplement 3’ you may also see that both dark lines localise in the same plane as the phosphotyrosine staining, evidencing that it is not background.

*In the subsection “The FBM of Crb regulates filopodia formation and organisation of the supracellular actomyosin cable“, sixth paragraph: Gorfinkiel and Arias 2007 show very convincingly that* D*E-cad initially localizes to the LE, but later during closure is lost from the leading edge (see Gorfinkiel and Arias Figure 1). The authors need to verify that their observations on the fosmid rescue embryos are consistent with the literature and consider how clearing the leading edge of* D*E-cad is consistent with the mechanisms they propose here. Moreover, the quality of the* D*E-cad stained images lacks contrast (compare Gorfinkial and Arias, Figure Aiii and Biii to this manuscripts Figure 2). Better stained preparations might give better insight into* D*E-cad distributions in the Y10A animals.*

This is in part the same as the comment #5; see answer above. For the Figure 2, we acquired new images, which are now shown in Figure 2 and Figure 2—figure supplement 5.

*In the subsection “The FBM of Crb is essential for the morphogenesis of the AS”, second paragraph: The authors claim that there are discontinuities in* D*E-cad localization in Y10A animals which appears to be the case, although if the structure of the AS is considerably different than wildtype. How do the authors know that the* D*E-cad junctions haven't simply gone out of the plane of section? Moreover, the authors cite large aggregates of* D*E-Cad-GFP in the cytoplasm of the Y10A cells, but there seem to be a few such aggregates even in* foscrb *animals and large numbers of such aggregates in the Y10F animals (i.e., numbers comparable to those seen in the Y10A cells, see 3A' and 3B', respectively).*

To resolve this issue, we generated recombinants of *D*E-cad::mTomato with the different *foscrb* variants. This led us to analyse in a better signal-to-noise ratio condition the localisation of *D*E-cad (Figure 3 and Video 4). The previous images, using the *D*E-cad::GFP allele, were misleading some reviewers, because the auto-fluorescence of the yolk was taken as intracellular aggregates of *D*E-cad::GFP.

In the image provided in the first version of our manuscript we were able to tell them apart from the *D*E-cad intracellular aggregates using the original data because the yolk granules localise deep in the embryo. Now, employing the *D*E-cad::mTomato allele it is possible to show the maximal projection without so much interference of the yolk as its auto-fluorescence is lower in the emission spectrum of mTomato (around 560-600 nm). Nevertheless, Video 5 still shows the analyses with the *D*E-cad::GFP allele, but a note was added to the Video legend.

*In the subsection “The FBM of Crb is essential for the morphogenesis of the AS”, fifth paragraph, the authors state: "Apoliner expression in the AS of* foscrb *and* foscrb_Y10F_
*embryos (Video 9) revealed some apoptotic cells at the posterior canthus at the end of GB retraction (Figure 4, arrows). In* foscrb_Y10A_
*embryos of a similar age, some detached cells (Figure 4, arrowheads) as well as those at the posterior edge of the remaining AS (Figure 4, arrow) appear as apoptotic cells." First, the authors miss a key paper on apoptosis in the amnioserosa (Shen et al. 2013 PLoS ONE 8:e60180), which presents data in form that is much easier to interpret (see Shen et al. Figure 5).*

In our hands, and with the GAL4 driver employed, the GFP signal from Apoliner is not visible in the nucleus, but appears to be degraded (as reported in Kolahgar et al., 2011 Development 138:3021 – which is the same group that generated the Apoliner marker).

*To me, the data presented here suggest that despite substantial differences in AS morphology, that the level of apoptosis in Y10A animals is about the same as in* foscrb *and Y10F embryos (there may be some additional apoptosis at lesions in the AS).*

If you see the current Video 6 when it shows the time 210 min, and you take into account that those embryos were collected, incubated and imaged at the same time, you can observe that at 10-10.5 h after egg collection, the apoptosis levels in the AS is very different between *foscrb* and *foscrb_Y10A_* embryos. Therefore, we conclude that apoptosis is premature in the *foscrb_Y10A_* embryos when compared with time-matched controls (*foscrb* embryos).

*The authors go on to write "At the end of DC, the internalised AS cells are localised in a central rod-like structure in* foscrb *and* foscrb_Y10F_
*embryos and subsequently die by apoptosis (Figure 4) [as has been reported for wild type embryos (Reed et al., 2004)], while in* foscrb_Y10A_
*embryos of similar age, the remaining AS cells have completely disaggregated (Figure 4)." What do they authors mean by "a similar age"?*

We removed the term “similar age” from all the text. Instead, the reader is referred to the Material and methods for details about embryo staging (Results, second paragraph and Materials and methods, second paragraph). We now use “incubated for the same period of time” or “at this time point”.

*To me this is completely consistent with the interpretation that morphogenesis is delayed and the programed cell death of the AS occurs "normally" in the animals with all three* foscrb *variants.*

As discussed in the comment to your point #7, the delayed morphogenesis in the *foscrb_Y10A_* embryos seems very unlikely.

*The authors go on to state "To summarise, the AS in* foscrb_Y10A_
*embryos breaks apart and undergoes premature apoptosis". I agree that the AS breaks apart, but it is not clear from the data as described that the apoptosis is "premature".*

We think that with the details added to the text, the movies, the figure legends and the Materials and methods, these questions are answered.

*They next write "suggesting that an intact FBM is required for maintaining the elasticity of the AS". There is nothing in this paper that evaluates the "elasticity" of the AS. Again, they need to know that* crb *is folding properly to draw conclusions about a defective FBM (vs. a key role for crb) and they can at best replace "elasticity" with "integrity".*

We have addressed this issue (please see the subsection “The FBM of Crb is essential for the integrity of the AS”, first paragraph).

Reviewer 4:

*We know from previous work that the Crumbs FERM domain-binding motif (FBM) directly interacts with the FERM-domain of Yurt (Yrt) and Expanded (Ex) and recruits Moesin to the apical membrane. And we know that mutation of this domain has no effect on the many other tissues where Crumbs has a role, but has lethal phenotype in the embryo, which this study shows to be due to an overactive actomyosin network. The live-imaging of Cadherins using* D*E-cad::GFP knock-in allele in a* crb *null background with formed introduced variants of crumbs is well-executed, systematic and key part of the study from which the conclusions are derived. The authors cleanly demonstrate a Yrt independent role and then show that* foscrb_Y10A_*show defects in the establishment of the actomyosin apparatus cells in a situation where components of the Zonula Adherens are disturb or missing. Next the authors go on to show that the FBM of Crb is an important regulator of cytoskeletal activity and cell-cell adhesion of the amnioserosa and required for its proper morphogenesis. The next set of experiments are the 'meat' of the paper and they comprehensively address the role of Crumbs FBM in actomyosin dynamics and show that the FBM of Crb regulates actomyosin dynamics in the amnioserosa during dorsal closure by down-regulating the Rho1 pathway. Finally the authors show that the dorsal-closure phenotype of Crb_Y10A_ embryos also due to reduced adhesion. The authors have carefully dissected out Crumb function through the FBM. As they point out we do not yet know if the actomyosin phenotype and the reduced adhesion ones are linked or not. The phenotypes could result from defects in amnioserosa differentiation or in its attachment to the basal lamina. The authors raise this point themselves and have 'data not shown' to negate such conclusions. It will be good to see this data.*

Data was included as Figure 1 and Figure 4—figure supplement 2.

*While the work convincingly demonstrates a novel function of the FBM in cytoskeleton dynamics and also show that Crb regulation of amnioserosa morphogenesis involves* D*Moesin, Rho- GTPases, class-I Pak, and the SCAR-Arp2/3 complex, two questions remain. How does this regulation take place (three ways are suggested) and why other epithelia which require Crumbs does not seem to require this domain (while requiring the downstream partners). The key question then is whether the role of the FBM show is a peculiarity that details out an aspect of Crumbs or if it is telling us something broader about its role in epithelial morphogenesis. It will be good to see a discussion on this.*

In fact, this is an interesting question, the solution of which goes beyond the scope of this work. We raised this question at the end of the Discussion, but possible scenarios appeared too speculative and therefore we did not elaborate on this point.

[Editors' note: further revisions were requested prior to acceptance, as described below.]

Reviewer 2:

*[…] I particularly like Figure 7—figure supplement 1 more than new Figure 7 find the genotypes are easier to follow and think the subtleties of the DC defects are important. The addition of error bars is a good idea but in this case I would also add statistical significance of the differences observed.*

We analysed the statistical significance of the data presented in the Figure 7—figure supplement 1 by a one-way-ANOVA followed by a Dunnet’s multiple comparisons test. Because this analysis requires data that is continuously variable, the percentages were first converted to arcsin values and then analysed with GraphPad Prism. This is also described in the Material and methods, subsection “Statistical analyses”.

We consider that the addition of more symbols, arrows or asterisks to the graph in the Figure 7—figure supplement 1 will make it hard to read and understand, so we present the results of such analyses as a summary in the form of a Table (Table 1), and added a note to the legend of the Figure 7—figure supplement 1.

*I suggest Figure 5–figure supplement 2 goes to the main text: it's a very important piece of data and captures very nicely the behaviour of actomyosin foci.*

Done, It now corresponds to Figure 5.

Reviewer 3:

*[…] I still have 6 concerns that should be corrected before publication. The authors assert that Baz is retained at the leading edge of Y10A embryos. The micrograph they present in 2J is simply not compelling and the rebuttal they provide is simply not convincing. If Baz is preserved at the leading edge in Y10A embryos surely a micrograph that provides more compelling evidence should be easy to obtain? Compare their localization of Baz in wild type to that presented by Laplante and Nilson, 2011, which they cite.*

It appears that our text was not completely clear. There are two effects on Baz in *foscrb_Y10A_* embryos: i) we wrote that “*foscrb_Y10A_* embryos preserve Baz at the LE of those cells that fail to elongate (Figure 2, arrow)”. So there is no general loss of Baz at the LE, as written by the reviewer; ii) there is a general reduction of Baz in the junctions of *foscrb_Y10A_* embryos, as shown in Figure 2—figure supplement 3 and now written in the fourth paragraph of the subsection “The FBM of Crb regulates filopodia formation and organisation of the supracellular actomyosin cable in the DME cells” and added to Figure 2—figure supplement 3 legend.

The difference between our staining and the one presented in Laplante and Nilson, 2011 may be due to many different factors (concentrations, secondary antibodies, incubation times, image processing, etc.). We repeated the stainings, but we obtained the same results shown in the referred Figure 2 and Figure 2—figure supplement 3 (see Figure 3). The higher background in our stainings may be due to the fact that the antibody in Laplante and Nilson 2011 is a rat antibody (from Wodarz, A., Ramrath, A., Kuchinke, U., and Knust, E. (1999). Nature 402, 544; but Laplante and Nilson do not specify whether they used the rat anti-Baz C-term or the rat anti-Baz N-term) and we used a rabbit one generated against Baz-GST-N-Term containing aa 1-297 of Bazooka protein from *Drosophila*. We also avoided to increase the contrast, which we think was done in Figure 4’ and B’ of Laplante and Nilson, 2011.

In Figure 11, there are 3 different *foscrb* and *foscrb_Y10A_* embryos, respectively, stained for Bazooka and phospho-tyrosine (associated with the ZA). The arrowheads in the control embryos mark where Baz is absent from the LE (as has been shown in wild type embryos, Laplante and Nilson 2011). The arrows in the *foscrb_Y10A_* embryos show that most cells that do not elongate in the LE of *foscrb_Y10A_* embryos preserve the localisation of Baz at the LE, as we report in the Figure 2 and Figure 2—figure supplement 3.

Author response image 2.**DOI:**
http://dx.doi.org/10.7554/eLife.07398.047

Author response image 3.**DOI:**
http://dx.doi.org/10.7554/eLife.07398.048

These results are identical to the ones presented in the manuscript in the Figure 2 and Figure 2—figure supplement 3, thus we do not see the necessity to modify those figures.

*Figure 2 show schematics to help the reader figure out what the main features of the Y10A phenotypes are. There is little or no evidence for the presence of a conventional myosin in filopodia. Interestingly, although the Sas data show filopodia, neither the actin nor the myosin panels show filopodia very well (there seem to be some filopodia in the actin panel E). The schematic that shows "actomyosin" should be removed. Actin might be shown in filopodia the actomyosin cable, myosin should be shown only in the actomyosin cable.*

Corrected as suggested by Reviewer 1.

*Gorfinkiel and Martinez-Arias (2007) showed that while Cadherin is initially localized to the leading edge of the DME cells, that it is cleared with time. In my first review, I was afraid that the phenotype shown for loss of cadherin in Y10A animals was due to the normal clearing of cadherin from the leading edge later in closure. The authors provide a convincing rebuttal that they are comparing early stage* foscrb *animals to early stage Y10A animals. They should simply add a sentence to the revised manuscript to indicate that they know that cadherin is cleared at later stages and that this comparison is between different genotypes during early closure stages. The readers need to know this as much as do the reviewers.*

This has been corrected (please see the subsection “The FBM of Crb regulates filopodia formation and organisation of the supracellular actomyosin cable in the DME cells”, sixth paragraph).

While pulsing or oscillating amnioserosa cells characterize dorsal closure in wild type embryos and are very interesting, it is not clear to me that oscillations "are required" for closure. Thus, oscillations can be suppressed and closure still occurs. The sentence that begins “Besides elongation of the DME cells and integrity of the AS, DC requires pulsed contractions of the AS cells…” should be revised.

We revised the sentence to read “perturbing actomyosin dynamics of the AS cells interferes with normal DC” and also added a reference about the effects of perturbed actomyosin dynamics on the DC: Fischer et al. (2014) PLoS ONE 9, e95695.

*Figure 6 is referenced as a model for how the FDB of Crb might regulate cytoskeletal function. I am puzzled by some of the details. In particular, the model has Rok and* D*Pak as positive regulators of MLCK. My understanding is that MLCK, Rok and* D*Pak all phosphorylate sqh directly, so arrows from Rok and Pak to MLCK and not to sqh are at best misleading and more likely simply wrong (unless there are data in the literature that I simply don't know about). Second, Rok has an inhibitory connection to flapwing. Is that really the case? Does Rok phosphorylate Flapwing to inhibit its phosphatase activity or otherwise bind to Flapwing to sequester its activity?*

Corrected, now the arrows indicate that MLCK, Rok and *D*Pak all phosphorylate Sqh directly. We also added the *Drosophila* myosin-binding-subunit (*D*MBS) to the scheme, which is phosphorylated by Rok, resulting in the inhibition of the phosphatase activity. We also added the definitions of the abbreviations to the figure legend.